# MANISKILL-HAB: A BENCHMARK FOR LOW-LEVEL MANIPULATION IN HOME REARRANGEMENT TASKS

**Arth Shukla, Stone Tao & Hao Su**
Hillbot Inc. and University of California, San Diego
{arshukla,stao,haosu}@ucsd.edu

## ABSTRACT

High-quality benchmarks are the foundation for embodied AI research, enabling significant advancements in long-horizon navigation, manipulation and rearrangement tasks. However, as frontier tasks in robotics get more advanced, they require faster simulation speed, more intricate test environments, and larger demonstration datasets. To this end, we present MS-HAB, a holistic benchmark for low-level manipulation and in-home object rearrangement. First, we provide a GPU-accelerated implementation of the Home Assistant Benchmark (HAB). We support realistic low-level control and achieve over 3x the speed of prior magical grasp implementations at a fraction of the GPU memory usage. Second, we train extensive reinforcement learning (RL) and imitation learning (IL) baselines for future work to compare against. Finally, we develop a rule-based trajectory filtering system to sample specific demonstrations from our RL policies which match predefined criteria for robot behavior and safety. Combining demonstration filtering with our fast environments enables efficient, controlled data generation at scale.

**Videos, models, data, code, and more at** **http://arth-shukla.github.io/mshab**

## 1 INTRODUCTION

An important goal of embodied AI is to create robots that can solve manipulation tasks in home-scale environments. Recently, faster and more realistic simulation, home-scale rearrangement benchmarks, and large robot datasets have provided important platforms to accelerate research towards this goal. However, there remains a need for all of these features in one unified benchmark.

We present **MS-HAB**, a holistic, open-sourced, home-scale manipulation benchmark with four key features: (1) fast simulation with realistic physics and manipulation, including low-level control, for efficient training, evaluation, and dataset generation, (2) home-scale manipulation tasks through the Home Assisitant Benchmark (HAB) (Szot et al., 2021), (3) extensive baselines for future work to compare against, and (4) scalable, controlled data generation using an automated, rule-based trajectory filtering system.

**Fast Manipulation Simulation with Realistic Physics and Rendering:** Using ManiSkill3 (Tao et al., 2024), we implement a GPU-accelerated version of the HAB (Szot et al., 2021), an apartment-scale rearrangement benchmark containing three long-horizon tasks using the Fetch mobile manipulator (ZebraTechnologies, 2024). While the original HAB uses magical grasp (teleport closest object within 15cm to the gripper), we require realistic grasping.

The MS-HAB environments support low-level control for realistic grasping, manipulation, and interaction, while the original Habitat 2.0 implementation does not support such kind of low-level control. Furthermore, by scaling parallel environments, MS-HAB environments achieve over 4300 samples per second (SPS) while the robot actively collides with multiple dynamic objects and the environment renders 2 128x128 RGB-D images — 3x faster than Habitat 2.0 at a fraction of the GPU memory usage. This significant speedup allows us to scale training, evaluation, and data generation.

**Reinforcement Learning (RL) Baselines:** Online RL provides a promising framework to learn from online interaction without needing preexisting demonstration data. As in Gu et al. (2023a), we train individual mobile manipulation skills and chain them to solve long-horizon tasks. We

hand-engineer dense rewards for the Fetch embodiment, designed for low-level control with mobile manipulation. Furthermore, we train manipulation policies overfit to one specific object's geometry, outperforming all-object policies when grasping many objects or in conditions with tight tolerances depending on object geometry. Leveraging our fast environments, we run extensive RL baselines, training 150 policies across 3 seeds (50 policies/seed) with 1.83 billion environment samples.

**Automated Event Labeling and Trajectory Categorization:** We use privileged information from the simulator to distill trajectories into chronologically ordered lists of events (e.g. Pick events include 'Contact (object)', 'Grasped', 'Dropped', 'Success', and 'Excessive (robot) Collisions'). Using these events lists, we define mutually exclusive, collectively exhaustive success and failure modes. For example, Pick success mode "reach success $\wedge$ cumulative robot collisions $< 5000\,\mathrm{N}$ $\wedge$ object not dropped" requires events list (Contact, Grasp, Success) and forbids events 'Dropped' and 'Excessive Collisions'. We filter our dataset by selecting demonstrations labeled with success modes that guarantee particular behaviors (e.g. pick without dropping) and safety constraints (e.g. cumulative robot collisions $< 5000\,\mathrm{N}$). Furthermore, we provide trajectory categorization statistics for all baselines in Appendix A.6 so future work can gear its methodology to solve frequent failure modes discovered by our policies.

**Dataset Generation and Imitation Learning (IL) Baselines:** When generating our dataset, we use trajectory categorization to filter demonstrations without needing manual labor, and we provide Imitation Learning (IL) baselines using our dataset. Our results show that selecting demonstrations with particular behavior biases IL policies towards that behavior. Paired with our fast simulator, users can generate massive datasets and control demonstration type in fast wall-clock time.

**Summary of Contributions:** The contributions of MS-HAB are summarized as follows: 1) GPU-accelerated HAB implementation which supports realistic low-level control and achieves over 4300 SPS while interacting and rendering, 2) extensive RL and IL baselines, 3) automated event labeling and trajectory categorization, providing success and failure mode statistics for all baseline policies, and 4) efficient, controlled vision-based robot dataset generation at scale.

## 2 RELATED WORK

**Simulators and Scene-Level Embodied AI Platforms:** Earlier scene-level simulators focus on navigation and simple interaction with realistic visuals (Savva et al., 2019). Other simulators add kinematic object state transitions (Kolve et al., 2017; Li et al., 2021), significant scene randomization (Deitke et al., 2022; Nasiriany et al., 2024), soft-body physics and audio (Gan et al., 2022), flexible and deformable materials, object composition rules, and so on (Li et al., 2022). However such complicated features often slow down simulation speed.

Habitat 2.0 forgoes additional features, supporting rigid-body dynamics, articulations, and magical grasping, to achieve best-in-class single-process scene-level simulation speed (Szot et al., 2021). However, it is constrained by the limited parallelization of CPU simulation.

Other simulators focus on low-level, contact-rich control in simpler settings (James et al., 2020; Zhu et al., 2020; Xiang et al., 2020). ManiSkill3 in particular achieves state-of-the-art GPU simulation speed (Tao et al., 2024), however its suite of tasks are simpler than the Home Assistant Benchmark (HAB) (Szot et al., 2021), which we implement for MS-HAB.

**Scalable Demonstration Datasets:** Real-world robot datasets are promising for direct deployment to the real world (Brohan et al., 2023). However, these initiatives are limited in scaling and use cases due to small-scale toy setups (Ebert et al., 2022), vision-only data (Dasari et al., 2019), or requiring massive coordinated (et al., 2024; Khazatsky et al., 2024) or distributed (Mandlekar et al., 2018) human effort over many months or even years. Furthermore, real robot datasets cannot efficiently generate new data, and do not support online sampling.

Generative interactive world models allow some interactivity on similarly realistic data by generating new frames based on provided actions (Yang et al., 2024). However, these models suffer from artifacts and long-term memory issues which rule out home-scale rearrangement, and low frame rates make training high-frequency low-level control policies intractable. Furthermore, neither real-robot datasets nor generative world models currently support querying privileged data from a simulator, which is necessary for MS-HAB's automated event labeling and trajectory categorization.

Meanwhile, classical physical simulation supports data generation from a variety of sources (expert teleoperated, suboptimal human, etc), and machine-generated data is largely scalable; however, datasets like Fu et al. (2020); Mandlekar et al. (2021); Gu et al. (2023b) only support smaller-scale continuous control tasks.

RoboCasa combines different aspects of above approaches (Nasiriany et al., 2024): a physical simulator, diverse AI-generated textures and models, 1250 human-teleoperated demonstrations, and MimicGen to scale data (Mandlekar et al., 2023). However, RoboCasa achieves only 31.9 SPS *without rendering*, does not support filtering trajectories by behavior, and its demonstrations alternate between manipulation and navigation. Meanwhile, we achieve 125x faster simulation *while rendering* 2 128x128 RGB-D images *and interacting* with multiple dynamic objects. We also support automated filtering under customizable constraints, and our demonstrations use whole-body control.

**Skill Chaining:** Chen et al. (2023) and Lee et al. (2021) use finetuning methods to bias the initial and terminal state distributions to increase handoff success while skill chaining. However, these methods are applied to tasks with unchanging order (e.g. furniture assembly, block orient/grasp/insert). Meanwhile, Gu et al. (2023a) formulate composable and reusable skills with mobility to create greater overlap in initial and terminal state distributions, achieving better results than stationary manipulation. However, Gu et al. (2023a) uses magical grasp with online RL, while we provide RL and IL baselines, and we include additional considerations for low-level grasping, such as new rewards and subtask success conditions, sampling grasp poses from Pick policies, and overfitting object manipulation policies to specific object geometries.

## 3 Preliminaries

### 3.1 Tasks, Subtasks, and Policies

The Home Assistant Benchmark (HAB) (Szot et al., 2021) includes three long-horizon tasks which involve rearranging objects from the YCB dataset (Çalli et al., 2015):

- **TidyHouse:** Move 5 target objects to different open receptacles (e.g. table, counter, etc).
- **PrepareGroceries:** Move 2 objects from the opened fridge to goal positions on the counter, then 1 object from the counter to the fridge.
- **SetTable:** Move 1 bowl from the closed drawer to the dining table and 1 apple from the closed fridge to the same dining table.

To solve these tasks, Szot et al. (2021) define parameterized skills: Pick, Place, Open Fridge/Drawer, Close Fridge/Drawer, and Navigate. For each skill, we define corresponding subtasks. Successful low-level grasping is heavily dependent on an object's pose. So, depending on the subtask, the simulator provides ground-truth pose $x_{pose} = [x_{rot}|x_{pos}]$ for target object $x$, ground-truth handle position $a_{pos}$ for target articulation $a$, or 3D goal position $g_{pos}$, updated each timestep during manipulation. Each subtask also fails if the robot cumulative force reaches beyond a set threshold. For more details, see Appendix A.1. We provide brief descriptions of the subtasks below:

- **Pick**[$a$, optional]($x_{pose}$)**:** pick object $x$ (from articulation $a$, if provided).
- **Place**[$a$, optional]($x_{pose}$ , $g_{pos}$)**:** place object $x$ in goal $g$ (in articulation $a$, if provided)
- **Open**[$a$]($a_{pos}$)**:** open articulation $a$ with handle at $a_{pos}$
- **Close**[$a$]($a_{pos}$)**:** close articulation $a$ with handle at $a_{pos}$
- **Nav**($*_{pos}$)**:** navigate to $*$

From a reinforcement learning perspective, we formulate each long-horizon task as a standard Markov Decision Process (MDP) which can be described as a tuple $\mathcal{M} = (\mathcal{S}, \mathcal{A}, \mathcal{R}, \mathcal{T}, \rho, \gamma)$ with continuous state space $\mathcal{S}$, action space $\mathcal{A}$, scalar reward function $\mathcal{R} : \mathcal{S} \times \mathcal{A} \to \mathbb{R}$, environment dynamics function $\mathcal{T} : \mathcal{S} \times \mathcal{A} \to \mathcal{S}$, initial state distribution $\rho$, and discount factor $\gamma \in [0, 1]$. Then, as in Gu et al. (2023a), define a subtask $\omega$ as a smaller MDP $(\mathcal{S}, \mathcal{A}_\omega, \mathcal{R}_\omega, \mathcal{T}, \rho_\omega, \gamma)$ derived from $M$. For each task $M$ with subtask $\omega$, we train low-level control policy $\pi_\omega : \mathcal{S} \to \mathcal{A}_\omega$ with RL or IL.

In this work, we study a partially observable variant of each task, where the policy must use 2 128x128 depth images to infer collisions and obstructions. We train different policies for each task/subtask combination (i.e., TidyHouse Pick, PrepareGroceries Pick, etc). Additionally, we train Pick and Place RL policies to overfit to specific objects, i.e., one policy for each task/subtask/object combination. Since this work focuses on low-level control, we use a teleport for the Navigation subtask. Additional details on training policies and teleport navigation are provided in Sec. 5.1.

## 3.2 SKILL CHAINING

Similar to Szot et al. (2021), we split each task into a sequence of subtasks using a perfect task planner. The sequences are defined below:

- **TidyHouse:** For $(x_i, g_i) \in \{(x_0, g_0), \ldots, (x_4, g_4)\}$, complete:
  $\mathrm{Nav}(x_{i,pos}) \to \mathrm{Pick}(x_{i,pose}) \to \mathrm{Nav}(g_{i,pos}) \to \mathrm{Place}(x_{i,pose}, g_{i,pos})$

- **PrepareGroceries:** For $(x_i, g_i) \in \{(x_0, g_0), (x_1, g_1), (x_2, g_2)\}$, complete:
  $\mathrm{Nav}(x_{i,pos}) \to \mathrm{Pick}_{\mathrm{Fr}[i \leq 1]}(x_{i,pose}) \to \mathrm{Nav}(g_{i,pos}) \to \mathrm{Place}_{\mathrm{Fr}[i=2]}(x_{i,pose}, g_{i,pos})$

- **SetTable:** For $(x_i, g_i, a_i) \in \{(x_0, g_0, \mathrm{Dr}), (x_0, g_0, \mathrm{Fr})\}$, complete:
  $\mathrm{Nav}(a_{i,pos}) \to \mathrm{Open}_{a_i}(a_{i,pos}) \to \mathrm{Nav}(x_{i,pos}) \to \mathrm{Pick}(x_{i,pose}) \to \mathrm{Nav}(g_{i,pos}) \to \mathrm{Place}(x_{i,pose}, g_{i,pos}) \to \mathrm{Nav}(a_{i,pos}) \to \mathrm{Close}_{a_i}(a_{i,pos})$

## 3.3 TRAIN AND VALIDATION SPLITS

The ReplicaCAD dataset (Szot et al., 2021) serves as the source for our apartment scenes. It comprises 105 scenes divided into 5 macro-variations, each containing 21 micro-variations. Macro-variations alter the layout of large furniture items such as refrigerators and kitchen counters, while micro-variations modify the placement of smaller furnishings like chairs and TV stands. The dataset is split into three parts: 3 macro-variations for training, 1 for validation, and 1 for testing. However, as the test split is not publicly accessible, our study utilizes only the train and validation splits.

Furthermore, for each long-horizon task, HAB provides 10,000 training episode configurations and 1,000 validation configurations. These configurations specify initial poses for YCB objects and define target objects, articulations, and goals. Importantly, these configurations exclusively utilize ReplicaCAD scenes from their respective splits.

# 4 ENVIRONMENT DESIGN AND BENCHMARKS

By scaling parallel environments with GPU simulation, MS-HAB achieves 4300 SPS on a benchmark involving representative interaction with dynamic objects — 3x Habitat 2.0's implementation. Our environments support realistic low-level control for successful grasping, manipulation, and interaction, while the Habitat 2.0 environments do not support such kind of low-level control. This section outlines environment design choices which leverage GPU acceleration and benchmarks MS-HAB against Habitat's implementation.

## 4.1 ENVIRONMENT DESIGN

**Evaluation and Training Environments:** First, we provide the base evaluation environment, `SequentialTask`, which supports executing different subtasks simultaneously on GPU. We perform physics simulation and rendering for all environments in parallel, then slice data by subtask to compute success and fail conditions. It does not support dense reward or spawn selection/rejection.

Second, we provide training environments for each subtask, `{SubtaskName}SubtaskTrain`, which extend the main evaluation environment. Each training environment provides dense rewards hand-engineered for the Fetch embodiment, supports spawning with randomization and rejection, and incorporates any additional features needed for training specific subtask skills.

**Observation Space:** We include target object pose, goal position, and TCP pose relative to the base, an indicator of whether the target object is grasped, 128x128 head and arm RGB-D images, and robot proprioception. For our experiments, we use only depth images. As is standard for the ManiSkill

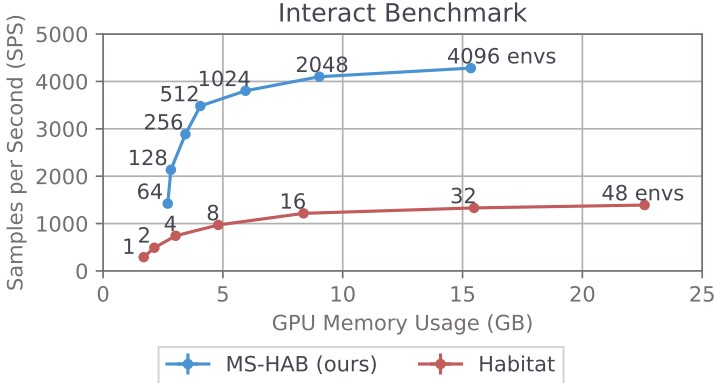

Figure 1: Interact benchmark comparing MS-HAB (ours) with Habitat. Each data point is annotated with the number of parallel environments used. SPS and GPU memory usage for each data point are averaged over 10 seeds; error bars representing 95% CIs are plotted, but are too small to see. Thanks to GPU acceleration, MS-HAB scales parallel environments to achieve over 3x the performance of Habitat while using a fraction of the GPU memory.

suite of tasks, the simulator computes ground-truth poses. We keep a consistent observation space across all subtasks via masking to support different subtasks running in parallel.

**Action Space:** We fully actuate the Fetch embodiment's arm, torso, and head pan/tilt joints. We support joint-based controllers and end-effector-based controllers. For our experiments, we use a PD joint delta position controller for the arm, torso, and head joints. The agent provides linear and angular velocity to control the base. The action space is normalized to $[-1, 1]$.

**Additional Details:** Our environments load the ReplicaCAD dataset provided by Habitat 2.0. However, since Habitat 2.0 uses magical grasp, the original ReplicaCAD dataset's collision meshes do not include handles for the kitchen drawers and fridge. So, we alter these collision meshes to include handles based on the provided visual meshes. We additionally provide navigable position meshes for the Fetch embodiment with Trimesh, as ManiSkill3 does not currently support navmeshes.

## 4.2 BENCHMARKING

We adapt Habitat 2.0's Interact benchmark, which originally had the Fetch robot execute a precomputed trajectory to collide with two dynamic objects (Szot et al., 2021). While we retain the same precomputed trajectory, assets, and scene configuration, we modify the robot's initial pose and disable magical grasp, allowing it to interact with five objects instead. Our setup includes two mounted 128x128 RGB-D cameras, with a simulation frequency of 100Hz and a control frequency of 20Hz (the standard for low-level control in ManiSkill3). We collect observation data from vectorized environments at each `step()` call. Our benchmarking is conducted on a machine equipped with a 16-core/32-thread Intel i9-12900KS processor and an Nvidia RTX 4090 GPU with 24 GB VRAM.

It is important to note that running the *exact* same episode in different simulators is exceedingly difficult since different simulation backends will result in interactions and collisions behaving slightly differently. Still, the full rollout is similar in both simulators, and the measured performance increase of MS-HAB in an interactive setting is significant.

**Habitat's Additional Optimizations:** While the Habitat simulator already has best-in-class single process simulation speed, it provides optional additional optimizations: concurrent rendering and auto sleep. However, their experiments suggest that concurrent rendering can negatively impact train performance (Szot et al., 2021), so we enable auto-sleep and disable concurrent rendering.

**Benchmark Analysis:** Per Fig. 1, while Habitat achieves stronger performance per parallel environment, its peak performance is limited to $1397.65 \pm 11.02$ SPS at 22.60 GB VRAM due CPU simulation's parallelization limitations and a less efficient renderer. Meanwhile, by scaling up to 4096 environments, MS-HAB is able to achieve $4299.18 \pm 26.36$ SPS at 15.35 GB VRAM usage

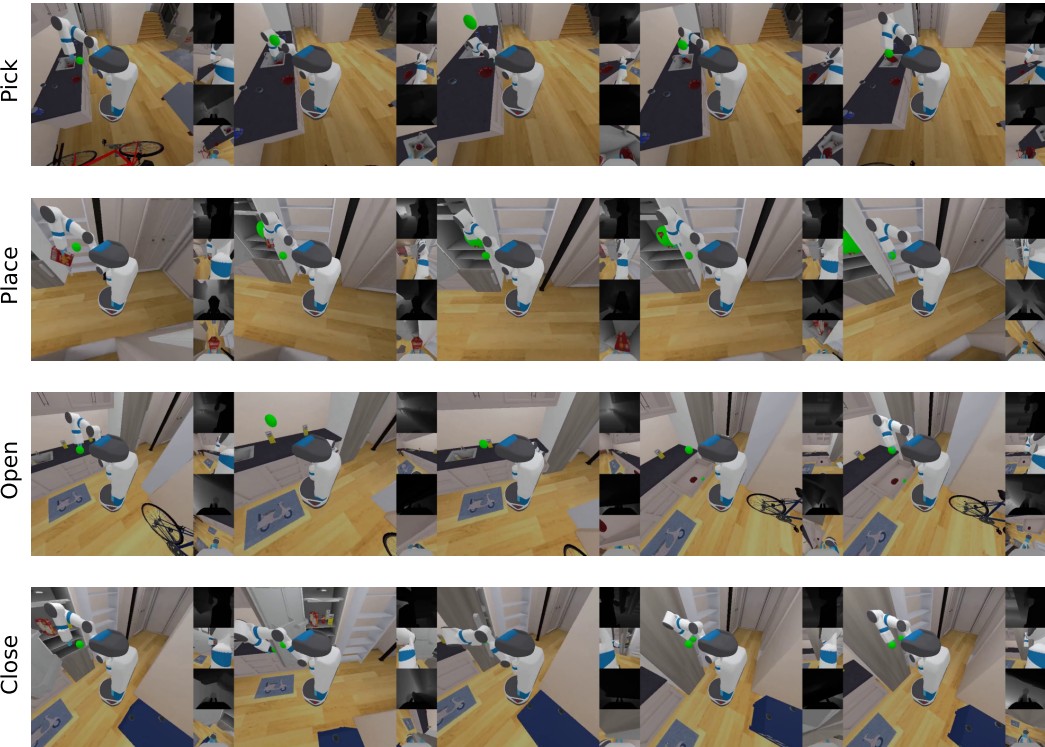

Figure 2: Renders of low-level, whole-body control policies solving Pick, Place, Open, and Close subtasks. We render 1 512x512 image and 4 128x128 sensor images. Note the base's moving position relative to surroundings. Goal spheres are invisible to sensors. Full videos in supplementary.

— 3.08x faster with 32% less VRAM. Furthermore, our environments support realistic low-level control for successful grasping, manipulation, and interaction, while the Habitat 2.0 environments do not support such kind of low-level control.

## 5 METHODOLOGY

### 5.1 TRAINING REINFORCEMENT LEARNING POLICIES

We choose Reinforcement Learning (RL) to learn our subtask policies as RL does not require prior demonstration data, and it can take advantage of our highly parallelized environments to solve tasks in fast wall-clock time. We use a similar subtask formulation as M3, which trains mobile manipulation skills to solve each subtask from a region of spawn points.

**Pick:** Without magical grasp, our Pick policies must learn grasp poses which are valid, stable, and reachable within the kinematic constraints of the mobile Fetch robot. Furthermore, the policy must learn action sequences which can reach these grasp poses and retrieve the target object within the specified horizon while keeping the robot under the cumulative collision force limit.

As a result, learning successful grasping for multiple objects with different geometries — in addition to whole body control with collision constraints — is difficult. So, we opt to train individual Pick policies for each object, thereby overfitting to the geometry of that object. Our experiments show these per-object Pick policies achieve improved subtask success rates compared to all-object policies when handling many objects with varied geometries. In other words, we train a unique per-object Pick policy for every task/subtask/object combination.

**Place**: We train per-object Place policies as well. Our experiments show that, in settings where object geometry is more important (e.g. placing in a fridge with tighter tolerances), per-object Place policies reach higher success rates than all-object policies.

Additionally, without magical grasp, there is not a ground-truth means of spawning the robot while grasping an object. So, we train our Pick policies before Place, then we sample grasp poses from our Pick policies to initialize the robot in the Place subtask.

**Open and Close:** Following (Gu et al., 2023a), we train different Open and Close policies for the kitchen drawer and the fridge. Furthermore, we find that opening the kitchen drawer is particularly difficult due to its small handle. So, we perturb the initial state distribution of our Open kitchen drawer subtask during training to accelerate learning: 10% of the time, we initialize the kitchen drawer opened 20% of the way. During evaluation, we do not alter the initial states.

**Algorithms and Hyperparameters:** We stack 3 consecutive frames for image observations to handle partial observability.

We train Pick and Place using SAC (Haarnoja et al., 2018; Xing, 2022) with a 1m replay buffer size. Visual observations are encoded by D4PG's 4-layer CNN (Barth-Maron et al., 2018) and concatenated with state observations. Actor and critic networks are 3-layer MLPs and the critic has LayerNorm to avoid value divergence (Ball et al., 2023). We train Pick with 50M timesteps and Place with 25M timesteps.

We train Open and Close using PPO (Schulman et al., 2017; Huang et al., 2022). Visual observations are encoded by a NatureCNN (Mnih et al., 2015) and concatenated with state observations. The actor and critic networks are 2-layer MLPs. We train Open Fridge with 15M timesteps, Open Drawer with 50M timesteps, Close Fridge with 25M timesteps, and Close Drawer with 15M timesteps.

We train 3 seeds for each task/subtask/object combination, evaluating on 189 episodes every 100,000 train samples. We select the checkpoint with highest evaluation success once rate as our final policy.

**Metrics:** We run 1000 episodes for every evaluation run (task/subtask evaluation, ablations, etc).

We evaluate subtask policies (Pick, Place, Open, Close) by success once rate (%), which is the percentage of trajectories that achieve success at least once in an episode with 200 timesteps. We evaluate long-horizon task success (TidyHouse, PrepareGroceries, SetTable) by Progressive Completion Rate (%). Here, the success of each subtask requires the success of every previous subtask. Hence, the completion rate of the final subtask is the completion rate of the entire long-horizon task.

Furthermore, since we are primarily interested in low-level control and manipulation, we replace navigation with a simple teleport. The robot is teleported to the target location with the same arm, base pose, and spawn location randomizations as in subtask training, described in Appendix A.1. In long-horizon tasks, we move to the next subtask as soon as the current subtask achieves success.

## 5.2 AUTOMATED TRAJECTORY CATEGORIZATION AND DATASET GENERATION

Thanks to fast simulation environments, we can quickly generate 10s to 100s of thousands of demonstrations. However, our experiments show that our Imitation Learning (IL) policies are sensitive to demonstration behavior. To filter out "suboptimal" demonstrations, we use privileged information from our simulator to group demonstrations into mutually exclusive, collectively exhaustive success and failure modes without significant manual labor. Furthermore, we use this trajectory labeling system to identify types and causes of failure in our baseline policies in Sec. 6.2 and Appendix A.6.

**Example of Pick Subtask:** We provide a high-level overview of trajectory labeling on the Pick subtask. For detailed definitions of events and labels, see Appendix A.6. First, we define "events" which occur at any timestep $t$: 1) Contact: nonzero robot/target pairwise force, 2) Grasped: object not grasped at step $t-1$ and grasped at step $t$, 3) Dropped: object grasped at step $t-1$ and not grasped at step $t$, and 4) Excessive Collisions: robot cumulative force exceeds $5000\,\mathrm{N}$. For Pick trajectory $\tau_{pick} = (s_0, a_0, \ldots, s_n, a_n)$, we create chronologically ordered event list $E_{pick} = (e_1, \ldots, e_k)$.

Next, we define success and failure modes. For example, one success mode is "straightforward success" with $E_{pick} = $ (Contact, Grasp, Success), requiring success without dropping the object or colliding too much. One failure mode is "dropped failure," defined as (Excessive Collisions $\notin E_{pick}) \wedge ($Dropped $\in E_{pick}) \wedge (i < j$ for maximal $i, j$ such that $e_i = $ Grasped, $e_j = $ Dropped). "Dropped failure" trajectories fail because the robot irrecoverably drops the target object.

In generating our Pick subtask dataset, we apply filters to include only "straightforward success" trajectories. These trajectories are characterized by the absence of dropping and minimal collisions.

As the dataset generation code is publicly available, users have the flexibility to create their own datasets with custom constraints tailored to their specific requirements.

**Imitation Learning Baselines:** We train IL baselines on our dataset using Behavior Cloning (BC) (Bain & Sammut, 1995; Ross et al., 2011; Daftry et al., 2016) (implementation based on Huang et al. (2022)). Visual observations are encoded by the 5-layer CNN from Bojarski et al. (2016), then concatenated with state observations. The actor network is a 3-layer MLP.

**Dataset Size:** With fast environments and automated trajectory filtering, users can generate as many 10s or 100s of thousands of demonstrations as needed in a controlled manner. For our baselines, we generate 1000 demonstrations per task/subtask/object combination using our per-object RL policies on the train split. The filters used are listed in Appendix A.6.1, with definitions in Appendix A.6.2.

## 6 RESULTS

### 6.1 BASELINES

Fig. 3 shows the RL and IL policies' progressive completion rate. We provide an optimistic upper bound on progressive completion rate by (incorrectly) assuming that the completion of each subtask is independent of every other subtask, thus directly multiplying subtask success once rates. Table 1 shows success once rate for individual subtasks. We find 4 major avenues for improvement.

First, our optimistic upper bound shows low expected success rate on the long-horizon tasks. Even with per-object RL policies, our low-level mobile manipulation subtasks are difficult to train on dense reward, and improving subtask success rate is the most direct way to improve overall task completion rate. Second, TidyHouse and SetTable RL baselines have some gap between upper bound and real completion rate, indicating potential handoff issues or disturbance to prior target objects in success states. Meanwhile, the PrepareGroceries RL baseline has a large drop in completion rate during the second $Pick_{Fr}$ subtask, indicating that the first $Pick_{Fr}$ causes too much disturbance to objects in the fridge. So, improving policy performance in cluttered spaces is important. Third, our IL policies perform notably worse in Pick and Place, indicating a need for methods and architectures which can handle multimodalities in the data. Finally, while most RL policies generalize well to the validation split, the Close Fridge policy completely fails on validation scenes because the fridge door opens into a wall, preventing the arm from reaching the handle. This is not an issue with magical grasping (Gu et al., 2023a), indicating that low-level control may need more scene diversity.

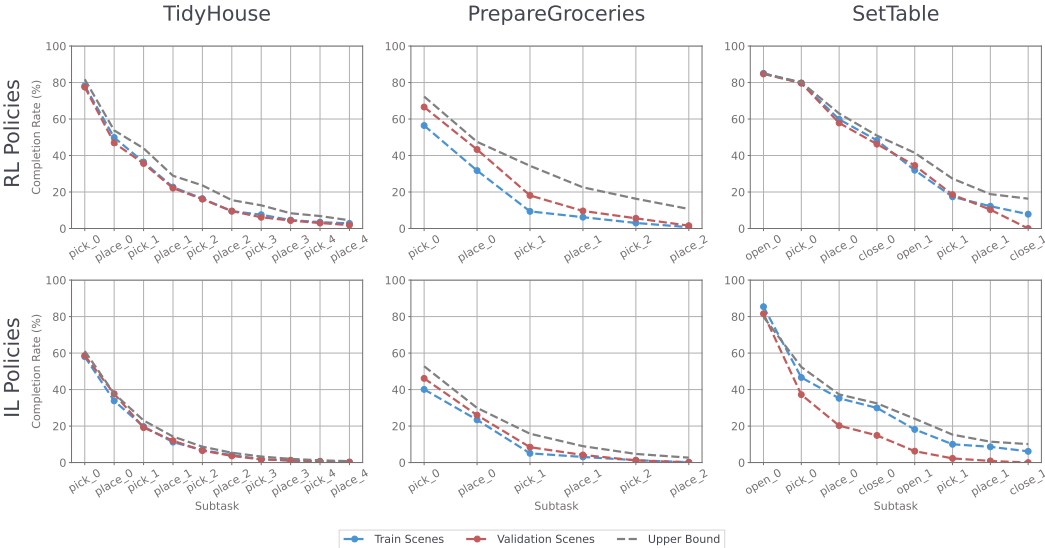

Figure 3: Long-horizon task progressive completion rates (%) on train and validation splits averaged over 1000 episodes. Futhermore, we provide an 'upper bound' on performance based on the success rates of each subtask policy. Best viewed zoomed.

Table 1: Subtask success once rates for RL and IL baselines. The RL-Per vs All column shows the difference in per-object RL policy performance and its all-object counterpart. We do not train all-object policies for Open or Close subtasks.

| TASK | SUBTASK | SPLIT | RL-PER | RL-ALL | IL | RL-PER vs ALL |
|---|---|---|---|---|---|---|
| TidyHouse | Pick | Train | 81.75 | 71.63 | 61.11 | +10.12 |
| | | Val | 77.48 | 68.15 | 59.03 | +9.33 |
| | Place | Train | 65.77 | 63.69 | 61.81 | +2.08 |
| | | Val | 65.97 | 66.07 | 63.79 | -0.10 |
| Prepare Groceries | Pick | Train | 66.57 | 51.88 | 44.64 | +14.69 |
| | | Val | 72.32 | 62.10 | 52.78 | +10.22 |
| | Place | Train | 60.22 | 53.37 | 50.00 | +6.85 |
| | | Val | 65.67 | 58.63 | 56.75 | +7.04 |
| SetTable | Pick | Train | 80.85 | 75.69 | 60.71 | +5.16 |
| | | Val | 88.49 | 79.86 | 72.62 | +4.63 |
| | Place | Train | 73.31 | 72.82 | 71.23 | +0.49 |
| | | Val | 67.06 | 68.25 | 62.20 | -1.19 |
| | $\text{Open}_{Fr}$ | Train | 83.43 | - | 74.01 | - |
| | | Val | 88.10 | - | 53.67 | - |
| | $\text{Open}_{Dr}$ | Train | 84.92 | - | 79.86 | - |
| | | Val | 84.52 | - | 78.57 | - |
| | $\text{Close}_{Fr}$ | Train | 86.81 | - | 86.90 | - |
| | | Val | 0.00 | - | 0.00 | - |
| | $\text{Close}_{Dr}$ | Train | 88.79 | - | 88.39 | - |
| | | Val | 89.29 | - | 87.60 | - |

## 6.2 ABLATIONS

### 6.2.1 RL POLICIES: ALL-OBJECT VS PER-OBJECT POLICIES

The goal of training per-object RL policies for Pick and Place is to improve subtask success rate since policies with higher success rates allow us to generate successful demonstrations under more initialization conditions. To verify this, we run two ablations.

**Does training per-object Pick and Place policies improve subtask success rate compared to all-object policies?** Per Table 1, per-object policies perform notably better in TidyHouse and Prepare Groceries Pick, which involve 9 objects, with more modest improvement in SetTable Pick, which has only 2 objects. Per-object policies perform significantly better in PrepareGroceries Place, which involves placing with tight tolerances on a cluttered fridge shelf, while performance differences are negligible in TidyHouse and SetTable Place, which only involve open receptacles. So, per-object Pick and Place policies learn improved manipulation when grasping a greater variety of objects, or when manipulating objects in areas with tighter constraints.

**Are per-object policies necessary to learn grasping for certain objects in the Pick subtask?** In Table 2, we run our automated trajectory labeling system on Pick YCB object #003, the Cracker Box (Çalli et al., 2015). The Fetch robot's parallel gripper can only grasp the Cracker Box along its shortest dimension, so the set of valid grasp poses are highly dependent on the object's pose relative to the robot. The all-object policy is 1.88-2.42x more likely to fail to excessive collisions and 1.87-12.37x more likely to fail to grasp the object, indicating that overfitting to a specific geometry is important for our RL policies to learn grasping on difficult geometries. For more detailed trajectory labeling definitions and statistics, please see Appendix A.6.

### 6.2.2 IL POLICIES: LABELING AND FILTERING DATASET TRAJECTORIES

**IL Policies: Can we control the behavior of our IL policies by filtering for specific demonstrations?** Our PrepareGroceries Place RL policies have two similarly frequent success modes: place

Table 2: Trajectory labeling on Pick Cracker Box with all and per-object RL policies. We group the trajectories into four categories: success once (**S-Once**), excessive collision failure (**F-Col**), cannot grasp failure (**F-Grasp**), and other failure modes (**F-Other**). We provide the percentage of trajectories which fall into each category, and each row sums to 100% (barring any rounding errors).

| TASK | SPLIT | TYPE | S-ONCE | F-COL | F-GRASP | F-OTHER |
|------|-------|------|--------|-------|---------|---------|
| TidyHouse | Train | RL-All | 29.46 | 34.52 | 28.17 | 7.85 |
| | | RL-Per | 71.63 | 17.26 | 2.48 | 8.63 |
| | Val | RL-All | 33.73 | 33.13 | 24.50 | 8.64 |
| | | RL-Per | 73.41 | 16.67 | 1.98 | 7.94 |
| Prepare Groceries | Train | RL-All | 11.51 | 60.62 | 16.17 | 11.70 |
| | | RL-Per | 51.98 | 25.10 | 8.63 | 14.29 |
| | Val | RL-All | 14.19 | 57.24 | 26.88 | 1.69 |
| | | RL-Per | 56.15 | 30.46 | 9.72 | 3.67 |

in goal (release the object within 15cm of $g_{pos}$) and drop to goal (release beyond 15cm). Although MS-HAB does not simulate state transitions like breaking, placing objects without dropping is a desirable, safe robot behavior to avoid excessive damage.

We generate 3 datasets with 500 demonstrations per object: 1) place in goal only, 2) drop in goal only, and 3) 50/50 split ("place", "drop", and "split"). We fit IL policies to each dataset and run trajectory labeling to determine policy behavior, shown in Table 3. The place and drop policies show bias towards executing place and drop trajectories respectively, but still perform the opposite behavior somewhat frequently. The split policy is somewhat biased towards dropping, likely because the 1-dim gripper action to drop is easier to learn under MSE loss than a 7-dim arm action to place.

Table 3: Success once rate (**S-Once**, %) and ratio of "place in goal" to "drop to goal" trajectories (**Place : Drop**). Note that some success trajectories are not labeled place in goal or drop to goal, as there are other possible success modes described in Appendix A.6.

| FILTERS | SPLIT | S-ONCE | PLACE : DROP |
|---------|-------|--------|--------------|
| Place in goal | Train | 45.73 | 3.17 : 1 |
| | Val | 54.46 | 2.55 : 1 |
| Drop to goal | Train | 49.21 | 1 : 2.22 |
| | Val | 51.19 | 1 : 2.86 |
| 50/50 Split | Train | 50.30 | 1 : 1.71 |
| | Val | 55.56 | 1 : 1.41 |

Thus, data filtering can generally influence IL policy behavior, but additional methods are needed to fully control behavior (e.g. online finetuning, reward relabeling, more advanced IL methods, etc).

## 7    CONCLUSION AND LIMITATIONS

We present MS-HAB, a holistic home-scale rearrangement benchmark including a GPU-accelerated implementation of the HAB which supports realistic low-level control, extensive RL and IL baselines, systematic evaluation using our trajectory labeling system, and demonstration filtering for efficient, controlled data generation at scale. However, there is significant room for improvement on our baselines, and we do not claim transfer to real robots; both of these we leave to future work. Whole-body low-level control under constraints in cluttered environments, long-horizon skill chaining, and scene-level rearrangement are challenging for current robot learning methods; we hope our benchmark and dataset aid the community in advancing these research areas.

## 8 ACKNOWLEDGEMENTS

Thank you to members of Hillbot, Inc. and the Hao Su Lab for providing valuable feedback on the project.

## 9 REPRODUCIBILITY STATEMENT

We open source all code for environments, training, evaluation, and data generation, and we release our dataset for public use, all available on our website: **http://arth-shukla.github.io/mshab**.

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

# A APPENDIX

## A.1 SUBTASK DEFINITIONS AND INITIALIZATION

### A.1.1 GENERAL INITIALIZATION

We refer to the robot end-effector as $ee$, and its rest position as $r_{pos}$. The end-effector resting position is $(0.5\,\text{m}, 0\,\text{m}, 1.25\,\text{m})$ relative to the base[1]. Let $q_{arm}$ be the arm joint positions, $r_{arm}$ be the arm resting joint positions, and $\dot{q}_{arm}$ be the arm joint velocities. Similarly, for the torso define $q_{tor}$, $r_{tor}$, and $\dot{q}_{tor}$. Let $v_{base}$ be the base linear velocity in $\text{m\,s}^{-1}$, and $v_{base,x}, v_{base,y}$ its x and y components respectively. Let $\omega_{base}$ be the base angular velocity in $\text{rad\,s}^{-1}$.

We initialize the robot to $r_{pos}, r_{arm}, r_{tor}$ with $\dot{q}_{arm} = (0\ldots0), \dot{q}_{tor} = 0, v_{base} = 0, \omega_{base} = 0$. Then, $q_{arm}$ is perturbed by clipped Gaussian noise $\text{clip}(\mathcal{N}(0, 0.1), -0.2, 0.2)$, the base position is perturbed by $\text{clip}(\mathcal{N}(0, 0.1), -0.2, 0.2)$, and the base rotation is perturbed by $\text{clip}(\mathcal{N}(0, 0.25), -0.5, 0.5)$.

## A.2 SUBTASK DEFINITIONS

Let $d_b^a = \|a_{pos} - b_{pos}\|_2^2$, units in m. For example, $d_{ee}^r$ is the distance in m between the end-effector and its rest position. Next, let $j_k = \max_{1 \le i \le |q_k|} |q_{k,i} - r_{k,i}|$. For example, $j_{arm}$ is the maximum absolute difference in the arm joint positions and corresponding resting positions. Let $C_{[0:t]}$ be the cumulative robot collisions in N until step $t$. Finally, we define two commonly-used success conditions for the Fetch robot:

$$\mathbf{1}_{\text{grasped}(x)} = \mathbf{1}\left\{x \text{ is grasped (computed by simulator)}\right\}$$

$$\mathbf{1}_{\text{is\_static}} = \mathbf{1}\left\{\left(\max_{1 \le i \le |\dot{q}_{\text{arm}}|} \dot{q}_{arm,i} \le 0.2\right) \wedge v_{base,x} \le 0.05 \wedge v_{base,y} \le 0.05 \wedge \omega_{base} \le 0.05\right\}$$

**Pick**$[a, optional](x_{pose})$: Pick object $x$ (from articulation $a$, if provided).

- Initialization: Spawn robot facing $x$, within $2\,\text{m}$ of $x$, with noise, and without collisions.
- Success:

$$\mathbf{1}_{\text{grasped}(x)} \wedge d_{ee}^r \le 0.05 \wedge j_{arm} \le 0.6 \wedge \mathbf{1}_{\text{is\_static}} \wedge C_{[0:t]} \le 5000$$

- Failure: $C_{[0:t]} > 5000\,\text{N}$

**Place**$[a, optional](x_{pose}, g_{pos})$: Place object $x$ at goal $g$ (in articulation $a$, if provided).

- Initialization: Spawn with grasp pose sampled from $\text{Pick}(x_{pose})$ policy, robot facing $g$, within $2\,\text{m}$ of $g$, with noise, and without collisions.
- Success:

$$\neg\mathbf{1}_{\text{grasped}(x)} \wedge d_x^g \le 0.15 \wedge d_{ee}^r \le 0.05 \wedge j_{arm} \le 0.2 \wedge j_{tor} \le 0.01 \wedge \mathbf{1}_{\text{is\_static}} \wedge C_{[0:t]} \le 7500$$

- Failure: $C_{[0:t]} > 7500\,\text{N}$

**Open**$[a](a_{pos})$: Open articulation $a$ with handle at $a_{pos}$.

- Initialization: Spawn facing $a$. If $a$ is a fridge, spawn within $[0.933, -0.6] \times [1.833, 0.6]$ region in front of $a$, otherwise within $[0.3, -0.6] \times [1.5, 0.6]$. With noise, without collisions.
- Success: Let $a_q, a_{qmax}, a_{qmin}$ be the current, max, and min joint positions for the target articulation (drawer or fridge). Then, let $a_{ofrac} = \{0.75 \text{ if } a \text{ is a fridge else } 0.9\}$. We define

$$\mathbf{1}_{\text{open}(a)} = \mathbf{1}\left\{a_q \ge a_{ofrac} \cdot (a_{qmax} - a_{qmin}) + a_{qmin}\right\}$$

Hence, we have success condition

$$\mathbf{1}_{\text{open}(a)} \wedge d_{ee}^r \le 0.05 \wedge j_{arm} \le 0.2 \wedge j_{tor} \le 0.01 \wedge \mathbf{1}_{\text{is\_static}} \wedge C_{[0:t]} \le 10000$$

---

[1]The z-axis is 'up' in ManiSkill3.

- Failure[2]: $C_{[0:t]} > 10\,000\,\text{N}$

**Close**$[a](a_{pos})$**:** Close articulation $a$ with handle at $a_{pos}$.

- Initialization: Spawn facing $a$. If $a$ is a fridge, spawn within $[0.933, -0.6] \times [1.833, 0.6]$ region in front of $a$, otherwise within $[0.3, -0.6] \times [1.5, 0.6]$. With noise, without collisions.
- Success: Let $a_q, a_{qmax}, a_{qmin}$ be the current, max, and min joint positions for the target articulation (drawer or fridge). We define

$$\mathbf{1}_{\text{close}(a)} = \mathbf{1}\left\{a_q \le 0.01 \cdot (a_{qmax} - a_{qmin}) + a_{qmin}\right\}$$

Hence, we have success condition

$$\mathbf{1}_{\text{close}(a)} \wedge d_{ee}^r \le 0.05 \wedge j_{arm} \le 0.2 \wedge j_{tor} \le 0.01 \wedge \mathbf{1}_{\text{is\_static}} \wedge C_{[0:t]} \le 10000$$

- Failure[2]: $C_{[0:t]} > 10\,000\,\text{N}$

### A.3 RL SUBTASK EVALUATION CURVES

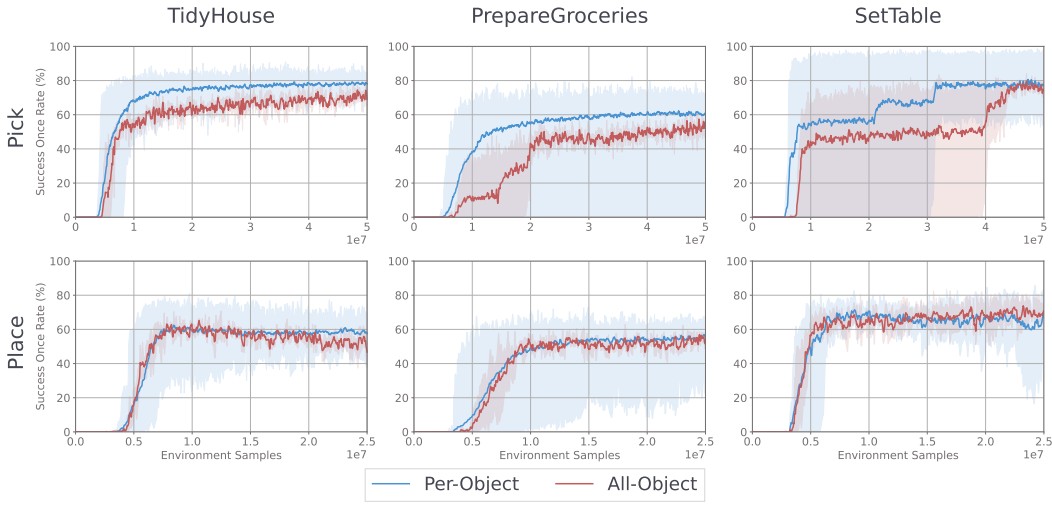

Figure 4: Per-object vs all-object RL success once rate (%) evaluation curves for Pick and Place policies across tasks. We run 3 seeds for each per-object policy and 3 seeds for the all-object policy. TidyHouse and PrepareGroceries involve 9 objects, while SetTable involves 2 objects. Since we group runs for different per-object policies into one curve, we use minimum and maximum for the shaded region. Best viewed zoomed.

During training, we evaluate our policies every 10000 steps on 189 episodes. The per vs all-object training curves in Fig. 4 demonstrate a similar trend as seen in Sec. 6.2.1: per-object policies show the most significant improvements when grasping many objects with different geometries (Tidy-House Pick and PrepareGroceries Pick) or when manipulating objects in tight constraints where object geometry is important (PrepareGroceries Place).

Per Tables 10-13, the performance limitations for Open and Close seen in Fig. 5 are caused primarily by the 10 000 N cumulative robot force limit we set, which is not used in the original implementation of the HAB (Szot et al., 2021).

See Appendix A.4.4 for more analysis on performance under lower collision thresholds.

---

[2]Originally, the HAB does not specify collision limits for Open or Close, but we add them to enforce safety.

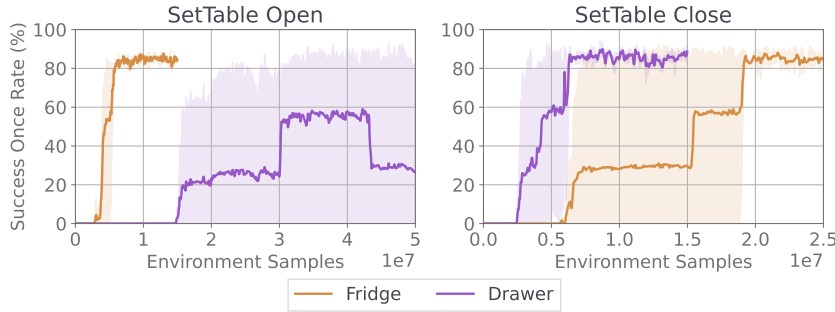

Figure 5: Open and Close training success once rate (%) curves for Drawer and Fridge. Since success once rate jumps very quickly once the policy learns to solve the task, we use minimum and maximum for the shaded region.

## A.4  ADDITIONAL EXPERIMENTS

### A.4.1  DATASET SIZE

To highlight the importance of generating scalable datasets, we train IL policies for TidyHouse/PrepareGroceroes/SetTable Pick/Place subtasks at 1, 10, 100, 500, and 1000 demonstrations per object. In Table 4, we run 1000 evaluation episodes per policy, and group results by demonstrations per object. We then report average success once rate and 95% CIs for each demonstrations per object value.

We find that 1000 demonstrations per object leads to the most performant policies. Furthermore there are large jumps in success rate as demonstrations per object increases from 10 to 100 to 500.

Table 4: Success once rate (**SoR**) with 95% CIs depending on demos per object (**Demos**).

| DEMOS | SoR |
|---|---|
| 1 | $0.00 \pm 0.00$ |
| 10 | $0.02 \pm 0.03$ |
| 100 | $0.27 \pm 0.19$ |
| 500 | $0.53 \pm 0.13$ |
| 1000 | $0.62 \pm 0.09$ |

### A.4.2  PERFORMANCE WITH TASK SIMPLIFICATIONS

In Sec. 6.1, we find that improving subtask success rate is the most effective way to increase long-horizon task success rate. In Fig. 6, we simplify the long-horizon tasks by (1) removing all collision requirements, and (2) marking Place subtasks successful if the object remains anywhere on the target receptacle surface. We find that progressive completion rate increases for both our RL and IL policies, but we achieve no higher than 20% overall task success on any task or split. Hence, the largest challenge in training subtask policies is low-level whole-body control. Meanwhile, collision requirements and subtask success conditions pose some difficulty, but not as much.

### A.4.3  SAC VS PPO FOR RL TRAINING

To justify our choices of RL algorithm for each policy, we compare SAC and PPO performance across tasks and subtasks. For Pick and Place, we compare all-object SAC and PPO subtask success once rate (%) on the train split. As described in Sec. 5.1, we train SAC with 25 million samples. Since PPO trains faster wall-time, we provide it 50 millions samples for a fair comparison.

Furthermore, for Open and Close, we compare per-object success once rate (%). We provide PPO with the same total samples as listed in Sec. 5.1, and we train SAC with 20 million samples per run.

For Pick and Place, despite training on double the total samples, PPO policies achieve notably lower performance compared to the SAC policies. We hypothesize this is because our large replay buffer can store a greater diversity of examples across objects, spawn locations, and obstructions, allowing

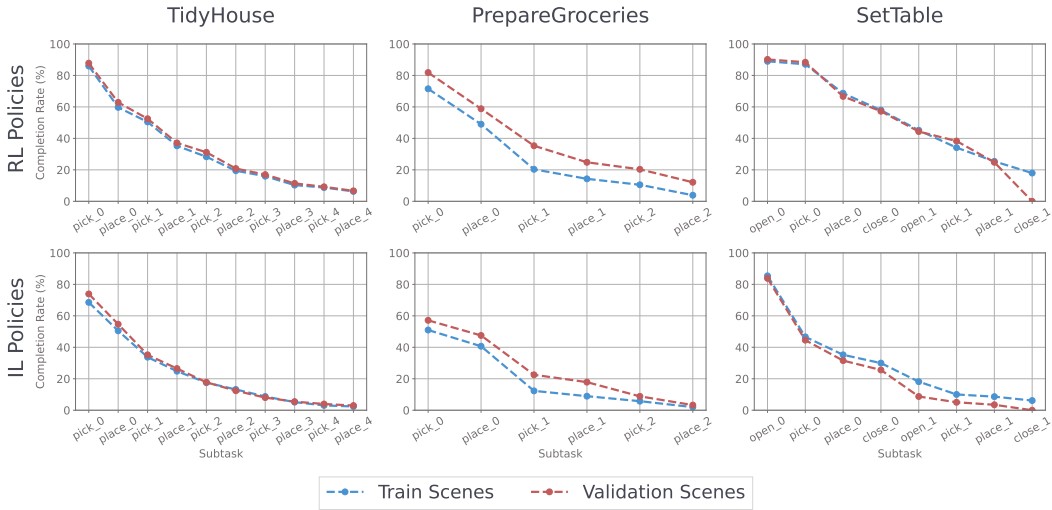

Figure 6: Progressive completion rate (%) on simplified long-horizon tasks with RL and IL policies. We remove all collision requirements, and allow placing on the full target receptacle surface. Best viewed zoomed.

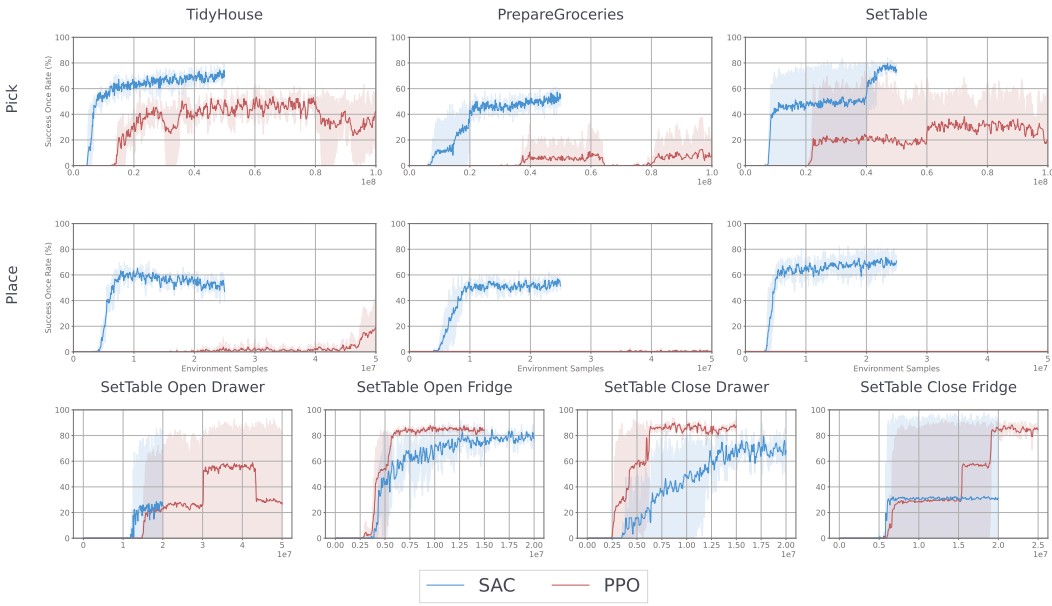

Figure 7: SAC vs PPO subtask success once rate (%) curves on the train split. Lines are averaged across 3 seeds; since success rate can jump rapidly, shaded regions represent min/max values. For Pick and Place, we compare all-object SAC and PPO policies, and for Open and Close, we compare per-object policies. Note that for PrepareGroceries and SetTable Place, lines are drawn but near-zero. Best viewed zoomed.

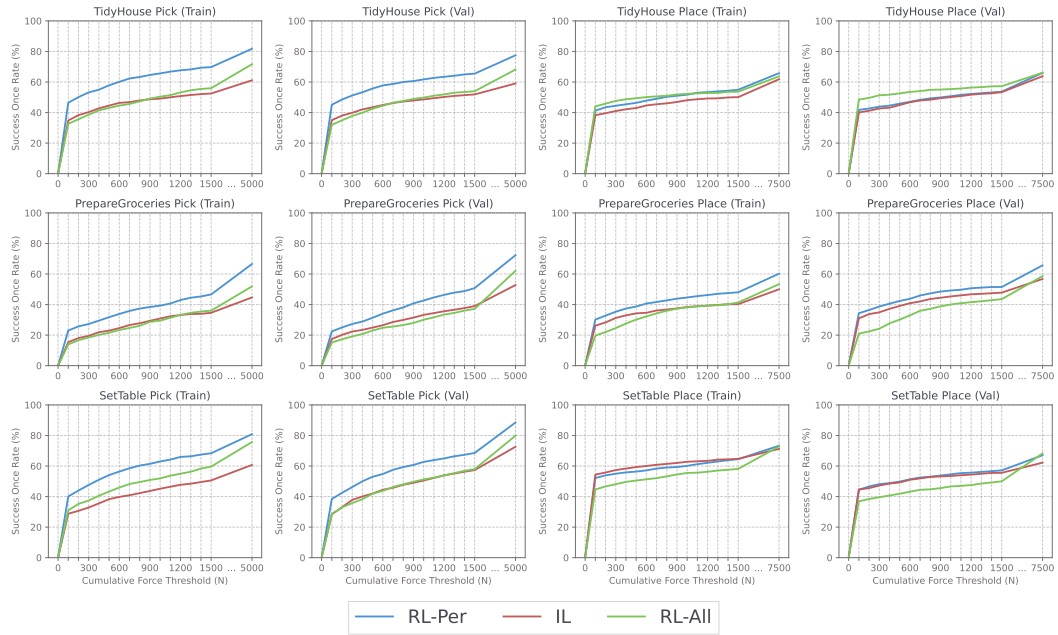

Figure 8: Success once rates (%) for RL-Per, RL-All, and IL policies in Pick and Place subtasks under varying cumulative collision thresholds. Best viewed zoomed.

SAC policies to better learn manipulation in diverse settings (Haarnoja et al., 2018). Hence, we use SAC for RL Pick and Place baselines.

Meanwhile, in Open Fridge and Close Drawer PPO policies perform better than SAC policies, in Close Fridge PPO policies perform marginally worse than SAC polices, and in Open Drawer PPO policies perform better only with many more samples. Since performance between PPO and SAC is generally comparable in Open and Close, we choose PPO for our baselines since it has faster wall-time training. For consistency, we use the same RL algorithm across all Open and Close variants.

### A.4.4 PERFORMANCE UNDER LOW COLLISION THRESHOLDS

To evaluate the safety of our policies in a real-world setting, we compare performance for RL-Per, RL-All, and IL policies on Pick and Place subtasks under low cumulative collision thresholds. Per industry safety standards, we use $1400\,\mathrm{N}$ for as the measure for safe execution (Mewes & Mauser, 2003).

As seen in Fig. 8, we observe a 5-20% drop in performance depending on subtask when using a cumulative collision threshold of $1400\,\mathrm{N}$. Additionally, the per-object RL policies notably outperform all-object RL policies under lower collision thresholds.

### A.4.5 PER VS ALL-OBJECT POLICY LONG-HORIZON PERFORMANCE

In addition to superior subtask success rates, as seen in Fig. 9, per-object RL policies outperform their all-object counterparts on full long horizon tasks in both train and validation splits.

### A.4.6 DIFFUSION POLICY BASELINES

To explore more complicated methods, we train diffusion policy (DP) baselines. We use a setup similar to the original DP paper, with a UNet backbone and a DDPM scheduler (Chi et al., 2023; 2024). For our visual encoders, we use a simpler 4-layer CNN rather than a ResNet. For consistency, we use the same architecture and hyperparmeters for all subtasks.

As seen in Table 5, our DP baselines surprisingly perform generally worse than our BC baselines. Performance is closer in Place subtasks, but worse in Pick, potentially due to the increased potential

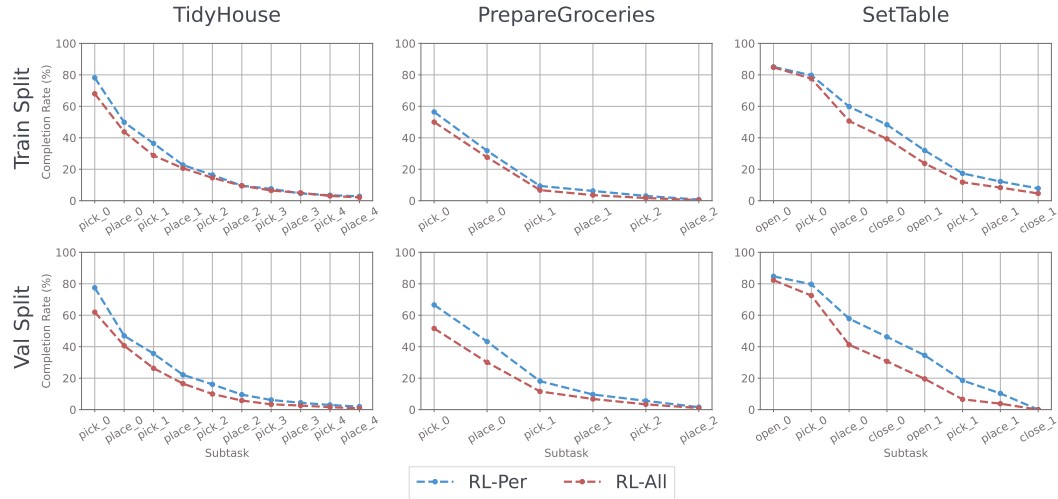

Figure 9: Success once rates (%) for RL-Per and RL-All policies on long-horizon tasks in train and validation split. Best viewed zoomed.

Table 5: Subtask success once rates for BC and DP baselines.

| TASK | SUBTASK | SPLIT | BC | DP |
|------|---------|-------|-----|-----|
| TidyHouse | Pick | Train | **61.11** | 28.37 |
| | | Val | **59.03** | 27.18 |
| | Place | Train | **61.81** | 58.63 |
| | | Val | **63.79** | 59.92 |
| Prepare Groceries | Pick | Train | **44.64** | 19.35 |
| | | Val | **52.78** | 19.74 |
| | Place | Train | **50.00** | 39.09 |
| | | Val | **56.75** | 50.40 |
| SetTable | Pick | Train | **60.71** | 23.71 |
| | | Val | **72.62** | 24.40 |
| | Place | Train | **71.23** | 64.19 |
| | | Val | **62.20** | 55.36 |
| | $Open_{Fr}$ | Train | **74.01** | 62.10 |
| | | Val | 53.67 | **63.79** |
| | $Open_{Dr}$ | Train | **79.86** | 16.17 |
| | | Val | **78.57** | 15.18 |
| | $Close_{Fr}$ | Train | **86.90** | 64.09 |
| | | Val | 0.00 | **0.10** |
| | $Close_{Dr}$ | Train | 88.39 | **89.29** |
| | | Val | **87.60** | 85.81 |

for collisions when picking objects. Interestingly, DP is the only baseline which achieves non-zero success on Open Fridge on the validation split, despite the fridge being against a wall (unseen in the train split).

Our results suggest that, while smaller backbones or limited tuning can solve simpler tasks like Push-T or those in the ManiSkill3 standard task suite, the ManiSkill-HAB tasks might require larger/different backbones (e.g. diffusion transformer), tuning hyperparemeters per subtask, or including newer methods like online finetuning (e.g. DPPO) (Dasari et al., 2024; Ren et al., 2024).

## A.5 Ray Tracing and Visual Fidelity

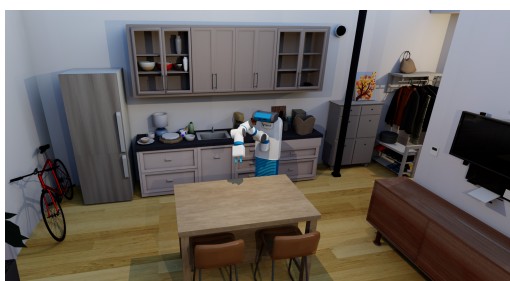 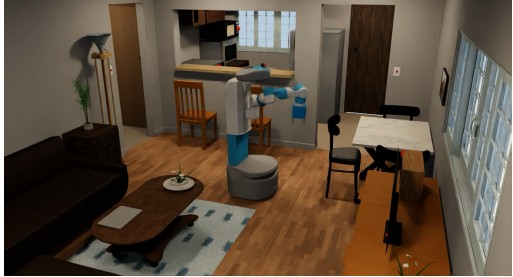

Figure 10: Left: ManiSkill-HAB with ray tracing on. Right: Behavior-1k with ray tracing on. Both images are live-rendered. The right image is taken from the Behavior-1k Google Colab demo notebook (Li et al., 2022).

For visual realism, we provide live-rendered ray-tracing with tuned lighting, which can be selected with only one line in the code. We compare rendering performance and quality with Behavior-1k, a platform known for its visual realism (Li et al., 2022).

To compare performance, we run an altered version of Behavior-1k's rendering benchmark. We use a single Nvidia RTX 4090, render 1 128x128 RGB-D image, and simulate dynamics with a simulation frequency of 120Hz and control frequency of 30Hz. Each evaluation run consists of 300 steps of random actions clipped to [-0.3, 0.3]. We report mean and 95% CIs over 10 evaluation runs.

While live-rendering with ray tracing, ManiSkill-HAB achieves $69.90 \pm 0.25$ samples per second (SPS) while using $6.26 \pm 0.00$ GB of GPU memory, while Behavior-1k is limited to $19.92 \pm 0.04$ SPS while using $7.62 \pm 0.04$ GB of GPU memory.

Hence, ManiSkill-HAB is 3.51x faster than Behavior-1k while using 17.85% less GPU memory, while also retaining similar ray-tracing render quality as seen in Fig. 10.

## A.6 Trajectory Categorization and Dataset Filtering

In this section, we provide definitions for our event labeling and trajectory categorization system. We additionally provide statistics on policy success and failure modes using our trajectory categorization system. Some example videos for Pick and Place failure modes are provided in the supplementary and project website.

### A.6.1 Dataset Filtering and Generation

We generate 1000 demonstrations per object/articulation for each subtask using per-object RL policies on the train split. We use our trajectory labeling system to filter demonstrations (full definitions in Appendix A.6.2). For Pick, we require "straightforward success" demonstrations, where the agent successfully picks the object without dropping it while remaining within the cumulative collision threshold. For Place, we require "placed in goal success" demonstrations, where the agent releases the object within 15cm of the goal, the object stays in the goal without rolling or falling out, and the agent remains within the cumulative collision threshold. For Open and Close, we require "open success" and "closed success" demonstrations, where the agent opens/closes the articulation without excessive collisions, and the articulation remains within the open/close state.

A.6.2 DEFINITIONS

For each success and failure mode definition, we provide a plain text description in addition to the boolean definitions. We heavily rely on notation used in Appendix A.1, in addition to those defined below.

Using the criteria defined below, for each trajectory $\tau_{\text{subtask}} = (s_0, a_0, \ldots, s_n, a_n)$, we create a chronologically ordered event list $E_{\text{subtask}} = (e_1, \ldots, e_k)$. Using $E_{\text{subtask}}$, we categorize $\tau_{\text{subtask}}$ into a success/failure mode.

Let $i_{\text{subtask,event}} = \{\text{index of } e_{\text{event}} \text{ in } E_{\text{subtask}} \text{ if } e_{\text{event}} \in E_{\text{subtask}} \text{ else } -1\}$. Also, let $F_{a,b,t}$ be the pairwise force between $a$ and $b$ in N at timestep $t$.

**Pick**$[a, optional](x_{pose})$: Pick object $x$ (from articulation $a$, if provided).

- Events: We define time-conditioned events at timestep $t$:

$$e_{\text{contact}} = |F_{ee,x,t-1}| = 0 \land |F_{ee,x,t}| \geq 0$$
$$e_{\text{grasped}} = \neg\mathbf{1}_{\text{grasped}(x),t-1} \land \mathbf{1}_{\text{grasped}(x),t}$$
$$e_{\text{dropped}} = \mathbf{1}_{\text{grasped}(x),t-1} \land \neg\mathbf{1}_{\text{grasped}(x),t}$$
$$e_{\text{success}} = \neg\mathbf{1}_{\text{success},t-1} \land \mathbf{1}_{\text{success},t}$$
$$e_{\text{excessive\_collisions}} = C_{[0:t-1]} \leq 5000 \land C_{[0:t]} > 5000$$

  For $t \in \{1, \ldots n\}$ (in increasing order), we evaluate each $e_{\text{event}}$ in the order shown above. If $e_{\text{event}} = 1$, we add it to $E_{\text{pick}}$.

- Success Modes: if $e_{\text{success}} \in E_{\text{pick}}$, then categorize using the following success modes:

  i Straightforward success: Agent successfully grasps $x$ and returns to rest without dropping or excessive collisions.
  $E_{\text{pick}} = (e_{\text{contact}}, e_{\text{grasped}}, e_{\text{success}})$

  ii Winding success: Agent (eventually) successfully grasps $x$ (but drops $x$ along the way) and returns to rest without excessive collisions.
  $E_{\text{pick}} = (e_{\text{contact}}, e_{\text{grasped}}, \ldots, e_{\text{success}}) \land |E_{\text{pick}}| > 3 \land e_{\text{excessive\_collisions}} \notin E_{\text{pick}}$

  iii Success then drop: Agent successfully picks $x$ and returns to rest without excessive collisions, but irrecoverably drops $x$ after.
  $e_{\text{dropped}} \in E_{\text{pick}} \land i_{\text{subtask,dropped}} > i_{\text{subtask,grasped}} \land e_{\text{excessive\_collisions}} \notin E_{\text{pick}}$

  iv Success then excessive collisions: Agent picks $x$ and returns to rest, but exceeds collision threshold afterwards.
  $e_{\text{excessive\_collisions}} \in E_{\text{pick}}$

- Failure Modes: if $e_{\text{success}} \notin E_{\text{pick}}$, then categorize using the following failure modes:

  v Excessive collision failure: Agent exceeds collision threshold.
  $e_{\text{excessive\_collisions}} \in E_{\text{pick}}$

  vi Mobility failure: Agent cannot reach $x$.
  $E_{\text{pick}} = ()$

  vii Can't grasp failure: Agent reaches $x$, but cannot grasp it.
  $E_{\text{pick}} = (e_{\text{contact}})$

  viii Drop failure: Agent grasps $x$, but drops it before returning to rest.
  $e_{\text{dropped}} \in E_{\text{pick}} \land i_{\text{subtask,dropped}} > i_{\text{subtask,grasped}} \land e_{\text{excessive\_collisions}} \notin E_{\text{pick}}$

  ix Too slow failure: Agent (eventually) grasps $x$, but the episode truncates before it can reach success.
  $e_{\text{grasped}} \in E_{\text{pick}} \land i_{\text{subtask,grasped}} > i_{\text{subtask,dropped}} \land e_{\text{excessive\_collisions}} \notin E_{\text{pick}}$

**Place**$[a, optional](x_{pose}, g_{pos})$: Place object $x$ at goal $g$ (in articulation $a$, if provided).

- Events: We define time-conditioned events at timestep $t$:

$$e_{\text{grasped}} = \neg\mathbf{1}_{\text{grasped}(x),t-1} \wedge \mathbf{1}_{\text{grasped}(x),t}$$

$$e_{\text{obj\_at\_goal}} = d^g_{x,t-1} > 0.15 \wedge d^g_{x,t} \leq 0.15$$

$$e_{\text{released\_at\_goal}} = d^g_x \leq 0.15 \wedge \mathbf{1}_{\text{grasped}(x),t-1} \wedge \neg\mathbf{1}_{\text{grasped}(x),t}$$

$$e_{\text{released\_outside\_goal}} = d^g_x > 0.15 \wedge \mathbf{1}_{\text{grasped}(x),t-1} \wedge \neg\mathbf{1}_{\text{grasped}(x),t}$$

$$e_{\text{obj\_left\_goal}} = d^g_{x,t-1} \leq 0.15 \wedge d^g_{x,t} > 0.15$$

$$e_{\text{success}} = \neg\mathbf{1}_{\text{success},t-1} \wedge \mathbf{1}_{\text{success},t}$$

$$e_{\text{excessive\_collisions}} = C_{[0:t-1]} \leq 7500 \wedge C_{[0:t]} > 7500$$

For $t \in \{1, \ldots n\}$ (in increasing order), we evaluate each $e_{\text{event}}$ in the order shown above. If $e_{\text{event}} = 1$, we add it to $E_{\text{place}}$.

- Success Modes: if $e_{\text{success}} \in E_{\text{place}}$, then categorize using the following success modes:

  i Place in goal success: Agent releases and successfully places $x$ to within 15cm of $g_{pos}$, then returns to rest.
  $|E_{\text{place}}| \leq 4 \wedge (e_{\text{released\_at\_goal}} \in E_{\text{place}} \vee d^g_{x,0} \leq 0.15) \wedge i_{\text{place,obj\_left\_goal}} \leq i_{\text{place,obj\_at\_goal}} \wedge e_{\text{excessive\_collisions}} \notin E_{\text{place}}$

  ii Dropped to goal success: Agent releases $x$ beyond 15cm of $g_{pos}$, $x$ drops into the region within 15cm of $g_{pos}$, and the agent returns to rest.
  $|E_{\text{place}}| \leq 4 \wedge (e_{\text{released\_outside\_goal}} \in E_{\text{place}} \vee d^g_{x,0} > 0.15) \wedge i_{\text{place,obj\_left\_goal}} \leq i_{\text{place,obj\_at\_goal}} \wedge e_{\text{excessive\_collisions}} \notin E_{\text{place}}$

  iii Dubious success: $x$ is manipulated to within 15cm of $g_{pos}$, and the robot returns to rest, but $x$ leaves $g$ before truncation.
  $i_{\text{place,obj\_at\_goal}} < i_{\text{place,obj\_left\_goal}} \wedge e_{\text{excessive\_collisions}} \notin E_{\text{place}}$

  iv Winding success: $x$ leaves the goal at least once, but the agent (eventually) successfully places/drops $x$ to within 15cm of $g_{pos}$, where it remains as the agent returns to rest.
  $|E_{\text{place}}| > 4 \wedge i_{\text{place,obj\_at\_goal}} > i_{\text{place,obj\_left\_goal}} \wedge e_{\text{excessive\_collisions}} \notin E_{\text{place}}$

  v Success then excessive collisions: The agent successfully places/drops $x$ to within 15cm of $g_{pos}$ and returns to rest, but exceeds collision threshold after.
  $e_{\text{excessive\_collisions}} \in E_{\text{place}}$

- Failure Modes: if $e_{\text{success}} \notin E_{\text{place}}$, then categorize using the following failure modes:

  vi Excessive collision failure: Agent exceeds collision threshold.
  $e_{\text{excessive\_collisions}} \in E_{\text{place}}$

  vii Didn't grasp failure: Agent fails to grasp $x$ at initialization.
  $E_{\text{place}} = () \wedge e_{\text{excessive\_collisions}} \notin E_{\text{place}}$

  viii Didn't reach goal failure: Agent grasps $x$, but cannot manipulate $x$ to within 15cm of $g_{pos}$.
  $|E_{\text{place}}| > 0 \wedge e_{\text{obj\_at\_goal}} \notin E_{\text{place}} \wedge e_{\text{excessive\_collisions}} \notin E_{\text{place}}$

  ix Place in goal failure: Agent places $x$ to within 15cm of $g_{pos}$, but $x$ leaves this region (i.e., rolls or falls out) before the agent returns to rest.
  First, we define
  $\mathbf{1}_{\text{placed\_is\_latest\_sequence}} = (|E_{\text{place}}| \leq 2 \wedge d^g_{x,0} \leq 0.15) \vee (i_{\text{place,released\_at\_goal}} > i_{\text{place,released\_outside\_goal}} \wedge i_{\text{place,released\_at\_goal}} > i_{\text{place,grasped}})$
  Hence, we have failure mode definition
  $e_{\text{obj\_at\_goal}} \in E_{\text{place}} \wedge \mathbf{1}_{\text{placed\_is\_latest\_sequence}} \wedge i_{\text{place,obj\_left\_goal}} > i_{\text{place,obj\_at\_goal}} \wedge e_{\text{excessive\_collisions}} \notin E_{\text{place}}$

  x Dropped to goal failure: Agent drops $x$ beyond 15cm away from $g_{pos}$, and $x$ drops into the region within 15cm of $g_{pos}$, but leaves this region (i.e., rolls or falls out) before the agent returns to rest. First, we define
  $\mathbf{1}_{\text{dropped\_is\_latest\_sequence}} = (|E_{\text{place}}| \leq 2 \wedge d^g_{x,0} > 0.15) \vee (i_{\text{place,released\_outside\_goal}} > i_{\text{place,released\_at\_goal}} \wedge i_{\text{place,released\_outside\_goal}} > i_{\text{place,grasped}})$
  Hence, we have failure mode definition
  $e_{\text{obj\_at\_goal}} \in E_{\text{place}} \wedge \mathbf{1}_{\text{dropped\_is\_latest\_sequence}} \wedge i_{\text{place,obj\_left\_goal}} > i_{\text{place,obj\_at\_goal}} \wedge e_{\text{excessive\_collisions}} \notin E_{\text{place}}$

    xi Won't let go failure: The agent is able to manipulate $x$ to within 15cm of $g_{pos}$, but does not release $x$.

$e_{\text{obj\_at\_goal}} \in E_{\text{place}} \wedge i_{\text{place,grasped}} > i_{\text{place,released\_at\_goal}} \wedge i_{\text{place,grasped}} > i_{\text{place,released\_outside\_goal}} \wedge e_{\text{excessive\_collisions}} \notin E_{\text{place}}$

    xii Too slow failure: The agent is able to manipulate $x$ to within 15cm of the goal, releases $x$, but is unable to return to rest before truncation.
The condition is no other failure mode is applicable. This also implies

$i_{\text{place,obj\_at\_goal}} > i_{\text{place,obj\_left\_goal}} \wedge e_{\text{excessive\_collisions}} \notin E_{\text{place}}$

**Open**$[a](a_{pos})$**:** Open articulation $a$ with handle at $a_{pos}$.

- Events: First, define indicator

$$\mathbf{1}_{\text{slightly\_opened}(a),t} = \mathbf{1}\left\{ a_{q,t} \geq 0.1 \cdot (a_{qmax} - a_{qmin}) + a_{qmin} \right\}$$

Now, we define time-conditioned events at timestep $t$:

$$e_{\text{contact}} = |F_{ee,a,t-1}| = 0 \wedge |F_{ee,a,t}| \geq 0$$
$$e_{\text{opened}} = \neg\mathbf{1}_{\text{open}(a),t-1} \wedge \mathbf{1}_{\text{open}(a),t}$$
$$e_{\text{slightly\_opened}} = \neg\mathbf{1}_{\text{slightly\_opened}(a),t-1} \wedge \mathbf{1}_{\text{slightly\_opened}(a),t}$$
$$e_{\text{closed}} = \mathbf{1}_{\text{open}(a),t-1} \wedge \neg\mathbf{1}_{\text{open}(a),t}$$
$$e_{\text{success}} = \neg\mathbf{1}_{\text{success},t-1} \wedge \mathbf{1}_{\text{success},t}$$
$$e_{\text{excessive\_collisions}} = C_{[0:t-1]} \leq 10000 \wedge C_{[0:t]} > 10000$$

For $t \in \{1, \dots n\}$ (in increasing order), we evaluate each $e_{\text{event}}$ in the order shown above. If $e_{\text{event}} = 1$, we add it to $E_{\text{open}}$.

- Success Modes: if $e_{\text{success}} \in E_{\text{open}}$, then categorize using the following success modes:

    i Open success: Agent successfully opens $a$ and returns to rest without excessive collisions.
$e_{\text{excessive\_collisions}} \notin E_{\text{open}} \wedge i_{\text{open,opened}} > i_{\text{open,closed}}$

    ii Dubious success: Agent successfully opens $a$ and returns to rest without excessive collisions, but accidentally closes $a$ after
$e_{\text{excessive\_collisions}} \notin E_{\text{open}} \wedge i_{\text{open,opened}} < i_{\text{open,closed}}$

    iii Success then excessive collisions: Agent successfully opens $a$ and returns to rest, but exceeds collision threshold after.
$e_{\text{excessive\_collisions}} \notin E_{\text{open}}$

- Failure Modes: if $e_{\text{success}} \notin E_{\text{open}}$, then categorize using the following failure modes:

    iv Excessive collision failure: Agent exceeds collision threshold.
$e_{\text{excessive\_collisions}} \in E_{\text{open}}$

    v Can't reach articulation failure: Agent cannot reach $a$.
$e_{\text{contact}} \notin E_{\text{open}} \wedge e_{\text{excessive\_collisions}} \notin E_{\text{open}}$

    vi Closed after open failure: Agent opens $a$, but closes it before returning to rest.
$e_{\text{closed}} \in E_{\text{open}} \wedge i_{\text{open,closed}} > i_{\text{open,opened}} \wedge i_{\text{open,closed}} > i_{\text{open,slightly\_opened}} \wedge e_{\text{excessive\_collisions}} \notin E_{\text{open}}$

    vii Slightly opened failure: Agent at least slightly opens $a$, but cannot fully open it.
Previous failure modes are not applicable, and $i_{\text{open,slightly\_opened}} > i_{\text{open,opened}} \wedge i_{\text{open,slightly\_opened}} > i_{\text{open,closed}} \wedge e_{\text{excessive\_collisions}} \notin E_{\text{open}}$

    viii Too slow failure: Agent is able to open $a$, but cannot return to rest in time.
Previous failure modes are not applicable, and $e_{\text{opened}} \in E_{\text{open}}$

    ix Can't open failure: Agent reaches $a$, but cannot open it.
The condition is no other failure mode is applicable. This also implies $e_{\text{contact}} \in E_{\text{open}} \wedge e_{\text{opened}} \notin E_{\text{open}}$

**Close**$[a](a_{pos})$**:** Close articulation $a$ with handle at $a_{pos}$.

- Events: First, define indicator

$$\mathbf{1}_{\text{slightly\_closed}(a),t} = \mathbf{1}\left\{a_{q,t} < a_{q,0} - 0.05 \cdot (a_{qmax} - q_{qmin})\right\}$$

  Now, we define time-conditioned events at timestep $t$:

$$e_{\text{contact}} = |F_{ee,a,t-1}| = 0 \wedge |F_{ee,a,t}| \geq 0$$
$$e_{\text{closed}} = \neg\mathbf{1}_{\text{closed}(a),t-1} \wedge \mathbf{1}_{\text{closed}(a),t}$$
$$e_{\text{slightly\_closed}} = \neg\mathbf{1}_{\text{slightly\_closed}(a),t-1} \wedge \mathbf{1}_{\text{slightly\_closed}(a),t}$$
$$e_{\text{open}} = \mathbf{1}_{\text{closed}(a),t-1} \wedge \neg\mathbf{1}_{\text{closed}(a),t}$$
$$e_{\text{success}} = \neg\mathbf{1}_{\text{success},t-1} \wedge \mathbf{1}_{\text{success},t}$$
$$e_{\text{excessive\_collisions}} = C_{[0:t-1]} \leq 10000 \wedge C_{[0:t]} > 10000$$

  For $t \in \{1, \ldots n\}$ (in increasing order), we evaluate each $e_{\text{event}}$ in the order shown above. If $e_{\text{event}} = 1$, we add it to $E_{\text{close}}$.

- Success Modes: if $e_{\text{success}} \in E_{\text{close}}$, then categorize using the following success modes:

  i Close success: Agent successfully closes $a$ and returns to rest without excessive collisions.
  $$e_{\text{excessive\_collisions}} \notin E_{\text{close}} \wedge i_{\text{close,closed}} > i_{\text{close,opened}}$$

  ii Dubious success: Agent successfully closes $a$ and returns to rest without excessive collisions, but accidentally opens $a$ after.
  $$e_{\text{excessive\_collisions}} \notin E_{\text{close}} \wedge i_{\text{close,closed}} < i_{\text{close,opened}}$$

  iii Success then excessive collisions: Agent successfully closes $a$ and returns to rest, but exceeds collision threshold after.
  $$e_{\text{excessive\_collisions}} \notin E_{\text{close}}$$

- Failure Modes: if $e_{\text{success}} \notin E_{\text{close}}$, then categorize using the following failure modes:

  iv Excessive collision failure: Agent exceeds collision threshold.
  $$e_{\text{excessive\_collisions}} \in E_{\text{close}}$$

  v Can't reach articulation failure: Agent cannot reach $a$.
  $$e_{\text{contact}} \notin E_{\text{close}} \wedge e_{\text{excessive\_collisions}} \notin E_{\text{close}}$$

  vi Opened after closed failure: Agent closes $a$, but opens it before returning to rest.
  $$e_{\text{closed}} \in E_{\text{close}} \wedge i_{\text{close,opened}} > i_{\text{close,closed}} \wedge i_{\text{close,opened}} > i_{\text{close,slightly\_closed}} \wedge e_{\text{excessive\_collisions}} \notin E_{\text{close}}$$

  vii Slightly closed failure: Agent at least slightly closes $a$, but cannot fully close it.
  Previous failure modes are not applicable, and $i_{\text{close,slightly\_closed}} > i_{\text{close,closed}} \wedge i_{\text{close,slightly\_closed}} > i_{\text{close,opened}} \wedge e_{\text{excessive\_collisions}} \notin E_{\text{close}}$

  viii Too slow failure: Agent is able to close $a$, but cannot return to rest in time.
  Previous failure modes are not applicable, and $e_{\text{closed}} \in E_{\text{close}}$

  ix Can't close failure: Agent reaches $a$, but cannot close it.
  The condition is no other failure mode is applicable. This also implies $e_{\text{contact}} \in E_{\text{close}} \wedge e_{\text{closed}} \notin E_{\text{close}}$

### A.6.3 TRAJECTORY CATEGORIZATION STATISTICS

In Tables 6-13, we categorize trajectories with our automated event labeling method. We run 1000 episodes for every task/subtask/target combination using all relevant policies (RL-Per, RL-All, IL) for each object/articulation. We provide success once rate (**SoR**), success at end rate (**SaeR**), failure rate (**FR**), and proportions for each success and failure mode as percentages (labeled with roman numerals corresponding to those use in Appendix A.6.2).

Table 6: Train split Pick policy trajectory labeling on 1000 episodes per target object. All numbers are percentages. Best viewed zoomed.

| TASK | OBJ | TYPE | SoR | SaeR | (i) | (ii) | (iii) | (iv) | FR | (v) | (vi) | (vii) | (viii) | (ix) |
|---|---|---|---|---|---|---|---|---|---|---|---|---|---|---|
| TH | 002 | RL-Per | 82.34 | 72.52 | 70.63 | 1.88 | 0.00 | 9.82 | 17.66 | 13.79 | 3.37 | 0.40 | 0.10 | 0.00 |
| | | RL-All | 78.97 | 60.81 | 58.43 | 2.38 | 0.00 | 18.15 | 21.03 | 13.69 | 5.65 | 0.89 | 0.50 | 0.30 |
| | | IL | 72.62 | 70.63 | 67.76 | 2.88 | 0.20 | 1.79 | 27.38 | 21.03 | 0.69 | 1.79 | 1.29 | 2.58 |
| | 003 | RL-Per | 71.63 | 62.80 | 60.91 | 1.88 | 0.10 | 8.73 | 28.37 | 17.26 | 7.64 | 2.48 | 0.89 | 0.10 |
| | | RL-All | 29.46 | 23.41 | 23.02 | 0.40 | 0.00 | 6.05 | 70.54 | 34.52 | 6.25 | 28.17 | 1.19 | 0.40 |
| | | IL | 17.96 | 17.16 | 16.87 | 0.30 | 0.00 | 0.79 | 82.04 | 49.01 | 0.20 | 31.25 | 0.89 | 0.69 |
| | 004 | RL-Per | 78.47 | 68.15 | 65.48 | 2.68 | 0.00 | 10.32 | 21.53 | 13.29 | 6.25 | 1.19 | 0.60 | 0.20 |
| | | RL-All | 74.60 | 57.64 | 55.16 | 2.48 | 0.00 | 16.96 | 25.40 | 16.27 | 6.25 | 1.88 | 0.40 | 0.60 |
| | | IL | 60.81 | 57.44 | 54.27 | 3.17 | 0.00 | 3.37 | 39.19 | 27.18 | 0.50 | 6.05 | 1.59 | 3.87 |
| | 005 | RL-Per | 81.05 | 78.27 | 75.10 | 3.17 | 0.00 | 2.78 | 18.95 | 15.67 | 2.58 | 0.40 | 0.10 | 0.20 |
| | | RL-All | 76.69 | 62.30 | 59.13 | 3.17 | 0.20 | 14.19 | 23.31 | 17.06 | 5.16 | 0.30 | 0.40 | 0.40 |
| | | IL | 74.01 | 70.73 | 66.37 | 4.37 | 0.10 | 3.17 | 25.99 | 21.13 | 0.40 | 0.79 | 0.89 | 2.78 |
| | 007 | RL-Per | 83.23 | 80.36 | 78.57 | 1.79 | 0.20 | 2.68 | 16.77 | 10.52 | 5.95 | 0.20 | 0.10 | 0.00 |
| | | RL-All | 77.98 | 70.44 | 69.44 | 0.99 | 0.00 | 7.54 | 22.02 | 15.97 | 5.56 | 0.10 | 0.10 | 0.30 |
| | | IL | 76.59 | 75.20 | 72.42 | 2.78 | 0.10 | 1.29 | 23.41 | 18.95 | 1.59 | 1.69 | 0.20 | 0.99 |
| | 008 | RL-Per | 75.99 | 72.22 | 63.19 | 9.03 | 0.50 | 3.27 | 24.01 | 17.36 | 3.08 | 2.18 | 0.79 | 0.60 |
| | | RL-All | 73.91 | 62.80 | 55.95 | 6.85 | 0.60 | 10.52 | 26.09 | 18.25 | 5.36 | 1.69 | 0.69 | 0.10 |
| | | IL | 65.18 | 62.50 | 54.07 | 8.43 | 0.50 | 2.18 | 34.82 | 26.39 | 0.40 | 3.77 | 2.18 | 2.08 |
| | 009 | RL-Per | 81.25 | 76.69 | 71.63 | 5.06 | 0.40 | 4.17 | 18.75 | 12.40 | 5.46 | 0.50 | 0.30 | 0.10 |
| | | RL-All | 71.43 | 60.42 | 50.10 | 10.32 | 0.40 | 10.62 | 28.57 | 20.54 | 4.86 | 2.28 | 0.60 | 0.30 |
| | | IL | 67.16 | 64.29 | 55.85 | 8.43 | 0.79 | 2.08 | 32.84 | 25.00 | 0.79 | 3.08 | 2.38 | 1.59 |
| | 010 | RL-Per | 81.65 | 52.78 | 48.61 | 4.17 | 0.10 | 28.77 | 18.35 | 12.40 | 4.86 | 0.79 | 0.20 | 0.10 |
| | | RL-All | 75.50 | 61.90 | 55.16 | 6.75 | 0.00 | 13.59 | 24.50 | 18.95 | 4.27 | 0.99 | 0.20 | 0.10 |
| | | IL | 58.33 | 54.17 | 48.02 | 6.15 | 0.50 | 3.67 | 41.67 | 26.98 | 0.79 | 8.43 | 1.19 | 4.27 |
| | 024 | RL-Per | 82.74 | 78.47 | 68.35 | 10.12 | 0.20 | 4.07 | 17.26 | 12.90 | 3.67 | 0.30 | 0.20 | 0.20 |
| | | RL-All | 73.91 | 65.08 | 56.15 | 8.93 | 0.00 | 8.83 | 26.09 | 19.05 | 6.15 | 0.69 | 0.10 | 0.10 |
| | | IL | 62.20 | 58.63 | 50.40 | 8.23 | 0.69 | 2.88 | 37.80 | 25.20 | 0.20 | 7.74 | 2.18 | 2.48 |
| | all | RL-Per | 81.75 | 73.41 | 67.26 | 6.15 | 0.20 | 8.13 | 18.25 | 12.40 | 4.17 | 1.19 | 0.30 | 0.20 |
| | | RL-All | 71.63 | 59.13 | 54.17 | 4.96 | 0.30 | 12.20 | 28.37 | 19.74 | 4.96 | 3.17 | 0.30 | 0.20 |
| | | IL | 61.11 | 59.42 | 54.56 | 4.86 | 0.20 | 1.49 | 38.89 | 26.39 | 0.69 | 7.64 | 0.89 | 3.27 |
| PG | 002 | RL-Per | 69.05 | 55.16 | 50.89 | 4.27 | 0.40 | 13.49 | 30.95 | 14.58 | 16.17 | 0.10 | 0.00 | 0.10 |
| | | RL-All | 62.70 | 49.40 | 38.10 | 11.31 | 1.49 | 11.81 | 37.30 | 24.60 | 11.31 | 0.89 | 0.50 | 0.00 |
| | | IL | 63.10 | 59.82 | 55.16 | 4.66 | 0.20 | 3.08 | 36.90 | 31.55 | 2.38 | 1.09 | 0.99 | 0.89 |
| | 003 | RL-Per | 51.98 | 43.15 | 40.67 | 2.48 | 2.08 | 6.75 | 48.02 | 25.10 | 12.30 | 8.63 | 1.79 | 0.20 |
| | | RL-All | 11.51 | 8.13 | 7.64 | 0.50 | 0.60 | 2.78 | 88.49 | 60.62 | 11.21 | 16.17 | 0.20 | 0.30 |
| | | IL | 16.27 | 14.38 | 13.79 | 0.60 | 1.29 | 0.60 | 83.73 | 67.06 | 2.28 | 12.10 | 1.98 | 0.30 |
| | 004 | RL-Per | 64.48 | 52.88 | 50.20 | 2.68 | 0.20 | 11.41 | 35.52 | 19.44 | 13.99 | 0.69 | 0.10 | 1.29 |
| | | RL-All | 59.82 | 46.03 | 37.10 | 8.93 | 1.39 | 12.40 | 40.18 | 26.98 | 11.41 | 0.99 | 0.40 | 0.40 |
| | | IL | 51.09 | 47.32 | 44.84 | 2.48 | 0.50 | 3.27 | 48.91 | 39.98 | 2.38 | 2.88 | 1.98 | 1.69 |
| | 005 | RL-Per | 61.21 | 48.51 | 44.74 | 3.77 | 0.30 | 12.40 | 38.79 | 25.79 | 11.51 | 0.10 | 0.20 | 1.19 |
| | | RL-All | 63.29 | 51.29 | 39.58 | 11.71 | 0.69 | 11.31 | 36.71 | 25.89 | 10.42 | 0.10 | 0.30 | 0.00 |
| | | IL | 61.01 | 56.25 | 54.07 | 2.18 | 0.20 | 4.56 | 38.99 | 34.33 | 2.28 | 0.50 | 0.89 | 0.99 |
| | 007 | RL-Per | 66.57 | 52.58 | 47.02 | 5.56 | 0.10 | 13.89 | 33.43 | 18.55 | 13.29 | 1.19 | 0.30 | 0.10 |
| | | RL-All | 64.38 | 45.63 | 34.42 | 11.21 | 0.89 | 17.86 | 35.62 | 23.71 | 10.81 | 0.99 | 0.10 | 0.00 |
| | | IL | 61.61 | 59.82 | 57.04 | 2.78 | 0.00 | 1.79 | 38.39 | 32.14 | 3.08 | 1.59 | 0.89 | 0.69 |
| | 008 | RL-Per | 62.90 | 45.73 | 37.00 | 8.73 | 0.89 | 16.27 | 37.10 | 21.33 | 14.38 | 0.79 | 0.50 | 0.10 |
| | | RL-All | 45.93 | 34.03 | 21.63 | 12.40 | 2.58 | 9.33 | 54.07 | 36.81 | 12.40 | 3.17 | 1.49 | 0.20 |
| | | IL | 40.38 | 35.71 | 32.24 | 3.47 | 1.19 | 3.47 | 59.62 | 49.80 | 2.78 | 3.97 | 2.58 | 0.50 |
| | 009 | RL-Per | 63.59 | 46.73 | 41.67 | 5.06 | 1.09 | 15.77 | 36.41 | 18.75 | 16.47 | 0.30 | 0.50 | 0.40 |
| | | RL-All | 49.01 | 38.49 | 26.19 | 12.30 | 1.69 | 8.83 | 50.99 | 32.14 | 13.00 | 3.67 | 1.69 | 0.50 |
| | | IL | 43.25 | 39.78 | 36.01 | 3.77 | 1.19 | 2.28 | 56.75 | 45.73 | 3.77 | 4.56 | 2.18 | 0.50 |
| | 010 | RL-Per | 65.18 | 51.49 | 44.64 | 6.85 | 0.30 | 13.39 | 34.82 | 20.93 | 13.29 | 0.20 | 0.30 | 0.10 |
| | | RL-All | 54.76 | 42.06 | 27.58 | 14.48 | 1.49 | 11.21 | 45.24 | 30.26 | 12.10 | 2.18 | 0.30 | 0.40 |
| | | IL | 50.99 | 48.61 | 41.77 | 6.85 | 0.10 | 2.28 | 49.01 | 41.27 | 2.68 | 2.38 | 1.69 | 0.99 |
| | 024 | RL-Per | 74.60 | 56.45 | 39.58 | 16.87 | 0.30 | 17.86 | 25.40 | 15.87 | 8.73 | 0.30 | 0.20 | 0.30 |
| | | RL-All | 51.09 | 35.02 | 22.92 | 12.10 | 0.79 | 15.28 | 48.91 | 37.00 | 10.12 | 1.09 | 0.60 | 0.10 |
| | | IL | 20.73 | 18.25 | 14.68 | 3.57 | 0.60 | 1.88 | 79.27 | 66.37 | 1.88 | 7.54 | 1.49 | 1.98 |
| | all | RL-Per | 66.57 | 52.78 | 46.63 | 6.15 | 0.40 | 13.39 | 33.43 | 19.54 | 11.81 | 1.09 | 0.60 | 0.40 |
| | | RL-All | 51.88 | 37.70 | 28.17 | 9.52 | 1.79 | 12.40 | 48.12 | 33.93 | 10.12 | 2.48 | 1.29 | 0.30 |
| | | IL | 44.64 | 42.36 | 39.38 | 2.98 | 0.40 | 1.88 | 55.36 | 47.22 | 1.49 | 4.27 | 1.49 | 0.89 |
| ST | 013 | RL-Per | 65.87 | 54.07 | 47.82 | 6.25 | 0.30 | 11.51 | 34.13 | 11.11 | 21.73 | 0.00 | 0.40 | 0.89 |
| | | RL-All | 59.03 | 35.71 | 33.04 | 2.68 | 0.10 | 23.21 | 40.97 | 16.57 | 24.01 | 0.10 | 0.00 | 0.30 |
| | | IL | 53.67 | 39.48 | 35.62 | 3.87 | 0.89 | 13.29 | 46.33 | 43.45 | 1.39 | 0.69 | 0.50 | 0.30 |
| | 024 | RL-Per | 94.35 | 85.81 | 75.30 | 10.52 | 0.20 | 8.33 | 5.65 | 4.86 | 0.50 | 0.10 | 0.20 | 0.00 |
| | | RL-All | 93.95 | 79.37 | 64.68 | 14.68 | 0.20 | 14.38 | 6.05 | 5.06 | 0.30 | 0.00 | 0.50 | 0.20 |
| | | IL | 65.58 | 58.63 | 51.88 | 6.75 | 0.00 | 6.75 | 34.42 | 24.21 | 1.29 | 5.26 | 2.68 | 0.99 |
| | all | RL-Per | 80.85 | 69.64 | 60.62 | 9.03 | 0.20 | 11.01 | 19.15 | 8.04 | 10.12 | 0.00 | 0.30 | 0.69 |
| | | RL-All | 75.69 | 56.55 | 47.52 | 9.03 | 0.10 | 19.05 | 24.31 | 11.01 | 12.40 | 0.30 | 0.50 | 0.10 |
| | | IL | 60.71 | 50.40 | 44.74 | 5.65 | 1.09 | 9.23 | 39.29 | 31.25 | 1.88 | 3.87 | 1.29 | 0.99 |

Table 7: Val split Pick policy trajectory labeling on 1000 episodes per target object. All numbers are percentages. Best viewed zoomed.

| TASK | OBJ | TYPE | SoR | SaeR | (i) | (ii) | (iii) | (iv) | FR | (v) | (vi) | (vii) | (viii) | (ix) |
|---|---|---|---|---|---|---|---|---|---|---|---|---|---|---|
| TH | 002 | RL-Per | 80.06 | 70.63 | 68.95 | 1.69 | 0.30 | 9.13 | 19.94 | 15.77 | 3.67 | 0.30 | 0.00 | 0.20 |
| | | RL-All | 78.57 | 61.71 | 59.82 | 1.88 | 0.00 | 16.87 | 21.43 | 15.18 | 5.36 | 0.69 | 0.10 | 0.10 |
| | | IL | 73.61 | 72.22 | 69.64 | 2.58 | 0.20 | 1.19 | 26.39 | 21.03 | 0.20 | 1.69 | 1.09 | 2.38 |
| | 003 | RL-Per | 73.41 | 64.38 | 61.61 | 2.78 | 0.10 | 8.93 | 26.59 | 16.67 | 6.94 | 1.98 | 0.99 | 0.00 |
| | | RL-All | 33.73 | 26.29 | 25.79 | 0.50 | 0.10 | 7.34 | 66.27 | 33.13 | 7.34 | 24.50 | 1.09 | 0.20 |
| | | IL | 16.67 | 16.17 | 15.58 | 0.60 | 0.10 | 0.40 | 83.33 | 47.52 | 0.10 | 33.93 | 0.69 | 1.09 |
| | 004 | RL-Per | 77.28 | 69.44 | 66.96 | 2.48 | 0.00 | 7.84 | 22.72 | 13.39 | 6.85 | 1.29 | 0.79 | 0.40 |
| | | RL-All | 75.20 | 57.44 | 54.07 | 3.37 | 0.00 | 17.76 | 24.80 | 15.97 | 6.35 | 1.88 | 0.20 | 0.40 |
| | | IL | 59.62 | 57.34 | 54.76 | 2.58 | 0.20 | 2.08 | 40.38 | 27.78 | 0.40 | 7.24 | 1.19 | 3.77 |
| | 005 | RL-Per | 81.15 | 78.08 | 75.79 | 2.28 | 0.10 | 2.98 | 18.85 | 15.87 | 2.78 | 0.20 | 0.00 | 0.00 |
| | | RL-All | 75.50 | 61.90 | 58.43 | 3.47 | 0.10 | 13.49 | 24.50 | 18.55 | 5.06 | 0.20 | 0.40 | 0.30 |
| | | IL | 71.03 | 69.44 | 65.77 | 3.67 | 0.20 | 1.39 | 28.97 | 21.92 | 0.99 | 2.28 | 0.79 | 2.98 |
| | 007 | RL-Per | 82.74 | 77.98 | 76.69 | 1.29 | 0.00 | 4.76 | 17.26 | 11.61 | 5.46 | 0.20 | 0.00 | 0.00 |
| | | RL-All | 77.78 | 69.44 | 68.75 | 0.69 | 0.00 | 8.33 | 22.22 | 15.18 | 6.55 | 0.40 | 0.00 | 0.10 |
| | | IL | 72.72 | 71.23 | 67.66 | 3.57 | 0.10 | 1.39 | 27.28 | 22.22 | 1.29 | 1.79 | 0.10 | 1.88 |
| | 008 | RL-Per | 75.40 | 70.34 | 60.62 | 9.72 | 0.69 | 4.37 | 24.60 | 18.45 | 2.58 | 2.48 | 0.79 | 0.30 |
| | | RL-All | 72.02 | 63.19 | 55.95 | 7.24 | 0.40 | 8.43 | 27.98 | 19.54 | 5.26 | 2.38 | 0.79 | 0.00 |
| | | IL | 61.61 | 60.32 | 54.76 | 5.56 | 0.50 | 0.79 | 38.39 | 29.37 | 0.89 | 3.87 | 2.58 | 1.69 |
| | 009 | RL-Per | 78.57 | 74.70 | 69.15 | 5.56 | 0.30 | 3.57 | 21.43 | 14.98 | 5.26 | 0.89 | 0.20 | 0.10 |
| | | RL-All | 71.63 | 60.62 | 52.78 | 7.84 | 0.30 | 10.71 | 28.37 | 20.34 | 6.35 | 0.60 | 0.99 | 0.10 |
| | | IL | 64.48 | 62.20 | 54.96 | 7.24 | 1.09 | 1.19 | 35.52 | 28.57 | 1.09 | 2.78 | 1.88 | 1.19 |
| | 010 | RL-Per | 82.24 | 53.37 | 50.00 | 3.37 | 0.00 | 28.87 | 17.76 | 13.19 | 4.07 | 0.50 | 0.00 | 0.00 |
| | | RL-All | 74.90 | 63.99 | 56.94 | 7.04 | 0.10 | 10.81 | 25.10 | 19.25 | 4.46 | 0.79 | 0.30 | 0.30 |
| | | IL | 55.75 | 52.68 | 46.83 | 5.85 | 0.10 | 2.98 | 44.25 | 27.88 | 0.30 | 9.42 | 1.69 | 4.96 |
| | 024 | RL-Per | 79.56 | 74.31 | 63.79 | 10.52 | 0.50 | 4.76 | 20.44 | 15.58 | 4.46 | 0.20 | 0.20 | 0.00 |
| | | RL-All | 69.74 | 62.90 | 53.37 | 9.52 | 0.00 | 6.85 | 30.26 | 22.32 | 7.14 | 0.50 | 0.20 | 0.10 |
| | | IL | 58.04 | 54.76 | 44.94 | 9.82 | 0.40 | 2.88 | 41.96 | 29.96 | 0.30 | 7.04 | 1.39 | 3.27 |
| | all | RL-Per | 77.48 | 69.44 | 65.77 | 3.67 | 0.30 | 7.74 | 22.52 | 16.57 | 4.86 | 0.89 | 0.00 | 0.20 |
| | | RL-All | 68.15 | 57.34 | 53.97 | 3.37 | 0.00 | 10.81 | 31.85 | 21.73 | 5.36 | 4.27 | 0.10 | 0.40 |
| | | IL | 59.03 | 57.04 | 53.87 | 3.17 | 0.20 | 1.79 | 40.97 | 28.47 | 0.30 | 8.43 | 1.39 | 2.38 |
| PG | 002 | RL-Per | 84.62 | 67.56 | 63.69 | 3.87 | 0.89 | 16.17 | 15.38 | 12.10 | 2.88 | 0.10 | 0.10 | 0.20 |
| | | RL-All | 76.98 | 54.27 | 41.47 | 12.80 | 2.68 | 20.04 | 23.02 | 20.34 | 0.99 | 1.29 | 0.30 | 0.10 |
| | | IL | 74.11 | 68.45 | 64.58 | 3.87 | 1.19 | 4.46 | 25.89 | 20.93 | 0.50 | 1.29 | 2.18 | 0.99 |
| | 003 | RL-Per | 56.15 | 42.56 | 39.68 | 2.88 | 2.38 | 11.21 | 43.85 | 30.46 | 2.08 | 9.72 | 1.39 | 0.20 |
| | | RL-All | 14.19 | 11.41 | 10.02 | 1.39 | 0.60 | 2.18 | 85.81 | 57.24 | 0.69 | 26.88 | 0.99 | 0.00 |
| | | IL | 18.25 | 16.77 | 16.37 | 0.40 | 0.69 | 0.79 | 81.75 | 61.90 | 0.20 | 16.67 | 2.58 | 0.40 |
| | 004 | RL-Per | 69.74 | 60.52 | 57.24 | 3.27 | 0.30 | 8.93 | 30.26 | 25.00 | 2.08 | 1.29 | 0.40 | 1.49 |
| | | RL-All | 71.23 | 47.82 | 38.99 | 8.83 | 4.17 | 19.25 | 28.77 | 23.61 | 1.29 | 2.78 | 0.89 | 0.20 |
| | | IL | 58.43 | 53.57 | 51.19 | 2.38 | 0.40 | 4.46 | 41.57 | 34.33 | 0.30 | 4.17 | 1.49 | 1.29 |
| | 005 | RL-Per | 76.69 | 64.88 | 60.12 | 4.76 | 0.20 | 11.61 | 23.31 | 21.43 | 0.69 | 0.60 | 0.30 | 0.30 |
| | | RL-All | 75.60 | 53.97 | 40.87 | 13.10 | 1.19 | 20.44 | 24.40 | 21.03 | 1.09 | 1.79 | 0.30 | 0.30 |
| | | IL | 75.00 | 68.35 | 65.67 | 2.68 | 0.30 | 6.35 | 25.00 | 20.54 | 0.79 | 1.19 | 1.29 | 1.19 |
| | 007 | RL-Per | 76.98 | 56.25 | 50.69 | 5.56 | 0.30 | 20.44 | 23.02 | 19.35 | 2.58 | 0.60 | 0.40 | 0.10 |
| | | RL-All | 76.98 | 56.15 | 41.96 | 14.19 | 0.40 | 20.44 | 23.02 | 19.64 | 1.59 | 1.79 | 0.00 | 0.00 |
| | | IL | 72.02 | 69.84 | 65.48 | 4.37 | 0.10 | 2.08 | 27.98 | 21.83 | 1.69 | 2.78 | 0.79 | 0.89 |
| | 008 | RL-Per | 73.21 | 48.91 | 37.60 | 11.31 | 0.69 | 23.61 | 26.79 | 22.52 | 1.59 | 1.98 | 0.30 | 0.40 |
| | | RL-All | 58.93 | 42.86 | 24.70 | 18.15 | 3.67 | 12.40 | 41.07 | 29.07 | 2.38 | 6.65 | 2.98 | 0.00 |
| | | IL | 44.35 | 37.40 | 32.04 | 5.36 | 1.88 | 5.06 | 55.65 | 44.15 | 1.59 | 5.65 | 3.47 | 0.79 |
| | 009 | RL-Per | 70.73 | 38.00 | 32.24 | 5.75 | 1.88 | 30.85 | 29.27 | 26.19 | 1.69 | 0.99 | 0.20 | 0.20 |
| | | RL-All | 62.90 | 46.92 | 29.37 | 17.56 | 2.88 | 13.10 | 37.10 | 26.09 | 2.38 | 5.36 | 2.98 | 0.30 |
| | | IL | 44.54 | 38.79 | 33.73 | 5.06 | 1.79 | 3.97 | 55.46 | 42.56 | 1.19 | 7.64 | 3.67 | 0.40 |
| | 010 | RL-Per | 74.50 | 55.26 | 48.61 | 6.65 | 0.60 | 18.65 | 25.50 | 23.91 | 0.50 | 0.79 | 0.30 | 0.00 |
| | | RL-All | 70.04 | 49.40 | 34.42 | 14.98 | 2.88 | 17.76 | 29.96 | 23.41 | 2.38 | 3.57 | 0.60 | 0.00 |
| | | IL | 59.23 | 57.04 | 52.18 | 4.86 | 0.50 | 1.69 | 40.77 | 33.93 | 0.69 | 2.98 | 2.18 | 0.99 |
| | 024 | RL-Per | 78.17 | 55.65 | 39.09 | 16.57 | 0.10 | 22.42 | 21.83 | 15.08 | 5.75 | 0.50 | 0.10 | 0.40 |
| | | RL-All | 61.21 | 37.00 | 25.79 | 11.21 | 1.19 | 23.02 | 38.79 | 35.81 | 0.69 | 1.49 | 0.79 | 0.00 |
| | | IL | 22.82 | 20.14 | 16.27 | 3.87 | 0.30 | 2.38 | 77.18 | 65.28 | 0.79 | 7.64 | 0.89 | 2.58 |
| | all | RL-Per | 72.32 | 53.47 | 45.93 | 7.54 | 0.50 | 18.35 | 27.68 | 23.02 | 1.39 | 2.38 | 0.79 | 0.10 |
| | | RL-All | 62.10 | 43.75 | 31.25 | 12.50 | 1.79 | 16.57 | 37.90 | 28.47 | 0.89 | 7.54 | 0.89 | 0.10 |
| | | IL | 52.78 | 47.52 | 43.75 | 3.77 | 0.50 | 4.76 | 47.22 | 36.90 | 0.69 | 6.55 | 1.79 | 1.29 |
| ST | 013 | RL-Per | 80.36 | 58.43 | 48.02 | 10.42 | 0.20 | 21.73 | 19.64 | 17.96 | 0.30 | 0.00 | 0.20 | 1.19 |
| | | RL-All | 66.67 | 26.29 | 19.74 | 6.55 | 0.10 | 40.28 | 33.33 | 31.94 | 0.89 | 0.30 | 0.20 | 0.00 |
| | | IL | 78.08 | 55.56 | 51.09 | 4.46 | 1.49 | 21.03 | 21.92 | 18.06 | 1.49 | 0.30 | 0.69 | 1.39 |
| | 024 | RL-Per | 93.65 | 83.33 | 71.83 | 11.51 | 0.20 | 10.12 | 6.35 | 5.85 | 0.10 | 0.20 | 0.20 | 0.00 |
| | | RL-All | 89.98 | 77.38 | 62.00 | 15.38 | 0.40 | 12.20 | 10.02 | 9.33 | 0.30 | 0.20 | 0.20 | 0.00 |
| | | IL | 62.20 | 54.86 | 48.61 | 6.25 | 0.20 | 7.14 | 37.80 | 26.59 | 2.28 | 5.95 | 1.39 | 1.59 |
| | all | RL-Per | 88.49 | 70.04 | 61.61 | 8.43 | 0.00 | 18.45 | 11.51 | 10.42 | 0.00 | 0.00 | 0.40 | 0.69 |
| | | RL-All | 79.86 | 52.58 | 40.58 | 12.00 | 0.00 | 27.08 | 20.14 | 19.44 | 0.30 | 0.10 | 0.20 | 0.10 |
| | | IL | 72.62 | 58.63 | 52.98 | 5.65 | 0.20 | 13.79 | 27.38 | 21.03 | 1.88 | 2.68 | 1.09 | 0.69 |

Table 8: Train split Place policy trajectory labeling on 1000 episodes per target object. All numbers are percentages. Best viewed zoomed.

| TASK | OBJ | TYPE | SoR | SaeR | (i) | (ii) | (iii) | (iv) | (v) | FR | (vi) | (vii) | (viii) | (ix) | (x) | (xi) | (xii) |
|---|---|---|---|---|---|---|---|---|---|---|---|---|---|---|---|---|---|
| TH | 002 | RL-Per | 69.44 | 65.38 | 44.35 | 19.05 | 1.39 | 1.98 | 2.68 | 30.56 | 20.14 | 0.00 | 2.88 | 2.28 | 3.67 | 1.59 | 0.00 |
| | | RL-All | 67.76 | 58.04 | 26.69 | 29.37 | 0.79 | 1.98 | 8.93 | 32.24 | 19.15 | 0.00 | 1.19 | 4.27 | 6.85 | 0.79 | 0.00 |
| | | IL | 70.44 | 61.11 | 51.59 | 7.84 | 0.69 | 1.69 | 8.63 | 29.56 | 18.25 | 0.00 | 3.47 | 4.66 | 2.38 | 0.40 | 0.40 |
| | 003 | RL-Per | 54.07 | 40.48 | 27.38 | 1.79 | 2.38 | 11.31 | 11.21 | 45.93 | 31.94 | 0.00 | 0.50 | 4.17 | 0.99 | 7.64 | 0.69 |
| | | RL-All | 52.58 | 42.86 | 22.92 | 18.25 | 0.99 | 1.69 | 8.73 | 47.42 | 31.05 | 0.00 | 0.79 | 8.43 | 5.95 | 0.89 | 0.30 |
| | | IL | 47.32 | 35.71 | 20.54 | 4.86 | 4.37 | 10.32 | 7.24 | 52.68 | 32.74 | 0.20 | 4.66 | 5.06 | 5.06 | 4.27 | 0.69 |
| | 004 | RL-Per | 54.76 | 46.23 | 6.25 | 34.92 | 0.50 | 5.06 | 8.04 | 45.24 | 24.60 | 0.00 | 5.16 | 1.29 | 6.85 | 6.85 | 0.50 |
| | | RL-All | 55.06 | 47.22 | 21.13 | 24.40 | 0.50 | 1.69 | 7.34 | 44.94 | 24.21 | 0.00 | 1.59 | 5.16 | 7.94 | 5.95 | 0.10 |
| | | IL | 51.59 | 45.44 | 15.97 | 27.08 | 0.89 | 2.38 | 5.26 | 48.41 | 22.32 | 0.10 | 5.16 | 4.66 | 6.65 | 9.03 | 0.50 |
| | 005 | RL-Per | 65.97 | 56.05 | 43.06 | 8.13 | 0.30 | 4.86 | 9.62 | 34.03 | 24.01 | 0.10 | 1.49 | 4.37 | 1.39 | 2.48 | 0.20 |
| | | RL-All | 65.77 | 59.23 | 21.53 | 35.42 | 0.69 | 2.28 | 5.85 | 34.23 | 22.62 | 0.10 | 2.18 | 2.48 | 6.55 | 0.20 | 0.10 |
| | | IL | 62.90 | 55.95 | 41.27 | 12.00 | 0.69 | 2.68 | 6.25 | 37.10 | 21.92 | 0.60 | 7.04 | 3.97 | 3.08 | 0.40 | 0.10 |
| | 007 | RL-Per | 73.61 | 66.37 | 29.96 | 33.93 | 0.10 | 2.48 | 7.14 | 26.39 | 17.96 | 0.00 | 0.69 | 1.39 | 6.05 | 0.00 | 0.30 |
| | | RL-All | 70.54 | 63.19 | 23.12 | 39.09 | 0.20 | 0.99 | 7.14 | 29.46 | 18.95 | 0.00 | 2.38 | 3.47 | 4.17 | 0.30 | 0.20 |
| | | IL | 70.93 | 63.10 | 33.93 | 27.48 | 0.30 | 1.69 | 7.54 | 29.07 | 17.76 | 0.40 | 4.86 | 1.59 | 3.77 | 0.10 | 0.60 |
| | 008 | RL-Per | 75.40 | 69.74 | 30.36 | 35.71 | 0.10 | 3.67 | 5.56 | 24.60 | 19.05 | 0.00 | 0.79 | 0.69 | 1.79 | 2.08 | 0.20 |
| | | RL-All | 68.06 | 61.51 | 20.93 | 38.79 | 0.00 | 1.79 | 6.55 | 31.94 | 23.51 | 0.00 | 3.27 | 2.08 | 1.88 | 1.19 | 0.00 |
| | | IL | 68.65 | 60.12 | 38.49 | 19.84 | 0.20 | 1.79 | 8.33 | 31.35 | 21.43 | 0.50 | 2.98 | 1.49 | 1.88 | 2.98 | 0.10 |
| | 009 | RL-Per | 68.35 | 53.67 | 13.69 | 38.79 | 0.10 | 1.19 | 14.58 | 31.65 | 24.11 | 0.40 | 2.08 | 1.49 | 2.98 | 0.40 | 0.20 |
| | | RL-All | 65.67 | 59.62 | 19.54 | 38.49 | 0.00 | 1.59 | 6.05 | 34.33 | 25.99 | 0.00 | 3.27 | 1.09 | 3.47 | 0.50 | 0.00 |
| | | IL | 59.62 | 51.09 | 35.81 | 13.29 | 0.00 | 1.98 | 8.53 | 40.38 | 27.48 | 0.69 | 4.56 | 2.98 | 1.19 | 2.68 | 0.79 |
| | 010 | RL-Per | 71.43 | 57.84 | 24.90 | 30.16 | 0.10 | 2.78 | 13.49 | 28.57 | 22.62 | 0.00 | 2.08 | 0.89 | 1.29 | 1.69 | 0.00 |
| | | RL-All | 73.31 | 64.98 | 24.01 | 39.48 | 0.10 | 1.49 | 8.23 | 26.69 | 17.46 | 0.00 | 1.98 | 2.28 | 4.07 | 0.89 | 0.00 |
| | | IL | 69.35 | 60.71 | 39.38 | 18.85 | 0.00 | 2.48 | 8.63 | 30.65 | 21.03 | 0.00 | 2.98 | 3.27 | 1.39 | 1.09 | 0.89 |
| | 024 | RL-Per | 70.73 | 60.62 | 45.24 | 7.64 | 0.00 | 7.74 | 10.12 | 29.27 | 21.53 | 0.00 | 0.40 | 5.36 | 0.79 | 0.79 | 0.40 |
| | | RL-All | 64.38 | 58.23 | 26.79 | 23.81 | 0.10 | 7.64 | 6.05 | 35.62 | 27.38 | 0.00 | 1.88 | 2.48 | 3.67 | 0.00 | 0.20 |
| | | IL | 64.78 | 57.54 | 43.06 | 9.62 | 0.10 | 4.86 | 7.14 | 35.22 | 20.83 | 0.10 | 5.36 | 6.45 | 2.18 | 0.00 | 0.30 |
| | all | RL-Per | 65.77 | 55.65 | 26.79 | 25.30 | 0.30 | 3.57 | 9.82 | 34.23 | 22.82 | 0.00 | 1.79 | 2.68 | 3.17 | 3.57 | 0.20 |
| | | RL-All | 63.69 | 56.65 | 23.12 | 31.75 | 0.50 | 1.79 | 6.55 | 36.31 | 24.50 | 0.10 | 2.38 | 2.58 | 4.76 | 1.79 | 0.20 |
| | | IL | 61.81 | 54.56 | 35.32 | 15.97 | 0.30 | 3.27 | 6.94 | 38.19 | 23.02 | 0.40 | 5.06 | 3.27 | 3.17 | 2.68 | 0.60 |
| PG | 002 | RL-Per | 62.50 | 52.48 | 14.88 | 33.93 | 1.29 | 3.67 | 8.73 | 37.50 | 26.69 | 0.10 | 6.05 | 0.60 | 1.79 | 1.98 | 0.30 |
| | | RL-All | 56.35 | 32.64 | 21.23 | 9.13 | 2.18 | 2.28 | 21.53 | 43.65 | 27.88 | 0.69 | 8.23 | 2.78 | 2.58 | 1.09 | 0.40 |
| | | IL | 56.65 | 50.40 | 19.84 | 29.17 | 2.78 | 1.39 | 3.47 | 43.35 | 23.41 | 0.40 | 11.21 | 3.77 | 4.17 | 0.00 | 0.40 |
| | 003 | RL-Per | 52.28 | 49.21 | 22.92 | 24.31 | 0.99 | 1.98 | 2.08 | 47.72 | 30.85 | 0.10 | 5.56 | 4.17 | 2.08 | 4.86 | 0.10 |
| | | RL-All | 47.82 | 28.57 | 20.93 | 4.76 | 3.57 | 2.88 | 15.67 | 52.18 | 34.52 | 0.10 | 6.94 | 8.13 | 1.49 | 0.50 | 0.50 |
| | | IL | 41.57 | 33.53 | 17.76 | 15.28 | 2.78 | 0.50 | 5.26 | 58.43 | 27.38 | 0.10 | 13.79 | 8.83 | 4.56 | 0.60 | 3.17 |
| | 004 | RL-Per | 55.95 | 46.92 | 22.52 | 20.54 | 0.10 | 3.87 | 8.93 | 44.05 | 32.34 | 0.00 | 3.57 | 2.68 | 2.68 | 2.58 | 0.20 |
| | | RL-All | 52.08 | 32.84 | 23.12 | 6.94 | 0.20 | 2.78 | 19.05 | 47.92 | 29.96 | 0.00 | 6.15 | 5.36 | 2.08 | 3.87 | 0.50 |
| | | IL | 49.11 | 44.35 | 30.16 | 12.70 | 0.40 | 1.49 | 4.37 | 50.89 | 30.06 | 0.30 | 8.83 | 6.75 | 3.47 | 1.09 | 0.40 |
| | 005 | RL-Per | 62.60 | 50.89 | 39.58 | 7.44 | 0.50 | 3.87 | 11.21 | 37.40 | 25.20 | 0.10 | 4.46 | 5.16 | 2.38 | 0.10 | 0.00 |
| | | RL-All | 56.15 | 37.20 | 26.39 | 8.53 | 0.20 | 2.28 | 18.75 | 43.85 | 29.96 | 0.30 | 7.04 | 1.19 | 3.47 | 1.29 | 0.60 |
| | | IL | 56.94 | 51.98 | 41.17 | 8.53 | 0.89 | 2.28 | 4.07 | 43.06 | 27.08 | 0.30 | 6.45 | 4.76 | 3.37 | 0.00 | 1.09 |
| | 007 | RL-Per | 63.79 | 55.95 | 28.17 | 22.52 | 0.20 | 5.26 | 7.64 | 36.21 | 27.08 | 0.00 | 5.65 | 1.49 | 1.88 | 0.00 | 0.10 |
| | | RL-All | 56.35 | 35.91 | 23.12 | 11.61 | 0.00 | 1.19 | 20.44 | 43.65 | 28.57 | 0.30 | 8.33 | 1.29 | 1.49 | 3.57 | 0.10 |
| | | IL | 55.95 | 51.49 | 29.17 | 21.23 | 0.60 | 1.09 | 3.87 | 44.05 | 28.17 | 0.30 | 8.13 | 3.27 | 3.17 | 0.10 | 0.89 |
| | 008 | RL-Per | 62.30 | 52.08 | 16.96 | 32.14 | 0.20 | 2.98 | 10.02 | 37.70 | 23.12 | 0.10 | 7.24 | 2.48 | 3.47 | 1.09 | 0.20 |
| | | RL-All | 55.46 | 38.19 | 23.02 | 13.49 | 0.00 | 1.69 | 17.26 | 44.54 | 30.95 | 1.09 | 7.94 | 1.19 | 1.39 | 1.59 | 0.40 |
| | | IL | 54.27 | 50.69 | 27.18 | 19.64 | 0.10 | 3.87 | 3.47 | 45.73 | 23.21 | 0.99 | 9.23 | 4.07 | 4.46 | 1.79 | 1.98 |
| | 009 | RL-Per | 63.39 | 54.56 | 24.31 | 28.77 | 0.10 | 1.49 | 8.73 | 36.61 | 25.89 | 0.10 | 6.35 | 1.88 | 2.38 | 0.00 | 0.00 |
| | | RL-All | 52.88 | 33.13 | 18.75 | 12.20 | 0.00 | 2.18 | 19.74 | 47.12 | 35.42 | 0.50 | 5.95 | 2.38 | 0.99 | 1.59 | 0.30 |
| | | IL | 56.65 | 53.77 | 35.62 | 15.77 | 0.10 | 2.38 | 2.78 | 43.35 | 24.90 | 0.79 | 8.83 | 4.07 | 1.98 | 0.89 | 1.88 |
| | 010 | RL-Per | 63.10 | 41.77 | 18.15 | 19.35 | 0.30 | 4.27 | 21.03 | 36.90 | 31.75 | 0.10 | 3.57 | 0.60 | 0.60 | 0.20 | 0.10 |
| | | RL-All | 57.74 | 38.39 | 22.22 | 13.29 | 0.10 | 2.88 | 19.25 | 42.26 | 27.98 | 0.60 | 6.94 | 1.98 | 2.58 | 1.98 | 0.20 |
| | | IL | 55.56 | 44.94 | 27.78 | 15.08 | 0.10 | 2.08 | 10.52 | 44.44 | 30.85 | 0.40 | 6.55 | 3.17 | 0.99 | 0.10 | 2.38 |
| | 024 | RL-Per | 62.30 | 46.43 | 13.89 | 22.82 | 0.00 | 9.72 | 15.87 | 37.70 | 27.78 | 0.00 | 2.88 | 0.89 | 6.05 | 0.00 | 0.10 |
| | | RL-All | 51.98 | 29.56 | 12.00 | 9.72 | 0.00 | 7.84 | 22.42 | 48.02 | 38.79 | 0.00 | 4.37 | 1.88 | 2.58 | 0.20 | 0.20 |
| | | IL | 50.30 | 46.73 | 21.33 | 19.15 | 0.30 | 6.25 | 3.27 | 49.70 | 31.55 | 0.10 | 12.20 | 2.28 | 3.37 | 0.00 | 0.20 |
| | all | RL-Per | 60.22 | 48.61 | 22.32 | 22.72 | 0.30 | 3.57 | 11.31 | 39.78 | 27.38 | 0.20 | 5.75 | 2.48 | 2.68 | 0.99 | 0.30 |
| | | RL-All | 53.37 | 33.13 | 18.85 | 10.12 | 0.89 | 4.17 | 19.35 | 46.63 | 33.13 | 0.60 | 7.34 | 2.18 | 1.69 | 1.59 | 0.10 |
| | | IL | 50.00 | 45.63 | 25.99 | 17.26 | 1.09 | 2.38 | 3.27 | 50.00 | 28.67 | 0.50 | 11.01 | 4.96 | 3.37 | 0.40 | 1.09 |
| ST | 013 | RL-Per | 68.95 | 61.71 | 18.45 | 39.38 | 1.59 | 3.87 | 5.65 | 31.05 | 22.02 | 0.00 | 0.60 | 1.69 | 1.98 | 4.76 | 0.00 |
| | | RL-All | 74.80 | 67.36 | 39.29 | 24.11 | 0.99 | 3.97 | 6.45 | 25.20 | 18.95 | 0.30 | 0.60 | 2.28 | 2.38 | 0.40 | 0.30 |
| | | IL | 65.58 | 62.50 | 11.31 | 46.33 | 1.29 | 4.86 | 1.79 | 34.42 | 21.03 | 0.89 | 5.95 | 1.49 | 4.66 | 0.00 | 0.40 |
| | 024 | RL-Per | 78.67 | 76.39 | 33.33 | 29.46 | 0.50 | 13.59 | 1.79 | 21.33 | 16.07 | 0.00 | 0.40 | 1.88 | 2.28 | 0.40 | 0.30 |
| | | RL-All | 71.33 | 62.90 | 30.65 | 12.00 | 0.20 | 20.24 | 8.23 | 28.67 | 21.23 | 0.00 | 1.98 | 3.27 | 1.88 | 0.20 | 0.10 |
| | | IL | 73.51 | 72.72 | 23.41 | 39.68 | 0.60 | 9.62 | 0.20 | 26.49 | 10.52 | 0.00 | 6.75 | 4.17 | 4.96 | 0.00 | 0.10 |
| | all | RL-Per | 73.31 | 68.35 | 25.20 | 34.23 | 0.69 | 8.93 | 4.27 | 26.69 | 17.56 | 0.20 | 0.79 | 2.58 | 2.88 | 2.38 | 0.30 |
| | | RL-All | 72.82 | 65.08 | 35.02 | 17.36 | 0.89 | 12.70 | 6.85 | 27.18 | 20.14 | 0.20 | 1.29 | 2.18 | 2.88 | 0.20 | 0.30 |
| | | IL | 71.23 | 69.15 | 19.05 | 43.55 | 0.89 | 6.55 | 1.19 | 28.77 | 14.19 | 0.50 | 6.05 | 3.17 | 4.56 | 0.00 | 0.30 |

Table 9: Val split Place policy trajectory labeling on 1000 episodes per target object. All numbers are percentages. Best viewed zoomed.

| TASK | OBJ | TYPE | SoR | SaeR | (i) | (ii) | (iii) | (iv) | (v) | FR | (vi) | (vii) | (viii) | (ix) | (x) | (xi) | (xii) |
|---|---|---|---|---|---|---|---|---|---|---|---|---|---|---|---|---|---|
| TH | 002 | RL-Per | 62.70 | 58.23 | 37.40 | 18.65 | 1.79 | 2.18 | 2.68 | 37.30 | 25.10 | 0.10 | 3.27 | 3.17 | 3.17 | 2.28 | 0.20 |
| | | RL-All | 66.37 | 58.04 | 26.88 | 29.17 | 0.60 | 1.98 | 7.74 | 33.63 | 22.02 | 0.00 | 1.39 | 3.27 | 4.96 | 1.79 | 0.20 |
| | | IL | 69.44 | 60.71 | 50.00 | 9.03 | 1.59 | 1.69 | 7.14 | 30.56 | 20.73 | 0.10 | 3.17 | 3.87 | 1.49 | 0.79 | 0.40 |
| | 003 | RL-Per | 56.65 | 40.87 | 27.98 | 1.98 | 3.27 | 10.91 | 12.50 | 43.35 | 28.17 | 0.00 | 0.40 | 4.37 | 1.39 | 8.43 | 0.60 |
| | | RL-All | 53.57 | 44.15 | 26.09 | 15.28 | 1.39 | 2.78 | 8.04 | 46.43 | 30.95 | 0.00 | 1.09 | 10.42 | 3.67 | 0.30 | 0.00 |
| | | IL | 46.23 | 36.21 | 19.84 | 4.86 | 4.86 | 11.51 | 5.16 | 53.77 | 34.82 | 0.00 | 2.98 | 5.85 | 4.76 | 3.97 | 1.39 |
| | 004 | RL-Per | 52.88 | 44.54 | 6.05 | 32.14 | 0.79 | 6.35 | 7.54 | 47.12 | 24.90 | 0.00 | 4.56 | 2.28 | 9.13 | 5.56 | 0.69 |
| | | RL-All | 52.88 | 45.73 | 21.92 | 22.22 | 0.50 | 1.59 | 6.65 | 47.12 | 26.79 | 0.00 | 1.88 | 5.46 | 7.24 | 5.36 | 0.40 |
| | | IL | 53.77 | 48.12 | 18.25 | 26.79 | 0.40 | 3.08 | 5.26 | 46.23 | 23.41 | 0.10 | 5.85 | 3.37 | 6.05 | 7.04 | 0.40 |
| | 005 | RL-Per | 65.97 | 58.83 | 48.02 | 6.15 | 0.40 | 4.66 | 6.75 | 34.03 | 22.62 | 0.00 | 1.69 | 5.26 | 0.89 | 2.88 | 0.69 |
| | | RL-All | 67.06 | 59.23 | 25.89 | 31.45 | 0.50 | 1.88 | 7.34 | 32.94 | 21.23 | 0.00 | 1.39 | 3.27 | 6.65 | 0.40 | 0.00 |
| | | IL | 65.08 | 59.82 | 47.92 | 9.92 | 0.60 | 1.98 | 4.66 | 34.92 | 21.43 | 0.40 | 4.66 | 3.77 | 3.87 | 0.20 | 0.60 |
| | 007 | RL-Per | 72.22 | 64.98 | 30.36 | 32.24 | 0.00 | 2.38 | 7.24 | 27.78 | 19.44 | 0.00 | 0.89 | 1.88 | 5.16 | 0.00 | 0.40 |
| | | RL-All | 69.74 | 64.19 | 25.00 | 38.10 | 0.20 | 1.09 | 5.36 | 30.26 | 20.93 | 0.00 | 1.69 | 2.28 | 4.66 | 0.50 | 0.10 |
| | | IL | 71.63 | 64.98 | 35.12 | 27.58 | 0.30 | 2.28 | 6.35 | 28.37 | 19.35 | 0.00 | 2.78 | 2.78 | 3.17 | 0.00 | 0.30 |
| | 008 | RL-Per | 75.30 | 69.44 | 28.37 | 39.48 | 0.00 | 1.59 | 5.85 | 24.70 | 18.65 | 0.00 | 0.99 | 0.69 | 1.79 | 2.28 | 0.30 |
| | | RL-All | 71.13 | 62.80 | 22.52 | 38.89 | 0.00 | 1.39 | 8.33 | 28.87 | 21.63 | 0.00 | 1.69 | 1.09 | 2.98 | 1.39 | 0.10 |
| | | IL | 73.41 | 65.87 | 44.54 | 19.74 | 0.00 | 1.59 | 7.54 | 26.59 | 18.75 | 0.20 | 1.88 | 0.79 | 0.99 | 3.57 | 0.40 |
| | 009 | RL-Per | 68.55 | 54.27 | 17.96 | 34.33 | 0.00 | 1.98 | 14.29 | 31.45 | 24.70 | 0.20 | 1.49 | 0.99 | 3.37 | 0.60 | 0.10 |
| | | RL-All | 63.49 | 57.24 | 20.54 | 34.82 | 0.00 | 1.88 | 6.25 | 36.51 | 28.27 | 0.00 | 3.57 | 1.29 | 2.68 | 0.69 | 0.00 |
| | | IL | 58.23 | 50.20 | 35.12 | 12.00 | 0.10 | 3.08 | 7.94 | 41.77 | 29.07 | 0.50 | 5.06 | 1.29 | 2.18 | 2.78 | 0.89 |
| | 010 | RL-Per | 68.75 | 54.56 | 23.41 | 27.18 | 0.10 | 3.97 | 14.09 | 31.25 | 25.20 | 0.00 | 1.98 | 0.89 | 1.49 | 1.59 | 0.10 |
| | | RL-All | 68.15 | 59.33 | 21.13 | 37.20 | 0.10 | 0.99 | 8.73 | 31.85 | 20.73 | 0.00 | 2.68 | 2.28 | 5.36 | 0.69 | 0.10 |
| | | IL | 66.37 | 61.41 | 40.28 | 18.15 | 0.00 | 2.98 | 4.96 | 33.63 | 24.70 | 0.00 | 3.57 | 2.28 | 1.09 | 1.29 | 0.69 |
| | 024 | RL-Per | 74.50 | 63.69 | 49.50 | 6.65 | 0.00 | 7.54 | 10.81 | 25.50 | 20.83 | 0.00 | 0.50 | 2.78 | 0.79 | 0.50 | 0.10 |
| | | RL-All | 67.66 | 58.73 | 27.88 | 22.32 | 0.00 | 8.53 | 8.93 | 32.34 | 24.40 | 0.10 | 1.19 | 2.08 | 4.37 | 0.10 | 0.10 |
| | | IL | 68.45 | 59.82 | 46.73 | 9.33 | 0.00 | 3.77 | 8.63 | 31.55 | 18.45 | 0.00 | 4.96 | 5.85 | 1.98 | 0.00 | 0.30 |
| | all | RL-Per | 65.97 | 56.45 | 32.04 | 20.24 | 0.20 | 4.17 | 9.33 | 34.03 | 24.31 | 0.00 | 1.09 | 2.08 | 3.67 | 2.68 | 0.20 |
| | | RL-All | 66.07 | 58.23 | 24.90 | 31.25 | 0.20 | 2.08 | 7.64 | 33.93 | 21.23 | 0.00 | 1.88 | 3.37 | 5.36 | 1.88 | 0.20 |
| | | IL | 63.79 | 56.94 | 37.30 | 15.38 | 0.89 | 4.27 | 5.95 | 36.21 | 21.83 | 0.20 | 3.97 | 3.37 | 3.77 | 2.68 | 0.40 |
| PG | 002 | RL-Per | 69.15 | 56.15 | 17.36 | 33.83 | 1.39 | 4.96 | 11.61 | 30.85 | 22.52 | 0.00 | 2.58 | 0.79 | 2.38 | 2.38 | 0.20 |
| | | RL-All | 58.83 | 29.96 | 21.92 | 6.45 | 1.98 | 1.59 | 26.88 | 41.17 | 29.27 | 1.39 | 3.67 | 3.08 | 2.48 | 1.19 | 0.10 |
| | | IL | 58.93 | 50.79 | 22.42 | 27.08 | 2.98 | 1.29 | 5.16 | 41.07 | 17.16 | 0.99 | 13.69 | 3.77 | 4.76 | 0.00 | 0.69 |
| | 003 | RL-Per | 58.33 | 53.37 | 24.50 | 25.50 | 1.69 | 3.37 | 3.27 | 41.67 | 27.18 | 0.00 | 3.97 | 4.46 | 3.47 | 2.98 | 0.50 |
| | | RL-All | 50.20 | 27.58 | 21.03 | 3.87 | 1.79 | 2.68 | 20.83 | 49.80 | 36.81 | 0.00 | 3.97 | 7.64 | 0.79 | 0.50 | 0.10 |
| | | IL | 44.15 | 39.58 | 20.73 | 17.56 | 2.48 | 1.29 | 2.08 | 55.85 | 20.63 | 0.00 | 17.36 | 8.13 | 5.75 | 0.30 | 3.67 |
| | 004 | RL-Per | 60.71 | 48.71 | 27.98 | 17.16 | 0.30 | 3.57 | 11.71 | 39.29 | 28.08 | 0.00 | 0.89 | 3.57 | 3.17 | 3.17 | 0.40 |
| | | RL-All | 59.52 | 30.56 | 20.73 | 5.85 | 0.00 | 3.97 | 28.97 | 40.48 | 27.98 | 0.00 | 2.28 | 4.37 | 1.09 | 4.46 | 0.30 |
| | | IL | 53.77 | 49.21 | 33.23 | 14.98 | 0.99 | 0.99 | 3.57 | 46.23 | 23.12 | 0.10 | 6.65 | 8.73 | 4.66 | 2.08 | 0.89 |
| | 005 | RL-Per | 67.36 | 47.72 | 38.99 | 5.36 | 1.39 | 3.37 | 18.25 | 32.64 | 23.02 | 0.10 | 1.69 | 5.85 | 1.49 | 0.40 | 0.10 |
| | | RL-All | 62.10 | 37.00 | 26.09 | 8.83 | 0.30 | 2.08 | 24.80 | 37.90 | 28.67 | 0.30 | 2.38 | 2.48 | 2.98 | 0.99 | 0.10 |
| | | IL | 62.20 | 55.56 | 44.25 | 8.23 | 1.59 | 3.08 | 5.06 | 37.80 | 19.64 | 0.30 | 11.11 | 3.77 | 1.88 | 0.00 | 1.09 |
| | 007 | RL-Per | 75.79 | 59.92 | 27.88 | 26.29 | 0.10 | 5.75 | 15.77 | 24.21 | 19.25 | 0.00 | 0.99 | 1.69 | 1.88 | 0.00 | 0.40 |
| | | RL-All | 62.70 | 35.02 | 22.72 | 10.32 | 0.10 | 1.98 | 27.58 | 37.30 | 28.97 | 0.40 | 2.78 | 0.79 | 1.88 | 2.28 | 0.20 |
| | | IL | 65.77 | 59.62 | 29.76 | 27.88 | 0.69 | 1.98 | 5.46 | 34.23 | 16.07 | 0.20 | 9.62 | 3.17 | 2.98 | 0.00 | 2.18 |
| | 008 | RL-Per | 71.53 | 58.33 | 17.46 | 36.90 | 0.00 | 3.97 | 13.19 | 28.47 | 18.45 | 0.10 | 2.28 | 2.28 | 4.17 | 0.89 | 0.30 |
| | | RL-All | 54.46 | 31.35 | 16.47 | 13.10 | 0.00 | 1.79 | 23.12 | 45.54 | 34.42 | 0.79 | 5.06 | 1.79 | 1.19 | 2.18 | 0.10 |
| | | IL | 59.42 | 55.56 | 28.87 | 23.51 | 0.30 | 3.17 | 3.57 | 40.58 | 17.46 | 0.69 | 9.82 | 3.37 | 4.37 | 2.78 | 2.08 |
| | 009 | RL-Per | 66.87 | 57.44 | 25.89 | 28.77 | 0.00 | 2.78 | 9.42 | 33.13 | 22.62 | 0.50 | 5.56 | 1.88 | 2.38 | 0.10 | 0.10 |
| | | RL-All | 54.86 | 33.04 | 20.24 | 11.31 | 0.00 | 1.49 | 21.83 | 45.14 | 32.24 | 0.79 | 6.25 | 1.98 | 1.49 | 1.88 | 0.50 |
| | | IL | 59.13 | 55.85 | 34.42 | 18.06 | 0.20 | 3.37 | 3.08 | 40.87 | 15.28 | 1.09 | 15.67 | 3.87 | 1.79 | 1.88 | 1.29 |
| | 010 | RL-Per | 68.06 | 39.38 | 17.86 | 19.64 | 0.10 | 1.88 | 28.57 | 31.94 | 27.98 | 0.00 | 1.39 | 0.79 | 1.39 | 0.40 | 0.00 |
| | | RL-All | 64.48 | 37.10 | 21.92 | 12.80 | 0.00 | 2.38 | 27.38 | 35.52 | 27.38 | 0.79 | 2.78 | 1.29 | 1.69 | 1.49 | 0.10 |
| | | IL | 62.30 | 49.31 | 29.76 | 17.66 | 0.50 | 1.88 | 12.50 | 37.70 | 21.63 | 0.50 | 8.13 | 3.37 | 1.69 | 0.10 | 2.28 |
| | 024 | RL-Per | 67.06 | 49.31 | 12.20 | 23.31 | 0.00 | 13.79 | 17.76 | 32.94 | 23.21 | 0.00 | 1.19 | 1.39 | 6.94 | 0.00 | 0.20 |
| | | RL-All | 56.75 | 28.37 | 12.50 | 9.72 | 0.00 | 6.15 | 28.37 | 43.25 | 37.00 | 0.00 | 1.69 | 1.69 | 2.58 | 0.30 | 0.00 |
| | | IL | 54.07 | 48.02 | 24.90 | 17.96 | 0.10 | 5.16 | 5.95 | 45.93 | 23.71 | 0.00 | 14.09 | 3.77 | 3.97 | 0.00 | 0.40 |
| | all | RL-Per | 65.67 | 52.38 | 24.31 | 22.72 | 0.79 | 5.36 | 12.50 | 34.33 | 25.10 | 0.00 | 2.28 | 2.58 | 3.37 | 0.99 | 0.00 |
| | | RL-All | 58.63 | 33.04 | 18.95 | 11.31 | 0.89 | 2.78 | 24.70 | 41.37 | 31.65 | 0.40 | 3.57 | 2.48 | 1.59 | 1.69 | 0.00 |
| | | IL | 56.75 | 52.08 | 30.16 | 19.15 | 0.60 | 2.78 | 4.07 | 43.25 | 21.33 | 0.40 | 12.00 | 3.97 | 2.98 | 0.69 | 1.88 |
| ST | 013 | RL-Per | 64.38 | 59.03 | 19.35 | 36.11 | 0.79 | 3.57 | 4.56 | 35.62 | 25.00 | 0.00 | 0.30 | 1.59 | 2.28 | 6.45 | 0.00 |
| | | RL-All | 67.96 | 60.91 | 40.08 | 16.27 | 1.09 | 4.56 | 5.95 | 32.04 | 25.60 | 0.10 | 1.19 | 2.78 | 1.69 | 0.60 | 0.10 |
| | | IL | 61.71 | 59.62 | 10.22 | 45.24 | 0.89 | 4.17 | 1.19 | 38.29 | 25.50 | 0.30 | 6.65 | 1.29 | 4.37 | 0.00 | 0.20 |
| | 024 | RL-Per | 69.44 | 66.47 | 30.95 | 24.21 | 0.30 | 11.31 | 2.68 | 30.56 | 25.10 | 0.00 | 0.40 | 1.88 | 2.68 | 0.20 | 0.30 |
| | | RL-All | 67.26 | 60.12 | 29.96 | 10.81 | 0.10 | 19.35 | 7.04 | 32.74 | 25.10 | 0.00 | 2.28 | 3.08 | 1.98 | 0.00 | 0.30 |
| | | IL | 65.67 | 64.68 | 21.03 | 37.10 | 0.40 | 6.55 | 0.60 | 34.33 | 18.65 | 0.10 | 8.83 | 2.68 | 3.97 | 0.00 | 0.10 |
| | all | RL-Per | 67.06 | 63.19 | 25.89 | 30.06 | 0.00 | 7.24 | 3.87 | 32.94 | 24.80 | 0.00 | 0.69 | 2.18 | 1.98 | 3.17 | 0.10 |
| | | RL-All | 68.25 | 61.31 | 33.73 | 16.07 | 0.40 | 11.51 | 6.55 | 31.75 | 24.90 | 0.00 | 1.19 | 3.08 | 2.58 | 0.00 | 0.00 |
| | | IL | 62.20 | 59.92 | 16.57 | 38.10 | 1.49 | 5.26 | 0.79 | 37.80 | 22.32 | 0.40 | 7.84 | 3.08 | 3.57 | 0.10 | 0.50 |

Table 10: Train split Open policy trajectory labeling on 1000 episodes per target articulation. All numbers are percentages. Best viewed zoomed.

| TASK | OBJ | TYPE | SoR | SaeR | (i) | (ii) | (iii) | FR | (iv) | (v) | (vi) | (vii) | (viii) | (ix) |
|------|-----|------|-----|------|-----|------|-------|-----|------|-----|------|-------|--------|------|
| ST | drawer | RL-Per | 84.92 | 84.92 | 84.92 | 0.00 | 0.00 | 15.08 | 11.41 | 1.69 | 0.00 | 1.49 | 0.20 | 0.30 |
|    |        | IL     | 79.86 | 79.86 | 79.86 | 0.00 | 0.00 | 20.14 | 13.59 | 0.89 | 1.09 | 3.67 | 0.10 | 0.79 |
|    | fridge | RL-Per | 83.43 | 82.24 | 82.24 | 0.00 | 1.19 | 16.57 | 14.58 | 0.20 | 0.00 | 1.49 | 0.20 | 0.10 |
|    |        | IL     | 74.01 | 68.85 | 68.85 | 0.00 | 5.16 | 25.99 | 22.12 | 0.99 | 0.10 | 1.69 | 0.40 | 0.69 |

Table 11: Val split Open policy trajectory labeling on 1000 episodes per target articulation. All numbers are percentages. Best viewed zoomed.

| TASK | OBJ | TYPE | SoR | SaeR | (i) | (ii) | (iii) | FR | (iv) | (v) | (vi) | (vii) | (viii) | (ix) |
|------|-----|------|-----|------|-----|------|-------|-----|------|-----|------|-------|--------|------|
| ST | drawer | RL-Per | 84.52 | 84.52 | 84.52 | 0.00 | 0.00 | 15.48 | 11.41 | 1.88 | 0.00 | 1.88 | 0.00 | 0.30 |
|    |        | IL     | 78.57 | 78.37 | 78.37 | 0.00 | 0.20 | 21.43 | 13.69 | 1.29 | 0.20 | 4.17 | 1.19 | 0.89 |
|    | fridge | RL-Per | 88.10 | 37.30 | 37.30 | 0.00 | 50.79 | 11.90 | 6.94 | 0.00 | 0.00 | 4.07 | 0.89 | 0.00 |
|    |        | IL     | 53.67 | 1.49 | 1.49 | 0.00 | 52.18 | 46.33 | 43.85 | 0.00 | 0.00 | 1.29 | 0.89 | 0.30 |

Table 12: Train split Close policy trajectory labeling on 1000 episodes per target articulation. All numbers are percentages. Best viewed zoomed.

| TASK | OBJ | TYPE | SoR | SaeR | (i) | (ii) | (iii) | FR | (iv) | (v) | (vi) | (vii) | (viii) | (ix) |
|------|-----|------|-----|------|-----|------|-------|-----|------|-----|------|-------|--------|------|
| ST | drawer | RL-Per | 88.79 | 88.79 | 88.79 | 0.00 | 0.00 | 11.21 | 3.17 | 2.68 | 0.30 | 0.89 | 4.17 | 0.00 |
|    |        | IL     | 88.39 | 88.39 | 88.39 | 0.00 | 0.00 | 11.61 | 3.27 | 3.08 | 0.00 | 0.79 | 4.46 | 0.00 |
|    | fridge | RL-Per | 86.81 | 25.00 | 25.00 | 0.00 | 61.81 | 13.19 | 13.19 | 0.00 | 0.00 | 0.00 | 0.00 | 0.00 |
|    |        | IL     | 86.90 | 29.96 | 29.96 | 0.00 | 56.94 | 13.10 | 12.90 | 0.00 | 0.00 | 0.20 | 0.00 | 0.00 |

Table 13: Val split Close policy trajectory labeling on 1000 episodes per target articulation. All numbers are percentages. Best viewed zoomed.

| TASK | OBJ | TYPE | SoR | SaeR | (i) | (ii) | (iii) | FR | (iv) | (v) | (vi) | (vii) | (viii) | (ix) |
|------|-----|------|-----|------|-----|------|-------|-----|------|-----|------|-------|--------|------|
| ST | drawer | RL-Per | 89.29 | 89.29 | 89.29 | 0.00 | 0.00 | 10.71 | 4.56 | 2.18 | 0.20 | 0.20 | 3.37 | 0.20 |
|    |        | IL     | 87.60 | 87.60 | 87.60 | 0.00 | 0.00 | 12.40 | 4.66 | 2.18 | 0.00 | 0.30 | 5.16 | 0.10 |
|    | fridge | RL-Per | 0.00 | 0.00 | 0.00 | 0.00 | 0.00 | 100.00 | 81.15 | 18.85 | 0.00 | 0.00 | 0.00 | 0.00 |
|    |        | IL     | 0.00 | 0.00 | 0.00 | 0.00 | 0.00 | 100.00 | 95.93 | 4.07 | 0.00 | 0.00 | 0.00 | 0.00 |

