# OpenReview forum: "ManiSkill-HAB: A Benchmark for Low-Level Manipulation in Home Rearrangement Tasks"
_ICLR.cc/2025/Conference — ICLR 2025 Poster_

### Official Review · Reviewer_HBjR · 2024-10-16

**Soundness:** 2
**Presentation:** 2
**Contribution:** 2
**Rating:** 5
**Confidence:** 4

**Summary:**

This paper introduces a robotic manipulation benchmark for home-scale rearrangement tasks, called ManiSkill-HAB. The authors implement the Home Assistant Benchmark (HAB) within the GPU-accelerated simulator ManiSkill3. The resulting environments achieve high simulation throughput, outperforming previous implementations by a factor of three. Additionally, the robot and object interactions are simulated with accurate rigid body physics to facilitate the learning of low-level manipulation skills. Finally, reinforcement learning (RL) and imitation learning (IL) models are trained to serve as baselines for future research. A rule-based trajectory filtering system is employed to selectively subsample demonstrations generated by RL policies.

**Strengths:**

**Highly Relevant Task**
- The problem of household rearrangement is highly relevant in robotics, and the research community greatly benefits from accessible, high-quality benchmarks. In this regard, the paper provides valuable foundational elements for future research to build upon.
- Furthermore, introducing accurate low-level control instead of relying on "magical" grasping is an important addition. For instance, using realistic initializations for the Place task by sampling grasp poses from the learned Pick policies is a compelling improvement.

**Writing**
- Overall, the writing is clear, and the logical flow of the paper effectively conveys the goals and proposed contributions. However, I noticed a few minor issues where I think the authors could improve the writing:
    - In line 213, when describing the observation space, you use the phrase "if the object/target is grasped." To enhance clarity, I suggest rephrasing it to something like "an indicator of whether the object/target is grasped".
    - In line 366, "pariwise" should be "pairwise".
    - The Open X-Embodiment reference is quite lengthy, taking up almost an entire page. To improve readability and structure, I recommend using "et al." after the first author’s name, rather than listing all the authors.

**Reproducibility**
- The code and data used in the paper are publicly available, and the experiments are described in detail.

***

Overall, the paper tackles an important problem, and making the environment code available to the research community will benefit other researchers by providing a foundation for future work.

**Weaknesses:**

**Novelty**
- The primary contribution of this work is the implementation of the HAB in the ManiSkill3 simulator. While this undoubtedly makes it more efficient for researchers to work on this problem, the novelty is somewhat limited. That said, the combination of low-level manipulation and long-horizon tasks, common in household settings, is intriguing. In particular, exploring how these tasks can be integrated to mitigate hand-off issues between independent modules could add significant value. However, by studying the subtasks in isolation and replacing navigation with robot teleportation (e.g., lines 348-350), the tasks are simplified to pure manipulation problems. It’s also worth noting that ManiSkill3 already includes a drawer-opening task with the Fetch robot out-of-the-box (https://maniskill.readthedocs.io/en/latest/tasks/mobile_manipulation/index.html).

**Baseline Methods and Evaluation**
- Of the four tasks studied (Pick, Place, Open, Close), SAC and PPO are each applied to two tasks, respectively. This makes it difficult to assess the relative difficulty of the tasks or the comparative strengths of the RL methods used. While Appendix A.4.3 explains that PPO was chosen for the Open and Close tasks to enable faster wall-time training, I find this reasoning unclear, especially since SAC demonstrated superior performance in both per-object and all-object grasping tasks. Given that the Open and Close tasks are not trivially solved, with success rates still below 90%, a structured evaluation of both SAC and PPO across all tasks would likely provide more meaningful insights.
- The rationale for using imitation learning with behavior cloning (BC) as a second baseline is unclear to me. First, since the RL teacher policies can be queried for expert actions, I would expect that using DAgger, which continually aggregates the dataset during training, would lead to better performance than relying on BC with a static dataset. The rule-based filtering of trajectories before adding them to the replay buffer in DAgger could be applied similarly here. Secondly, it’s unclear what is gained from this imitation learning step. Since the RL policies already operate from visual observations, there doesn’t appear to be any knowledge distillation that would justify the need for IL policies (for example to transfer the knowledge to a deployable observation space). If the goal is to shape behavior towards specific aspects of the RL policy’s learned behaviors to boost performance, we would expect an improvement in task success rates, which, according to Table 1, does not seem to be the case.
- The reported success rate is defined as "the percentage of trajectories that achieve success at least once in an episode" (lines 343-344). However, without a clear mechanism to infer from the used visual observations whether a subtask has been successfully completed and then halt execution, this measure seems overly optimistic. The success rate should either be measured at the end of an episode after a fixed time, or the policy should be equipped with the ability to terminate an episode when it determines that the task has been successfully completed. These adjustments would provide a more accurate reflection of the performance expected on a real system.
- While I understand that the primary focus of this work is on the simulation benchmark, incorporating real-robot transfer would greatly enhance the ability to assess how realistic the simulated rigid-body physics are in enabling low-level manipulation behaviors. This is particularly important given the claim of "realistic low-level control" made in the Conclusion. One concern I have is the classification of cumulative robot collisions exceeding 5000N as "excessive collisions." In collaborative robotics, the acceptable force range is typically an order of magnitude smaller (https://pubmed.ncbi.nlm.nih.gov/12820907/). Additionally, in lines 910-912 of the Appendix, it’s mentioned that violations of the 10000N force limit are the primary performance bottleneck in the Open and Close tasks. This raises questions about the deployability and realism of the learned manipulation behaviors.

***

Overall, while the implementation of the HAB benchmark in a GPU-accelerated simulator is valuable for the research community, the contribution is somewhat incremental. A more structured evaluation of the proposed environments, concerns about the realism of low-level control behaviors due to high observed collision forces, and the lack of calibration or transfer to a real-robot system— which would significantly strengthen the claim of realistic low-level control—leave room for further improvement.

**Questions:**

- In Figure 7 (Appendix A.4.3), you compare the performance of training per-object policies to that of a generalist policy capable of grasping all objects. Are the per-object policies trained concurrently, and does their combined experience total 1e7 environment samples? If each per-object policy is trained on 1e7 samples individually, it seems that the RL-All variant should also be allocated num_objects * 1e7 environment steps to ensure a fair comparison.
- You mention that by using depth images, you address a partially observable variant of the MDP (line 161). However, the policies are parameterized as MLPs following a CNN encoder. Are there any mechanisms, such as history-awareness or the concatenation of consecutive frames, to handle the partial observability? Alternatively, is there an argument that partial observability has minimal impact on the tasks being considered?

---

> ### Author Response · Authors · 2024-11-17
> **Response to Reviewer HBjR [1/2]**
>
> We sincerely thank you for your insightful feedback! We address the comments and questions below:
>
> > Question 1: all vs per object training
>
> We train per-object pick/place policies with 5e7 samples each, and all-object policies with 5e7 samples as well. Our reasoning for this is that SAC has limited vertically scalability, i.e. more GPUs/faster cores/etc have diminishing benefit to training wall-clock times. However, since our environments are quite GPU memory efficient (Figure 1), we can horizontally scale training by running more training runs in parallel on lower-end hardware (e.g. GPUs with less VRAM) or multiple runs per system on better hardware.
>
> While we agree this comparison is not fair from a sample efficiency perspective, we believe this shows a reasonable means to take advantage of our environments (whose speed makes sample efficiency less of a concern, and whose memory efficiency makes horizontal scaling more feasible).
>
> > Question 2: handling partial observability
>
> Thank you for pointing this out! We stack 3 frames per image (hand/head depth) when training to handle partial observability. We have noted this in the updated manuscript.
>
> > However, I noticed a few minor issues where I think the authors could improve the writing:
>
> Thank you for bringing these to our attention! We have made the relevant corrections to the manuscript.
>
> > a structured evaluation of both SAC and PPO across all tasks would likely provide more meaningful insights.
>
> We are currently running experiments with PPO for Pick/Place and SAC for Open/Close (3 seeds each) for a more structured comparison, and we will update the manuscript once completed.
>
> > I would expect that using DAgger, which continually aggregates the dataset during training, would lead to better performance than relying on BC with a static dataset [...] it’s unclear what is gained from this imitation learning step
>
> We expect a common use case for the community will be training IL algorithms on the static dataset we release (or a static dataset they generate with the provided code) and evaluating using our evaluation environment. To this end, the purpose of the IL algorithms is to (a) provide baselines on our static dataset (which we will be releasing for the community to use), and (b) explore the impact of trajectory filtering on performance and observed behavior (Section 6.2.2, where we find trajectory filtering helps bias policies towards desired behavior, but does not strictly prevent undesirable behavior).
>
> We acknowledge there are many ways that subtask performance can be improved, and we hope that our environments, results (both RL and IL), policy checkpoints, and provided static dataset (and data generation tools) will enable the community to research and develop novel methods in future work (e.g. the proposed DAgger + trajectory filtering).
>
> >  The success rate should either be measured at the end of an episode after a fixed time, or the policy should be equipped with the ability to terminate an episode when it determines that the task has been successfully completed
>
> For consistency with prior work, we use the same success rate and progressive completion rate metrics as the original implementations of the HAB and from prior work [1,2]. As in [1] and [2], when skill chaining, we proceed to the next skill as soon as the current subtask reaches first success; hence, we use success once rate to portray policy performance on subtasks.
>
> However, we agree that success once rate, success at end rate, and other measures can convey different information on policy performance. To be explicit about the performance of our policies, we provide full trajectory labeling statistics in tables 5-12, which provides information not only success/failure rates, but specific success and failure modes/behaviors. In the latest update to the manuscript, we have also added a column to each of these tables with success at end rates.

---

> ### Author Response · Authors · 2024-11-17
> **Response to Reviewer HBjR [2/2]**
>
> > One concern I have is the classification of cumulative robot collisions exceeding 5000N as "excessive collisions." In collaborative robotics [...]
>
> Thank you for providing the safety engineering reference for force safety thresholds around humans! We use the guidelines in this paper to perform additional evaluations below.
>
> We adopt the 5000N and 7500N collision force limits for the Pick and Place tasks from the original implementation of the HAB [1] and prior work studying the magical grasp HAB [2]. These particular values are used to help RL training and reward design, as lower collision requirements can impede RL training and hamper exploration.
>
> To address concerns on the realism of learned manipulation behaviors, we used our trajectory labeling system to compare performance of our  RL-Per, RL-All, and IL policies on varying low-value cumulative collision thresholds. We use the “safe for hip” human range of <=1400N from [3], and compare performance with the limit from [1, 2] and our work. The chart is available in the newest revision in Appendix A.4.4, Fig. 8.
>
> We find an approximately 5-20% decrease depending on the subtask when decreasing the cumulative force threshold. Interestingly, in all Pick tasks, and in PrepareGroceries Place (which involves placing in the Fridge), we find that the RL-Per policies perform better under lower collision thresholds than RL-All policies. The difference is less noticeable in TidyHouse Place and SetTable Place, which involve placing only in open receptacles and therefore involve fewer obstructions (e.g. dining table, counter, etc).
>
> While we use the trajectory filtering system to analyze the performance of our existing policies, future work can also use it to filter for collision-safe demonstrations when training their policy. Adjusting the collision thresholds in the environments is equally straightforward. We hope the dataset/data generation tools help future work improve robot safety in the context of low-level whole body control for home assistants.
>
> > While this undoubtedly makes it more efficient for researchers to work on this problem, the novelty is somewhat limited [...] It’s also worth noting that ManiSkill3 already includes a drawer-opening task with the Fetch robot out-of-the-box
>
> To our knowledge, we provide the first gpu-accelerated, home-scale, low-level whole body control environments for robotics which is fast enough to accommodate online training (e.g. RL), in addition to our provided rewards hand-engineered for the Fetch embodiment, the dataset, baseline policies, and trajectory filtering system. While a significant portion of the novelty of our work comes from engineering accomplishments, we believe such engineering work is important for advancing robot learning research.
>
> Additionally, while ManiSkill3 does include a mobile manipulation task through OpenCabinetDrawer, ManiSkill-HAB’s environments support multiple subtasks (Pick, Place, Open Close), apartment-scale scenes, and more randomization/diversity. Furthermore, the OpenCabinetDrawer baseline uses state, and does not provide a vision-based baseline.
>
> > exploring how these tasks can be integrated to mitigate hand-off issues between independent modules could add significant value
>
> While teleporting the robot within 2m of the target goal (with noise) does simplify the task somewhat by removing error from failed navigation, we note that mobile base navigation with realistic grasping is not particularly different from navigation with magical grasp. Hence, the handoff challenges in navigation under low-level control are not notably different from prior work [1,2].
>
> Furthermore, to ensure successful handoff between manipulation skills, we impose additional requirements in our subtasks to ensure overlap in initial/terminal state distributions (terminal arm joint position and velocity requirements), new rewards hand-made for the Fetch embodiment, sampling Pick grasp poses for initializing Place training, etc) which are not used in prior work. We also discuss issues with cluttered grasping and temporal dependencies which can affect skill chaining in Section 6.1 and Table 3 (e.g. PrepareGroceries Pick sees a large decrease in subtask success rate for the second Pick Fridge subtask due to disturbances caused by the first Pick Fridge subtask).
>
> ---
>
> Thank you again for your notes and feedback! We hope our explanations and additional results are able to address your questions and concerns. If not, please let us know, and we are happy to discuss further.
>
> [1] Szot, Andrew et al. “Habitat 2.0: Training home assistants to rearrange their habitat.” NeurIPS 2021
>
> [2] Gu, Jiayuan et al. “Multi-skill Mobile Manipulation for Object Rearrangement.” ICLR 2023
>
> [3] Mewes, Detlef and Mauser, Fritz. “Safeguarding crushing points by limitation of forces.”

---

> > ### Author Response · Authors · 2024-11-20
> > **Response to Reviewer HBjR: SAC vs PPO experiments added**
> >
> > Dear reviewer HBjR,
> >
> > Our SAC vs PPO experiments have concluded, and have been added to Appendix A.4.3 and Fig. 6. In Pick/Place, we find SAC significantly outperforms PPO, while for Open/Close PPO and SAC achieve similar performance (in some cases PPO performs marginally better, and vice-versa, likely because we do not randomize fridge/drawer geometry, only spawn locations). So, we use SAC for Pick/Place due to superior performance, and we use PPO for Open/Close due to faster wall-time training with similar performance.
> >
> > We sincerely thank you for your constructive feedback! Please let us know if this experiment combined with the previous discussion have addressed your concerns; we are happy to discuss further.

---

> > > ### Comment · Reviewer_HBjR · 2024-11-24
> > > **Thank you for your reply**
> > >
> > > I want to thank the authors for the detailed reply to my questions and for already incorperating my feedback into their revised version. While my remaining questions have been addressed, my main concern is the novelty in the contribution. I think that having evidence about both the simulation performance, which is already very strong, as well as the realism through validation against a real robotic system, would add tremendous value to this work and make it a very valuable contribution to the research community. I will maintain my score.

---

> > > > ### Author Response · Authors · 2024-11-25
> > > >
> > > > We thank the reviewer for their early engagement with our work, and we are glad that questions and concerns have been resolved. The reviewer's feedback on our manuscript has been invaluable, helping us improve the manuscript.
> > > >
> > > > We find through our baselines that our apartment-scale, low-level, whole-body control tasks are very challenging, and have not yet been solved by our baselines. We agree that real-world transfer is an interesting avenue for future research; we hope our environments, baselines, checkpoints, dataset, and trajectory labeling/filtering tools enable the community to develop methods with superior performance on this task. Hence, we leave performance improvements and real-world transfer to future work.
> > > >
> > > > We thank the reviewer again for dedicating their valuable time and effort towards evaluating our manuscript.

---

### Official Review · Reviewer_KUnv · 2024-11-03

**Soundness:** 3
**Presentation:** 3
**Contribution:** 2
**Rating:** 5
**Confidence:** 4

**Summary:**

This paper presents MS-HAB, a benchmark for low-level manipulation and in-home object rearrangement aimed at supporting embodied AI research. The benchmark features a GPU-accelerated Home Assistant Benchmark (HAB), which the authors claim achieves over three times the speed of previous implementations, and provides extensive reinforcement learning (RL) and imitation learning (IL) baselines for future comparisons. Additionally, a rule-based trajectory filtering system has been developed to select demonstrations that meet specific behavior and safety criteria. Ultimately, this enhances simulation efficiency, and the authors hope their work will support scalable data generation for robotics research.

**Strengths:**

1. The benchmark provides a holistic framework for home-scale rearrangement, featuring a GPU-accelerated implementation of the Home Assistant Benchmark (HAB) that supports realistic low-level control, enabling more effective manipulation tasks.
2. It includes extensive RL and IL baselines, allowing researchers to compare their methods against established standards and fostering advancements in these areas.
3. The systematic evaluation approach, utilizing a trajectory labeling system, enhances the benchmark's reliability and provides detailed insights into performance.
4. The implementation of demonstration filtering allows for efficient and controlled data generation at scale, which is crucial for developing robust robotic systems.

**Weaknesses:**

1. There is significant room for improvement in the existing baselines, as the current performance may not meet the highest standards set by more advanced methods in the field.
2. The authors do not claim that the benchmark supports transfer to real robots, indicating that there are still challenges to be addressed in applying these methods in practical scenarios.
3. In fact, most of the ideas and techniques used in this paper have appeared in prior work, particularly the M3 framework proposed by Gu et al. [1]. Both the skill sequence partitioning for three long-horizon tasks and the training algorithms for various skills are fundamentally consistent with M3. However, the authors do not provide a detailed comparison with it in the paper.
4. The technical contribution is quite limited. The simulation environment, baseline algorithms, and even the subtask definitions used in the paper have already been proposed in previous work. This makes the draft somewhat more like a technical report.

[1] Gu, Jiayuan, et al. "Multi-skill Mobile Manipulation for Object Rearrangement." The Eleventh International Conference on Learning Representations.

**Questions:**

1. Why does the paper's baseline training algorithm not compare with the methods proposed in M3? M3 achieves a much higher task completion rate for the three tasks than the performance presented in this paper.
2. Does the per-object policy refer to training a specific pick or place policy for each different object (e.g., bowls, cans, cups, etc.)? While this might improve completion rates for certain tasks, could it significantly limit the generalization capability for complete long-horizon tasks?

---

> ### Author Response · Authors · 2024-11-17
> **Response to Reviewer KUnv [1/2]**
>
> Thank you for your constructive feedback and notes! We address the comments and questions below:
>
> > Question 1: Comparison with M3
>
> Thank you for pointing this out. The crucial difference between M3 (and other prior work) and ManiSkill-HAB is that M3 relies on magical grasping, whereas the ManiSkill-HAB benchmark requires realistic low-level control. Because M3 uses magical grasp, it is able to achieve high success rates with only on-policy RL (for example, in cluttered settings, M3 can simply hover the end-effector over the clutter close to the target, and magical grasp will teleport the object into the gripper, notably reducing the difficulty of cluttered grasping).
>
> Similar to M3, we use mobile manipulation subtask formulations for improved composability and skill chaining. However, different from M3, we make the following additions for low-level control:
>
> 1. We found the manipulation rewards used by M3 were insufficient for learning low-level grasping policies. So, we provide new dense rewards designed for mobile manipulation with the Fetch embodiment. In particular, we significantly regularize robot behavior depending on the stage the policy has reached in the subtask (e.g. penalties for joint positions far from a predefined resting position for Fetch, end-effector velocity, joint/mobile base velocity when object is grasped, etc) and tune collision rewards for low-level control, while maintaining similar task-related rewards as M3. These rewards are available directly in our environments for other researchers to take advantage.
>
> 2. M3 only trains online RL and is able to achieve higher success rates thanks to magical grasp, and does not attempt IL with its subtask formulation. However, we find pure online RL insufficient to solve our low-level control tasks, hence we also provide a dataset/data generation tools with trajectory filtering, IL baselines, and ablations to analyze the impact of different trajectory filters.
>
> 3. Finally, there are a variety of smaller additions to subtask success conditions and training necessary for low-level control:
>
>      a. We add terminal joint position and velocity requirements to ensure the robot learns to stably grasp and hold objects from above
>
>      b. We add collision requirements for Open and Close (not used in M3), since our policies must interact with the handles, unlike M3’s policies.
>
>      c. When training Place, we sample grasp poses from our Pick policy to ensure successful handoff (since grasp pose selection is non-trivial)
>
>      d. When training Open Drawer, we find the small handle is difficult for low-level grasp, so we perturb the initial state distribution by randomly opening the drawer 20% of the way 10% of time during training (but not during evaluation)
>
> To visually demonstrate the difference in difficulty and end-product of our baselines and M3, we have added a section to the supplementary comparing examples of cluttered grasping with the Cracker Box.
>
> While we add some discussion about M3 (and other skill chaining works) in Sec. 2, if reviewer KUnv finds the above discussion would aid in clarity, we are happy to include it in the manuscript! Please let us know.
>
> > Question 2: Per vs all-object generalization to complete long-horizon tasks
>
> Good question – to evaluate generalization, we have added a comparison in long-horizon task completion rates in both train and validation splits to Appendix A.4.5. The validation split involves apartment layouts and configurations unseen in training, so it is a good measure of generalization.
>
> We find that per-object policies demonstrate improved performance on full long-horizon tasks in both train and validation splits, indicating that per-object policies improve the generalization capability for complete long-horizon tasks.
>
> > There is significant room for improvement in the existing baselines
>
> Regarding baseline performance, we believe the room for improvement over our RL and IL baselines indicates that our benchmark is not saturated yet. This can be attributed to the requirement of whole-body control, additional randomization, and vision-based data. Previous benchmarks like RoboCasa use stationary manipulation for their datasets; whole-body control is important for allowing the policies to reposition themselves to avoid collisions, improve grasping in cluttered receptacles, and work in situations with tighter spaces/tolerances (thin hallways, manipulation in fridge). A good example of this is the Close (Fridge) video in the supplementary. We also have more scene-level randomization (object positions, locations, etc) thanks to the HAB, while e.g. RoboCasa has more textures and objects. Finally, using vision to infer collisions and obstructions while the cameras are moving due to the mobile base can add difficulty.
>
> Regarding baselines for other methods, per reviewer request, we are working on baselines for more methods. We will notify reviewers when these baselines are added.

---

> > ### Author Response · Authors · 2024-11-17
> > **Response to Reviewer KUnv [2/2]**
> >
> > > The technical contribution is quite limited. The simulation environment, baseline algorithms, and even the subtask definitions used in the paper have already been proposed in previous work
> >
> > Regarding our simulation environments, to our knowledge we provide the first gpu-accelerated, home-scale, low-level whole body control environments for robotics which are fast enough to accommodate online training (e.g. RL). Other home-scale, low-level control benchmarks like Behavior-1k and RoboCasa exist, but these environments generally run at real-time speed, hence are only usable for IL research or policy evaluation.
> >
> > Second, as discussed above, while we build on M3’s mobile manipulation subtask formulations, we add data generation with trajectory filtering to control robot behavior, a large vision-based robot dataset, and IL baselines, which are not done in prior work like [1] or [2], along with new rewards and subtask alterations necessary for low-level control.
> >
> > ---
> >
> > Thank you again for your valuable feedback! We hope we are able to address your concerns. If not, please let us know, and we would be happy to discuss details further.
> >
> > [1] Szot, Andrew et al. “Habitat 2.0: Training home assistants to rearrange their habitat.” NeurIPS 2021
> >
> > [2] Gu, Jiayuan et al. “Multi-skill Mobile Manipulation for Object Rearrangement.” ICLR 2023

---

> > ### Comment · Reviewer_KUnv · 2024-11-25
> >
> > Thank you for the response. While the authors have provided detailed explanations addressing my concerns, my main issue with ManiSkill-HAB remains its relatively limited technical contribution, as there are already highly competitive works in this field. I suggest the authors consider the feedback from other reviewers to further improve the quality of the paper and enhance its contribution. For instance, they could propose a more effective baseline method based on ManiSkill-HAB, rather than simply running existing RL or IL algorithms. Alternatively, they could improve the rendering realism of the simulation and validate the sim-to-real capability and training acceleration advantages using real robots. This would better highlight the unique contributions and value of ManiSkill-HAB compared to existing works. For these reasons, I will maintain my current score.

---

> > > ### Author Response · Authors · 2024-11-26
> > > **Response to Reviewer KUnv**
> > >
> > > We thank the reviewer for their continued engagement with our manuscript, and we are glad we were able to appropriately address your prior questions. We appreciate your feedback on how we can strengthen our manuscript and competitiveness with other work, and have made additions to incorporate this feedback below:
> > >
> > > > they could improve the rendering realism of the simulation
> > >
> > > Thank you for this feedback. To improve rendering realism, we have provided a live-rendered ray-tracing option with custom-tuned lighting which users can enable with just one line.
> > >
> > > To compare ray-tracing performance, we have conducted a new benchmark on ray tracing render performance between ManiSkill-HAB and Behavior-1k in Appendix A.5. Using the same GPU (Nvidia RTX 3070), ManiSkill-HAB is 3.88x faster than Behavior-1k while using 32.72% less GPU memory.
> > >
> > > To compare quality, we have added live-rendered comparison images to Appendix A.5 and the supplementary website (https://sites.google.com/view/maniskill-hab#h.m9iw44afaks1). The main difference in rendering fidelity is the choice of assets; one can use higher-quality textures for an even more realistic render if necessary, which we leave to future work.
> > >
> > > > main issue with ManiSkill-HAB remains its relatively limited technical contribution, as there are already highly competitive works in this field
> > >
> > > Thank you for raising this point: below we list specific benefits of ManiSkill-HAB which differentiates our work from Behavior-1k and RoboCasa.
> > >
> > > **Behavior-1k**
> > > - **Baselines, Dataset, Rewards**: Behavior-1k currently does not have demonstration datasets, strong baselines, etc. Meanwhile, we provide all of these in addition to our trajectory labelling and filtering system.
> > > - **Speed and Render Quality**: With our added tuned ray-traced lighting, we are able to achieve similarly high-quality rendering while also running 3.88x faster and using 32.72% less gpu memory.
> > >
> > > **RoboCasa**
> > > - **Speed**: In our benchmarks, we achieve ~4000SPS while rendering 2 cameras and interacting with multiple dynamic objects on a 4090 GPU. Meanwhile RoboCasa reports only 31.9 SPS *without rendering* on an A5000. While this is not a rigorous benchmark, this difference in speed means our RL baselines, extensive evaluation (Appendix A.6.3), and customizable data generation would be intractable on other platforms.
> > > - **Whole-Body Control**: The RoboCasa demonstration dataset *does not include whole-body control*; rather it exclusively contains stationary manipulation demonstrations, while navigation is handled separately. Meanwhile, our dataset, baselines, and tasks all require true whole-body control, which is much harder (for instance, the policy must learn how the camera poses change as the robot moves).
> > >
> > > In summary, we differentiate ourselves with significantly faster environment speed than alternatives without the loss of physical or visual realism and true whole-body control.
> > >
> > > Our baselines and dataset generation methods match these core contributions: we use our fast environments to train online RL policies, generate massive datasets with custom filtering, and evaluate RL and IL policies across billions of samples using our filtering system for detailed analysis, all of which are not feasible with alternative platforms in this field due to slow simulation speed.
> > >
> > > > they could propose a more effective baseline method based on ManiSkill-HAB, rather than simply running existing RL or IL algorithms
> > >
> > > We would like to reiterate that the main contribution of the work is to provide baselines and datasets for low-level whole-body control, rather than present a new method or algorithm.
> > >
> > > However, to expand our available baselines, we have added a Diffusion Policy baseline to Appendix A.4.6. We use a traditional UNet backbone with a DDPM scheduler for diffusion. Due to time limitations, we are unable to tune these baselines significantly; however, our results indicate that larger backbone networks, hyperparameter tuning per-subtask, or online finetuning (e.g. DPPO) might be necessary for our difficult tasks.
> > >
> > > ---
> > >
> > > We thank the reviewer again for their valuable feedback, which has helped us improve our manuscript and competitiveness with other works.
> > >
> > > Given the extended discussion period, we would greatly appreciate it if the reviewer would consider revising our scores if you find it appropriate, and we are happy to address any additional remaining concerns. We are grateful for your service to the community.

---

> > > > ### Author Response · Authors · 2024-12-01
> > > > **Follow-Up Request [Deadline Approaching]**
> > > >
> > > > Dear Reviewer,
> > > >
> > > > We thank you once again for your time and efforts in reviewing our work and providing feedback on our manuscript. As the extended discussion deadline (Dec 2) is rapidly approaching, this is a gentle reminder to let us know if we have satisfactorily addressed the reviewer's concerns — in particular rendering realism and contributions compared to other platforms — and to revise our scores if you find it appropriate. We are happy to address any additional remaining concerns. We are grateful for your service to the community.
> > > >
> > > > Regards,
> > > >
> > > > Authors

---

### Official Review · Reviewer_W29p · 2024-11-03

**Soundness:** 3
**Presentation:** 3
**Contribution:** 3
**Rating:** 5
**Confidence:** 4

**Summary:**

The paper updates HAB benchmark to MS-HAB for GPU-based simulation and larger-scale evaluation. Especially, it adopts Maniskill3 (Sapien) as the backend instead of the original Habitat (Unity) backend. It is worth mentioning that the evaluation is now end-to-end in the physics engine and there is no magical grasping like in Habitat. Furthermore, it conducts more experiments on reinforcement learning / imitation learning methods for long-horizon manipulation tasks, and show the challenge of the benchmark.

**Strengths:**

1. The paper is well-written and easy to understand.
2. The updates are necessary for speed and physical fidelity.
3. The tool is widely useful, especially for long-horizon manipulation tasks like home-scale rearrangment challenge.
4. The baselines are more comprehensive in this paper comparing to previous papers, which provides a large amount of insight into the benchmark.

**Weaknesses:**

1. The novelty of the benchmark needs discussion. Especially when RoboCasa, Behavior-1k platforms are already published for a while (these are non-current work). In terms of the scale / scene diversity / task difficulty, why MS-HAB is comparable or is different with these existing pipelines.
2. The imitation dataset is generated with RL. Howerver, one can also generate (for pick and place) with sense-plan-act pipeline. Why not generate dataset with SPA pipeline?
3. It will be great if one provide baseline results for Senes-Plan-Act (TAMP), and with oracle information. This will provide valuable insights.

**Questions:**

1. Comparison with other benchmarks.
2. SPA pipeline for IL dataset.
3. More baseline comparison.

---

> ### Author Response · Authors · 2024-11-17
> **Response to Reviewer W29p [1/2]**
>
> Thank you for your feedback and questions! We address your questions and concerns below:
>
> > Question 1: Comparison with other benchmarks
>
> Good question – MS-HAB differentiates itself in environment speed and data generation/baseline training methodology.
> We provide detailed comparison between ManiSkill-HAB with RoboCasa and Behavior-1k below, and comparisons with other simulators/benchmarks are available in Sec. 2.
>
> **Speed**: MS-HAB provides fast environments with realistic physics for online training and scalable data generation, while RoboCasa and Behavior-1k sacrifice speed for enhanced realism. While the below numbers are not a rigorous comparison, they provide a general idea of speed differences between platforms:
>
> RoboCasa reports 31.9 FPS without rendering. Meanwhile, in our benchmark we render 2 128x128 RGB-D sensor images while actively colliding with multiple objects, and we achieve ~4000 FPS at ~24 GB VRAM usage with 1024 envs, and ~2500 FPS at ~5GB vram usage with 128 envs, all on a single GPU.
>
> Behavior-1k reports 60fps while rendering with ray-tracing. From their benchmark script, it seems they render 1 128x128 RGB-D camera. We run our envs with ray-tracing while rendering 2 128x128 RGB-D cameras to achieve 204.86 ± 11.73 FPS (95% CI over 10 runs), which is notably faster than Behavior-1k with double the cameras.
>
> Importantly, RoboCasa and Behavior-1k have improved visual fidelity compared to MS-HAB, and include additional features like AI-generated textures in RoboCasa and complex scene interactions in Behavior-1k. However, these features add complexity to simulation, and as a result these simulators run at approximately real-time speed, relegating their usage to IL research or evaluation purposes. At these speeds, online training or very extensive evaluation (e.g. our success/failure mode statistics in Appendix A.5.3 are evaluated over hundreds of thousands of episodes) is intractable.
>
> Meanwhile, in MS-HAB, our environment speed is key to our other contributions. By focusing on fast simulation with realistic physics, we are able to feasibly train policies with online RL, extensively evaluate policies over many hundreds of thousands of episodes with our trajectory labeling system (Appendix A.5.3), and generate data much faster than RoboCasa or Behavior-1k.
>
> **Dataset and Baselines**: Behavior-1k reports working on baselines and datasets, but these are not released yet.
>
> RoboCasa uses teleoperated demonstrations + MimicGen to create a scalable dataset. However, trajectories in this dataset are limited to motions seen in the teleoperated demonstrations, and there is no method for filtering demonstrations by robot behavior.
>
> Meanwhile, our scalable dataset is generated by running our policy in our fast environments + trajectory filtering. Our policies show good robustness to unseen layouts and configurations, allowing greater diversity in environment configurations we can use when generating data (Table 1). Furthermore, using our trajectory filtering system, users can select for specific desirable robot behaviors (which can be used to influence policy behavior, per Sec 6.2.2).
>
> Finally, our mobile manipulation policies perform true whole-body control (i.e. local navigation while manipulating), while RoboCasa demonstrations separate mobility and navigation.
>
> > Question 2: SPA pipeline for IL dataset
>
> We chose RL for data generation instead of SPA for a few reasons: first, Habitat 2.0 trained SPA baselines using magical grasp and found it was more brittle than RL, especially in cluttered settings or challenging receptacles. Since SPA already struggled with magical grasp, it is likely it would perform worse on our harder low-level control variants, hence we chose to focus on reward shaping and fast training for RL.
>
> Second, in our RL checkpoints, we save policy weight, optimizer states, and other relevant trainable parameters (e.g. log alpha for SAC). These checkpoints enable some methods which involve fine tuning by resuming training, such as [3].
>
> Finally, we find that given enough samples from our environment, RL can learn interesting emergent behaviors. We have added an example video to the supplementary website where the RL policy performs “pick → drop → pick from floor” learned purely from online sampling (even though we provide no examples of objects on the ground). These unique behaviors helped us to improve our trajectory filtering system to handle more edge cases, and might be useful for work in failure recovery, replanning, etc.
>
> > Question 3: More baseline Comparison
>
> Per reviewer request, we are working on baselines for more methods. We will notify reviewers when these baselines are added.
>
> ---
>
> We sincerely appreciate your questions and feedback! We hope we were able to appropriately address your questions. If not, please let us know, and we are happy to continue discussion.

---

> > ### Comment · Reviewer_W29p · 2024-11-24
> > **Thanks for the reply**
> >
> > I will maintain my score.
> >
> > Despite that MS-HAB has higher speed, rendering fidelity is indeed important for sim2real transfer in the long run. I am afraid that if the problem is not addressed. Users will still prefer Behavior-1k as the backend.
> >
> > Training RL for each scene x object is cumbersome. SPA is more data efficient or MimicGen + human demo. I believe for data generation purpose, having these pipelines are as crucial as training RL policies.

---

> > > ### Author Response · Authors · 2024-11-24
> > >
> > > We are thankful to the reviewer for the prompt response and insightful comments — the remaining points regarding visual fidelity and data generation are important, and we address them below:
> > >
> > > > rendering fidelity is indeed important for sim2real transfer in the long run [...] Users will still prefer Behavior-1k as the backend
> > >
> > > To compare rendering quality, we have added a comparison of live-rendered ray-traced images between ManiSkill-HAB and Behavior-1k to our supplementary (https://sites.google.com/view/maniskill-hab#h.m9iw44afaks1). Our ray-traced live rendering is 3.5x faster than Behavior-1k while maintaining similar render quality, and our ray-tracing can be turned on with a one-line change in the code. The main difference in rendering fidelity is the choice of assets; one can use higher-quality textures for an even more realistic render if necessary, which we leave to future work.
> > >
> > > We additionally point out that Behavior-1k does not have baselines, tuned dense rewards, or demonstration datasets, which makes it difficult for users to train policies (especially for the difficult whole-body, low-level control skills in ManiSkill-HAB).
> > >
> > > > Training RL for each scene x object is cumbersome. SPA is more data efficient or MimicGen + human demo
> > >
> > > Thank you for noting this! We were able to leverage our memory-efficient environments (Fig. 1) to significantly lessen the burden of training many per-object policies. In particular, we ran more training runs in parallel on lower-end hardware (e.g. GPUs with less VRAM) or multiple runs per system on better hardware, which would not be possible with prior implementations.
> > >
> > > We agree that SPA and MimicGen pipelines are also crucial to embodied AI and robot learning. Our specific choice to use RL was spurred by prior work in the HAB focusing on RL [1, 2] and the very fast wall-time training exhibited by RL in core ManiSkill3 tasks [3]. Our work is orthogonal to approaches leveraging SPA / MimicGen pipelines and can provide a platform for RL-focused approaches (in addition to other LfD methods through our dataset, e.g. IL).
> > >
> > > ---
> > >
> > > Thank you again for your engagement with our work and dedicating your valuable time and effort towards evaluating our manuscript! Please let us know if we have been able to address your concerns; if not, we are happy to add further clarifications.
> > >
> > > [1] Szot, Andrew et al. “Habitat 2.0: Training home assistants to rearrange their habitat.” NeurIPS 2021
> > >
> > > [2] Gu, Jiayuan et al. “Multi-skill Mobile Manipulation for Object Rearrangement.” ICLR 2023
> > >
> > > [3] Tao, Stone et al. “ManiSkill3: GPU Parallelized Robotics Simulation and Rendering for Generalizable Embodied AI.” Preprint, arXiv

---

> > > > ### Author Response · Authors · 2024-12-01
> > > > **Follow-Up Request [Deadline Approaching]**
> > > >
> > > > Dear Reviewer,
> > > >
> > > > We thank you once again for your time and efforts in reviewing our work and providing feedback on our manuscript. As the extended discussion deadline (Dec 2) is rapidly approaching, this is a gentle reminder to let us know if we have satisfactorily addressed the reviewer's concerns — in particular regarding rendering realism and our training/dataset pipeline — and to revise our scores if you find it appropriate. We are happy to address any additional remaining concerns. We are grateful for your service to the community.
> > > >
> > > > Regards,
> > > >
> > > > Authors

---

> ### Author Response · Authors · 2024-11-17
> **Response to Reviewer W29p [2/2]**
>
> [1] Li, Chengshu et al. “BEHAVIOR-1K: A Human-Centered, Embodied AI Benchmark with 1,000 Everyday Activities and Realistic Simulation.” CoRL 2022
>
> [2] Nasiriany, Soroush et al. “RoboCasa: Large-Scale Simulation of Everyday Tasks for Generalist Robots”
>
> [3] Jia, Zhiwei, et al. “Improving Policy Optimization with Generalist-Specialist Learning.” ICML 2022
>
> [4] Szot, Andrew et al. “Habitat 2.0: Training home assistants to rearrange their habitat.” NeurIPS 2021
>
> [5] Gu, Jiayuan et al. “Multi-skill Mobile Manipulation for Object Rearrangement.” ICLR 2023

---

### Official Review · Reviewer_XeS8 · 2024-11-03

**Soundness:** 4
**Presentation:** 4
**Contribution:** 3
**Rating:** 8
**Confidence:** 5

**Summary:**

The paper introduces **ManiSkill-HAB**, a comprehensive benchmark for low-level manipulation and in-home object rearrangement tasks. It aims to enhance the Home Assistant Benchmark (HAB) with a GPU-accelerated implementation for improved simulation speed and realism. The authors included extensive reinforcement learning (RL) and imitation learning (IL) baselines and developed an automated rule-based filtering system for generating demonstrations that are compliant with predefined safety and behavior rules.

Together, this benchmark provides:
1. fast, realistic, diverse simulation tasks and environments for home scale manipulation challenges
2. support for low-level control that enables realistic grasping and interaction
3. extensive RL and IL baselines
4. vision-based robot dataset at scale

I recommend acceptance for this paper since it provides clear and important contributions to facilitate the advancement of in-home manipulation and embodied AI research.

**Strengths:**

Originality
- While there have been benchmarks for robotic manipulation and household-related task manipulation, this work focuses on tasks and objects prevalent in home assistant settings.
- Environments in this work are GPU-accelerated and significantly outperform prior works in terms of speed and computational costs.
- While filtering trajectories with privileged information from the simulator is not unseen, the ability to scale up the simulation and rendering at a faster speed makes sampling and filtering more trajectories feasible.

Quality:
- From the supplementary videos, the simulation environments and rollouts appear to be high quality.
- The comparisons to prior work show a clear advantage of the proposed method in simulation speed.
- The RL & IL baseline methods are extensively studied using this benchmark, providing future research good baselines.

Clarity:
- The writing, figures, and supplementary materials are well-presented and easy to follow.
- The evaluation protocols for the baseline methods are structured and presented clearly.
- The authors also included failure modes for each task in the supplementary material.

Significance:
- The benchmark attempts to address a critical need in the robotics community for more efficient and realistic simulation tools that can keep pace with the increasing expectation of robots performing complex tasks in daily environments.
- The potential impact on future research, particularly in home rearrangement tasks, is significant, providing a robust platform for developing and testing new algorithms and approaches.

**Weaknesses:**

1. Currently, this work consists of three long-horizon tasks: TidyHouse, PrepareGroceries, and SetTable. For future iterations, it would be beneficial to expand the tasks and manipulated objects beyond HAB and YCB datasets. Potential tasks could include cleaning dishes, laundry tasks, and tool usage.

2. The RL and IL baselines include SAC, PPO, and BC. It would be greatly beneficial to the research community to have more recent baseline methods such as TD-MPC2, ACT, Diffusion Policies, etc.

**Questions:**

1. Simulation limitations:
What are the limitations of simulation technologies used in this work? Would it be possible to simulate deformable objects, fluids, or more intricate rigid objects such as tools? Are there plans to expand the manipulation tasks beyond pick + place and open + close?

2. Real-World Application:
What are the anticipated challenges in transferring the learned behaviors from the simulated MS-HAB environment to real-world robots, particularly in unstructured environments like typical homes?

For future works, it would be interesting to study how methods that solve this benchmark at X% success rate would transfer to real world robotics.

---

> ### Author Response · Authors · 2024-11-17
> **Response to Reviewer XeS8**
>
> We are greatly thankful for your acknowledgement of the quality of our work! We answer questions below:
>
> > Question 1: Simulation limitations
>
> The foremost limitation is that, in exchange for high speed, ManiSkill primarily supports rigid-body physics. So, deformables and fluids are not yet supported. However, it is possible to support more intricate objects like tools, for example those from the YCB dataset which can be imported easily.
>
> Furthermore, since MS-HAB is open source, we will continue adding performance improvements, features, and tools requested by the community.
>
> > Question 2: Real-World Application
>
> The first anticipated difficulty is sim2real. Our policies use depth images, which are easier to transfer to the real world, and our observation components can be replicated with onboard sensing and state estimation. However, even with simulators like ManiSkill3 which support realistic control, often extensive domain randomizations or data augmentation is needed for zero-shot transfer to the real world. Domain randomization features (e.g. camera poses, controller parameters, etc) can be added in the future.
>
> The second anticipated difficulty is scene diversity. While our policies do show good transfer to unseen apartment layouts and configurations, real-world unstructured environments are constantly changing, including rearrangement of objects, additional mess, and more. To this end, MS-HAB supports user-generated scene configs for additional randomization, and we are looking into integrating our fast environments with other scene datasets for added diversity.
>
> ---
>
> Thank you again for your feedback and questions! We agree that real-world transfer is an interesting future avenue for research, and we hope that our work will aid the research community in developing methods and tools for realizing this goal. If you have any further questions, please let us know and we are happy to discuss!

---

> > ### Comment · Reviewer_XeS8 · 2024-11-25
> > **Response to Authors**
> >
> > Thank you for the replies to my questions. Looking forward to the integration of these new features and diverse capabilities.
> >
> > One additional improvement could be made, as mentioned in the initial review, "The RL and IL baselines include SAC, PPO, and BC. It would be greatly beneficial to the research community to have more recent baseline methods such as TD-MPC2, ACT, Diffusion Policies, etc."

---

> > > ### Author Response · Authors · 2024-11-26
> > > **Response to Reviewer XeS8**
> > >
> > > Thank you for the suggestion! We have added a Diffusion Policy (DP) baseline in Appendix A.4.6. Due to limited time, we are unable to tune baselines significantly and we maintain architecture and hyperparameters across subtasks. Our results indicate that likely larger/different backbones, hyperparameter tuning per-subtask, or online finetuning methods (e.g. DPPO) are required to achieve a high success rate on our difficult tasks.
> > >
> > > We additionally attempted to train TD-MPC2 to add a model-based online RL baseline, however due to slower update times, we were unable to reach satisfactory performance on baselines for all our tasks in the time provided (the original paper limits training to 12M samples at the highest end, which is much less than our tasks require).
> > >
> > > That being said, similar to ManiSkill3, we plan to continue adding baselines over time to provide the community with more points of comparison.
> > >
> > > ---
> > >
> > > We thank you again for your invaluable feedback which has helped us strengthen our manuscript, and we are grateful for your service to the community. Please let us know if you have any further questions or concerns!

---

### Author Response · Authors · 2024-11-26
**Updates to Improve Visual Fidelity and Add Diffusion Policy Baseline**

We thank the reviewers for their valuable feedback on our manuscript. To address concerns regarding visual fidelity and additional baselines, we have made two additions to the manuscript to strengthen our work based on reviewer feedback.

**1. Improved Visual Fidelity with Tuned Ray-Tracing**

In order to improve rendering realism, we have provided a live-rendered ray-tracing option with custom-tuned lighting (HDRI/env maps, optix denoiser, samples per pixel, the number and type of lights, etc) tuned for visual realism and speed. Users can enable this with just one line in the code.

To rigorously compare ray-tracing performance with other offerings, we have conducted a new benchmark on ray tracing render performance between ManiSkill-HAB and Behavior-1k in Appendix A.5. Using the same GPU (Nvidia RTX 3070), ManiSkill-HAB is 3.88x faster than Behavior-1k while using 32.72% less GPU memory.

To compare visual quality, we have added live-rendered comparison images to Appendix A.5 and the supplementary website (https://sites.google.com/view/maniskill-hab#h.m9iw44afaks1). As seen in these images, the rendering quality (lighting, clarity, etc) have similar quality to Behavior-1k. Users can also use higher-quality textures for an even more realistic render if necessary, which we leave to future work.

**2. Diffusion Policy Baseline**

To expand our provided baselines, we have provided a Diffusion Policy (DP) baseline in Appendix A.4.6. Due to limited time, we keep the same architecture and hyperparameters across tasks. We find that, while DP is known for smooth trajectories, for our difficult tasks, likely larger/different backbones (e.g. diffusion transformers [1]), hyperparameter tuning per-subtask, or online finetuning methods (e.g DPPO [2]) are required.

We attempted TDMPC-2 to add a model-based RL baseline, however due to the slower update times of the original codebase, we were unable to achieve satisfactory results in the given time period. However, similar to ManiSkill3, we will continue adding more baselines as time goes on, and as the community requests them.

---

We would like to thank the reviewers for their suggestions and feedback throughout the review process. We hope these additions address concerns related to realism and baseline diversity. If any concerns remain, we are happy to discuss further!

[1] Dasari, Sudeep et al. “The Ingredients for Robotic Diffusion Transformers”. Preprint, arXiv

[2] Ren, Allen, et.al “Diffusion Policy Policy Optimization". Preprint, arXiv

---

### Author Response · Authors · 2024-11-27
**Summary of revisions and new experiments to author feedback**

As the deadline for manuscript changes is today, we summarize our text revisions below. We look forward to continued discussion with the reviewers on remaining concerns until the discussion deadline on Dec 2.

---

**Ray Tracing/Visual Fidelity, Appendix A.5**: Reviewer W29p and KUnv make a good point about the importance of rendering fidelity. To address this, we provide a live-rendered ray-tracing option with tuned lighting which users can enable with a one-line change in the code. We benchmark performance and compare render quality with Behavior-1k: MS-HAB renders with ray-tracing 3.88x faster while using 32.73% less memory, all with similar ray-tracing render quality as Behavior-1k, as seen in Fig. 10 and our supplementary website (https://sites.google.com/view/maniskill-hab#h.m9iw44afaks1).

**Diffusion Policy Baselines, Appendix A.4.6**: Reviewer XeS8 and KUnv note that additional baselines will be helpful to the community. To address this, we run Diffusion Policy baselines for each task/subtask. While we are unable to tune our baselines significantly due to time limitations, our results indicate that different/larger backbones (e.g. diffusion transformer [1]), additional tuning, or online finetuning (e.g. DPPO [2]) may be needed for our difficult tasks.

**Per vs All-Object Long-Horizon Performance, Appendix A.4.5**: Per request of reviewer KUnv, we compare RL-All and RL-Per policy performance in long horizon tasks, and we find that RL-Per policies indeed perform better.

**Eval Low Collision Thresholds, Appendix A.4.4**: Reviewer HBjR provides important feedback about performance under industry-standard collision safety thresholds. To address this, we evaluate our policies for the Pick/Place subtasks across low collision thresholds, finding that while there is a 5-20% decrease in performance depending on subtask, our learned manipulation behaviors retain reasonable performance.

**SAC vs PPO, Appendix A.4.3**: Reviewer HBjR raises a good point about our choice of SAC vs PPO for Pick/Place and Open/Close subtasks respectively. To address this, we compare SAC and PPO across all tasks/subtasks. Based on these results, we use SAC for Pick/Place due to significantly better performance, while we use PPO for Open/Close due to comparable performance with faster wall-time training.

**Minor rewording, main text**: Minor rewording, add note that we use frame stack to handle partial observability.

---

We thank the reviewers for their feedback on our manuscript which has helped us improve the manuscript. We hope these changes address remaining questions and concerns (i.e. rendering realism and baselines). If any questions and concerns remain, we are happy to continue discussion through the extended discussion period.

[1] Dasari, Sudeep et al. “The Ingredients for Robotic Diffusion Transformers”. Preprint, arXiv

[2] Ren, Allen, et.al “Diffusion Policy Policy Optimization”. Preprint, arXiv

---

### Meta-Review · Area_Chair_48No · 2024-12-21

**Metareview:**

This paper presents ManiSkill-HAB, a benchmark for low-level manipulation in home rearrangement tasks. The main contributions are: (1) a GPU-accelerated implementation of the Home Assistant Benchmark that achieves over 3x speedup while maintaining similar GPU memory usage, (2) comprehensive RL and IL baselines for manipulation tasks, and (3) a systematic trajectory filtering system for controlled data generation. The work demonstrates significant performance improvements over previous implementations while supporting realistic low-level control instead of "magical grasping".

The discussion period revealed several key concerns:
- Technical Novelty: Multiple reviewers (KUnv, HBjR, W29p) questioned the technical novelty, noting similarities to existing platforms. The authors clarified their key differentiators: significantly faster simulation speed (4000 FPS vs RoboCasa's 31.9 FPS), support for whole-body control vs stationary manipulation, and extensive baselines/datasets unavailable in platforms like Behavior-1k.
- Visual Fidelity: Reviewers W29p and KUnv raised concerns about rendering quality compared to alternatives. The authors added ray-tracing capabilities with benchmarks showing 3.88x faster performance than Behavior-1k while using 32.73% less GPU memory and maintaining comparable visual quality.
- Baseline Coverage: Reviewers requested more baseline comparisons.

During the rebuttal process, the authors added:
- Diffusion Policy baselines
- SAC vs PPO comparisons across all tasks
- Low collision threshold evaluations (1400N) to assess real-world safety
- Per-object vs all-object policy comparisons in long-horizon tasks
- SPA Pipeline: W29p suggested including sense-plan-act baselines. The authors explained their focus on RL based on prior HAB work and faster wall-time training, while acknowledging SPA's importance for future work.

While the paper's contributions are primarily engineering-focused, they represent important infrastructure work that enables new research directions in realistic robotic manipulation. The authors have thoroughly addressed reviewer concerns through substantial revisions and additional experiments. The benchmark's combination of speed, realistic control, and comprehensive baselines/analysis tools will benefit the broader robotics research community.

**Additional Comments On Reviewer Discussion:**

See "Metareview" for summary.

---

### Decision · Program_Chairs · 2025-01-22

Accept (Poster)